# Towards Discovering Neural Architectures from Scratch

## Abstract

The discovery of neural architectures from scratch is the long-standing goal of Neural Architecture Search (NAS). Searching over a wide spectrum of neural architectures can facilitate the discovery of previously unconsidered but well-performing architectures. In this work, we take a large step towards discovering neural architectures from scratch by expressing architectures algebraically. This algebraic view leads to a more general method for designing search spaces, which allows us to compactly represent search spaces that are 100s of orders of magnitude larger than common spaces from the literature. Further, we propose a Bayesian Optimization strategy to efficiently search over such huge spaces, and demonstrate empirically that both our search space design and our search strategy can be superior to existing baselines. We open source our algebraic NAS approach and provide APIs for PyTorch and TensorFlow.

## 1 Introduction

Neural Architecture Search (NAS), a field with over $1\,000$ papers in the last two years (Deng & Lindauer, 2022), is widely touted to automatically discover novel, well-performing architectural patterns. However, while state-of-the-art performance has already been demonstrated in hundreds of NAS papers (prominently, e.g., (Tan & Le, 2019; 2021; Liu et al., 2019a)), success in automatically finding truly novel architectural patterns has been very scarce (Ramachandran et al., 2017; Liu et al., 2020). For example, novel architectures, such as transformers (Vaswani et al., 2017; Dosovitskiy et al., 2021) have been crafted manually and were *not* found by NAS.

There is an accumulating amount of evidence that over-engineered, restrictive search spaces (e.g., cell-based ones) are major impediments for NAS to discover truly novel architectures. Yang et al. (2020b) showed that in the DARTS search space (Liu et al., 2019b) the manually-defined macro architecture is more important than the searched cells, while Xie et al. (2019) and Ru et al. (2020) achieved competitive performance with randomly wired neural architectures that do not adhere to common search space limitations. As a result, there are increasing efforts to break these impediments, and the discovery of novel neural architectures has been referred to as the holy grail of NAS.

Hierarchical search spaces are a promising step towards this holy grail. In an initial work, Liu et al. (2018) proposed a hierarchical cell, which is shared across a fixed macro architecture, imitating the compositional neural architecture design pattern widely used by human experts. However, subsequent works showed the importance of both layer diversity (Tan & Le, 2019) and macro architecture (Xie et al., 2019; Ru et al., 2020).

In this work, we introduce a general formalism for the representation of hierarchical search spaces, allowing both for layer diversity and a flexible macro architecture. The key observation is that any neural architecture can be represented algebraically; e.g., two residual blocks followed by a fully-connected layer in a linear macro topology can be represented as the algebraic term

$$\omega = \texttt{Linear(Residual(conv, id, conv), Residual(conv, id, conv), fc)} \quad . \quad (1)$$

We build upon this observation and employ Context-Free Grammars (CFGs) to construct large spaces of such *algebraic architecture terms*. Although a particular search space is of course limited in its overall expressiveness, with this approach, we could effectively represent any neural architecture, facilitating the discovery of truly novel ones.

Due to the hierarchical structure of algebraic terms, the number of candidate neural architectures scales exponentially with the number of hierarchical levels, leading to search spaces 100s of orders of magnitudes larger than commonly used ones. To search in these huge spaces, we propose an efficient search strategy, Bayesian Optimization for Algebraic Neural Architecture Terms (BANAT), which leverages hierarchical information, capturing the topological patterns across the hierarchical levels, in its tailored kernel design.

Our contributions are as follows:

- We present a novel technique to construct hierarchical NAS spaces based on an algebraic notion views neural architectures as algebraic architecture terms and CFGs to create algebraic search spaces (Section 2).
- We propose BANAT, a Bayesian Optimization (BO) strategy that uses a tailored modeling strategy to efficiently and effectively search over our huge search spaces (Section 3).
- After surveying related work (Section 4), we empirically show that search spaces of algebraic architecture terms perform on par or better than common cell-based spaces on different datasets, show the superiority of BANAT over common baselines, demonstrate the importance of incorporating hierarchical information in the modeling, and show that we can find novel architectural parts from basic mathematical operations (Section 5).

We open source our code and provide APIs for PyTorch (Paszke et al., 2019) and TensorFlow (Abadi et al., 2015) at `https://anonymous.4open.science/r/iclr23_tdnafs`.

## 2 ALGEBRAIC NEURAL ARCHITECTURE SEARCH SPACE CONSTRUCTION

In this section we present an algebraic view on Neural Architecture Search (NAS) (Section 2.1) and propose a construction mechanism based on Context-Free Grammars (CFGs) (Section 2.2 and 2.3).

### 2.1 ALGEBRAIC ARCHITECTURE TERMS FOR NEURAL ARCHITECTURE SEARCH

We introduce *algebraic architecture terms* as a string representation for neural architectures from a (term) algebra. Formally, an algebra $(A, \mathcal{F})$ consists of a non-empty set $A$ (universe) and a set of operators $f \colon A^n \to A \in \mathcal{F}$ of different arities $n \geq 0$ (Birkhoff, 1935). In our case, $A$ corresponds to the set of all (sub-)architectures and we distinguish between two types of operators: (i) nullary operators representing primitive computations (e.g., `conv()` or `fc()`) and (ii) k-ary operators with $k > 0$ representing topological operators (e.g., `Linear(·, ·, ·)` or `Residual(·, ·, ·)`). For sake of notational simplicity, we omit parenthesis for nullary operators (i.e., we write `conv`). Term algebras (Baader & Nipkow, 1999) are a special type of algebra mapping an algebraic expression to its string representation. E.g., we can represent a neural architecture as the algebraic architecture term $\omega$ as shown in Equation 1. Term algebras also allow for variables $x_i$ that are set to terms themselves that can be re-used across a term. In our case, the *intermediate variables* $x_i$ can therefore share patterns across the architecture, e.g., a shared cell. For example, we could define the intermediate variable $x_1$ to map to the residual block in $\omega$ from Equation 1 as follows:

$$\omega' = \texttt{Linear}(x_1,\ x_1,\ \texttt{fc}), \quad x_1 = \texttt{Residual}(\texttt{conv},\ \texttt{id},\ \texttt{conv}) \quad . \tag{2}$$

**Algebraic NAS** We formulate our algebraic view on NAS, where we search over algebraic architecture terms $\omega \in \Omega$ representing their associated architectures $\Phi(\omega)$, as follows:

$$\underset{\omega \in \Omega}{\arg\min}\, f(\Phi(\omega)) \quad , \tag{3}$$

where $f(\cdot)$ is an error measure that we seek to minimize, e.g., final validation error of a fixed training protocol. For example, we can represent the popular cell-based NAS-Bench-201 search space(Dong & Yang, 2020) as algebraic search space $\Omega$. The algebraic search space $\Omega$ is characterized by a fixed macro architecture `Macro(...)` that stacks 15 instances of a shared cell `Cell(`$p_i, p_i, p_i, p_i, p_i, p_i$`)`, where the cell has six edges, on each of which one of five primitive computations can be placed (i.e., $p_i$ for $i \in \{1,\ 2,\ 3,\ 4,\ 5\}$ corresponding to `zero`, `id`, `conv1x1`, `conv3x3`, or `avg_pool`, respectively). By leveraging the intermediate variable $x_1$

**Algebraic architecture terms**  **Neural architectures**

Figure 1: Derivations from Equation 6 of algebraic architecture terms (left) correspond to edge replacements (Habel & Kreowski, 1983; 1987; Drewes et al., 1997) in the associated architecture (right). Appendix A provides the vocabulary for topological operators and primitive computations.

we can effectively share the cell topology across the architecture. For example, we can express an architecture $\omega_i \in \Omega$ from the NAS-Bench-201 search space $\Omega$ as:

$$\omega_i = \texttt{Macro}(\underbrace{x_1, x_1, ..., x_1}_{15\times}), \quad x_1 = \texttt{Cell}(\texttt{p}_1, \texttt{p}_2, \texttt{p}_1, \texttt{p}_5, \texttt{p}_4, \texttt{p}_3) \quad . \tag{4}$$

Algebraic NAS over such algebraic architecture terms then amounts to finding the best-performing primitive computation $\texttt{p}_i$ for each edge, as the macro architecture is fixed. In contrast to this simple cell-based algebraic space, the search spaces we consider can be much more expressive and, e.g., allow for layer diversity and a flexible macro architecture over several hierarchical levels (Section 5.1).

## 2.2 CONSTRUCTING NEURAL ARCHITECTURE TERMS WITH CONTEXT-FREE GRAMMARS

We propose to use *Context-Free Grammars (CFGs)* (Chomsky, 1956) since they can naturally generate (hierarchical) algebraic architecture terms. Compared to other search space designs, CFGs give us a formally grounded way to naturally and compactly define very expressive hierarchical search spaces (e.g., see Section 5.1). We can also unify popular search spaces from the literature with our general search space design in one framework (Appendix E). They give us further a simple mechanism to evolve architectures while staying within the defined search space (Section 3).

Formally, a CFG $\mathcal{G} = \langle N, \Sigma, P, S \rangle$ consists of a finite set of nonterminals $N$ and terminals $\Sigma$ with $N \cap \Sigma = \emptyset$, a finite set of production rules $P = \{A \to \beta | A \in N, \beta \in (N \cup \Sigma)^*\}$, where the asterisk $*$ denotes the Kleene star operation (Kleene et al., 1956), and a start symbol $S \in N$. To generate an algebraic architecture term, starting from the start symbol $S$, we recursively replace nonterminals of the current algebraic term with a right-hand side of a production rule consisting of nonterminals and terminals, until the resulting string does not contain any nonterminals. For example, consider the following CFG in extended Backus-Naur form (Backus, 1959) (see Appendix B for background):

$$S ::= \texttt{Linear(S, S, S)} \mid \texttt{Residual(S, S, S)} \mid \texttt{conv} \mid \texttt{id} \mid \texttt{fc} \tag{5}$$

From this CFG, we can derive the algebraic architecture term $\omega$ (with three hierarchical levels) from Equation 1 as follows:

$S \to \texttt{Linear(S, S, S)}$  Level 1

$\to \texttt{Linear(Residual(S, S, S), Residual(S, S, S), fc)}$  Level 2
$$\tag{6}$$

$\to \texttt{Linear(Residual(conv, id, conv), Residual(conv, id, conv), fc)}$  Level 3

Figure 1 makes the above derivation and the connection to the associated architecture explicit. The set of all (potentially infinite) algebraic terms generated by a CFG $\mathcal{G}$ is the language $L(\mathcal{G})$, which naturally forms our search space $\Omega$. Thus, the algebraic NAS problem from Equation 3 becomes:

$$\underset{\omega \in L(\mathcal{G})}{\arg\min} f(\Phi(\omega)) \quad . \tag{7}$$

### 2.3 Extensions to the construction mechanism

**Constraints**  In many search space designs, we want to adhere to some constraints, e.g., to limit the number of nodes or to ensure that for all architectures in the search space there exists at least one path from the input to the output. We can simply do so by allowing only the application of production rules which guarantee compliance to such constraints. For example, to ensure that there is at least one path from the input to the output, it is sufficient to ensure that each derivation connects its input to the output due to the recursive nature of CFGs. Note that this makes CFGs context-sensitive w.r.t. those constraints. For more details, please refer to Appendix D.

**Fostering regularity through substitution**  To implement intermediate variables $x_i$ (Section 2.1) we leverage that context-free languages are closed under substitution: we map terminals, representing the intermediate variables $x_i$, from one language to algebraic terms of other languages, e.g., a shared cell. For example, we can split a CFG $\mathcal{G}$, constructing entire algebraic architecture terms, into the CFGs $\mathcal{G}_{macro}$ and $\mathcal{G}_{cell}$ for the macro- or cell-level, respectively. Further, we add a single (or multiple) intermediate terminal(s) $x_1$ to $\mathcal{G}_{macro}$ which maps to an algebraic term $\omega_1 \in L(\mathcal{G}_{cell})$, e.g., the searchable cell. Thus, we effectively search over the macro-level as well as a single, shared cell. Note that by using a fixed macro architecture (i.e., $|L(\mathcal{G}_{macro})| = 1$), we can represent cell-based search spaces, e.g., NAS-Bench-201 (Dong & Yang, 2020), while also being able to represent more expressive search spaces (e.g., see Section 5.1). More generally, we could extend this by adding further intermediate terminals which map to other languages $L(\mathcal{G}_j)$, or by adding intermediate terminals to $\mathcal{G}_2$ which map to languages $L(\mathcal{G}_{j\neq1})$. In this way, we can effectively foster regularity.

**Representing common architecture patterns for object recognition**  Neural architectures for object recognition commonly build a hierarchy of features that are gradually downsampled, e.g., by pooling operations. However, previous works in NAS were either limited to a fixed macro architecture (Zoph et al., 2018), only allowed for linear macro architectures (Liu et al., 2019a), or required post-sampling testing for resolution mismatches (Stanley & Miikkulainen, 2002; Ru et al., 2020). While this produced impressive performance on popular benchmarks (Tan & Le, 2019; 2021; Liu et al., 2019a), it is an open research question whether a different type of macro architecture (e.g., one with multiple branches) could yield even better performance.

To accommodate flexible macro architectures, we propose to overload the nonterminals. In particular, the nonterminals indicate how often we apply downsampling operations in the subsequent derivations of the nonterminal. Consider the production rule D2 $\rightarrow$ Residual(D1, D2, D1), where D$i$ with $i \in \{1, 2\}$ are a nonterminals which indicate that $i$ downsampling operations have to be applied in their subsequent derivations. That is, in both paths of the residual the input features will be downsampled twice and, consequently, the merging paths will have the same spatial resolution. Thereby, this mechanism distributes the downsampling operations recursively across the architecture. For the channels, we adopted the common design to double the number of channels whenever we halve the spatial resolution in our experiments. Note that we could also handle a varying number of channels by using, e.g., depthwise concatenation as merge operation.

## 3 Bayesian Optimization for algebraic neural architecture search

We propose a BO strategy, Bayesian Optimization for Algebraic Neural Architecture Terms (BANAT), to efficiently search in the huge search spaces spanned by our algebraic architecture terms: we introduce a novel surrogate model which combines a Gaussian Process (GP) surrogate with a tailored kernel that leverages the hierarchical structure of algebraic neural architecture terms (see below), and adopt expected improvement as the acquisition function (Mockus et al., 1978). Given the discrete nature of architectures, we adopt ideas from grammar-guided genetic programming (McKay et al., 2010; Moss et al., 2020) for acquisition function optimization. Furthermore, to reduce wallclock time by leveraging parallel computing resources, we adapt the Kriging Believer (Ginsbourger et al., 2010) to select architectures at every search iteration so that we can train and evaluate them in parallel. Specifically, Kriging Believer assigns hallucinated values (i.e., posterior mean) of pending evaluations at each iteration to avoid redundant evaluations. For a more detailed explanation of BANAT, please refer to Appendix F.

**Hierarchical Weisfeiler-Lehman kernel (hWL)**    Inspired by the state-of-the-art BO approach for NAS (Ru et al., 2021), we adopt the WL graph kernel (Shervashidze et al., 2011) in a GP surrogate, modeling performance of the algebraic architecture terms $\omega_i$ with the associated architectures $\Phi(\omega_i)$. However, modeling solely based on the final architecture ignores the useful hierarchical information inherent in our algebraic representation. Moreover, the large size of the architectures also makes it difficult to use a single WL kernel to capture the more global topological patterns.

Since our hierarchical construction can be viewed as a series of gradually unfolding architectures, with the final architecture containing only primitive computations, we propose a novel hierarchical kernel design assigning a WL kernel to each hierarchy and combine them in a weighted sum. To this end, we introduce fold operators $F_l$, that removes algebraic terms beyond the $l$-th hierarchical level. For example, the fold operators $F_1$, $F_2$ and $F_3$ yield for the algebraic term $\omega$ (Equation 1)

$$F_3(\omega) = \omega = \texttt{Linear(Residual(conv, id, conv), Residual(conv, id, conv), fc)},$$
(8)

$$F_2(\omega) = \texttt{Linear(Residual, Residual, fc)} \quad , \quad F_1(\omega) = \texttt{Linear} \quad .$$

Note the similarity to the derivations in Figure 1. Furthermore note that, in practice, we also add the corresponding nonterminals to integrate information from our hierarchical construction process. We define our hierarchical WL kernel (hWL) for two architectures $\Phi(\omega_i)$ and $\Phi(\omega_j)$ with algebraic architecture terms $\omega_i$ or $\omega_j$, respectively, constructed over a hierarchy of $L$ levels, as follows:

$$k_{hWL}(\omega_i, \omega_j) = \sum_{l=2}^{L} \lambda_l \cdot k_{WL}(\Phi(F_l(\omega_i)), \Phi(F_l(\omega_j))) \qquad ,$$
(9)

where the weights $\lambda_l$ govern the importance of the learned graph information at different hierarchical levels (granularities of the architecture) and can be tuned (along with other hyperparameters of the GP) by maximizing the marginal likelihood. We omit $l = 1$ in the additive kernel as $F_1(\omega)$ does not contain any edge features which are required for our WL kernel $k_{WL}$. For more details on our novel hierarchical kernel design, please refer to Appendix F.2. Our proposed kernel efficiently captures the information in all algebraic term construction levels, which substantially improves its search and surrogate regression performance on our search space as demonstrated in Section 5.

**Acquisition function optimization**    To optimize the acquisition function, we adopt ideas from grammar-based genetic programming (McKay et al., 2010; Moss et al., 2020). For mutation, we randomly replace a sub-architecture term with a new randomly generated term, using the same non-terminal as start symbol. For crossover, we randomly swap two sub-architecture terms with the same corresponding nonterminal. We consider two crossover operators: a novel *self-crossover* operation swaps two sub-terms of a *single* architecture term, and the common crossover operation swaps sub-terms of two different architecture terms. Importantly, all evolutionary operations by design only result in valid terms. We provide examples for the evolutionary operations in Appendix F.

## 4    RELATED WORK

We discuss related works in NAS below and discuss works beyond NAS in Appendix G.

**Neural Architecture Search**    Neural Architecture Search (NAS) aims to automatically discover architectural patterns (or even entire architectures) (Elsken et al., 2019). Previous approaches, e.g., used reinforcement learning (Zoph & Le, 2017; Pham et al., 2018), evolution (Real et al., 2017), gradient descent (Liu et al., 2019b), or Bayesian Optimization (BO) (Kandasamy et al., 2018; White et al., 2021; Ru et al., 2021). To enable the effective use of BO on graph-like inputs for NAS, previous works have proposed to use a GP with specialized kernels (Kandasamy et al., 2018; Ru et al., 2021), encoding schemes (Ying et al., 2019; White et al., 2021), or graph neural networks as surrogate model (Ma et al., 2019; Shi et al., 2020; Zhang et al., 2019). Different to prior works, we explicitly leverage the hierarchical construction of architectures for modeling.

**Searching for novel architectural patterns**    Previous works mostly focused on finding a shared cell (Zoph et al., 2018) with a fixed macro architecture while only few works considered more expressive hierarchical search spaces (Liu et al., 2018; 2019a; Tan et al., 2019). The latter works

considered hierarchical assembly (Liu et al., 2018), combination of a cell- and network-level search space (Liu et al., 2019a; Zhang et al., 2020), evolution of network topologies (Miikkulainen et al., 2019), factorization of the search space (Tan et al., 2019), parameterization of a hierarchy of random graph generators (Ru et al., 2020), a formal language over computational graphs (Negrinho et al., 2019), or a hierarchical construction of TensorFlow programs (So et al., 2021). Similarly, our formalism allows to design search spaces covering a general set of architecture design choices, but also permits the search for macro architectures with spatial resolution changes and multiple branches. We also handle spatial resolution changes without requiring post-hoc testing or resizing of the feature maps unlike prior works (Stanley & Miikkulainen, 2002; Miikkulainen et al., 2019; Stanley et al., 2019). Other works proposed approaches based on string rewriting systems (Kitano, 1990; Boers et al., 1993), cellular (or tree-structured) encoding schemes (Gruau, 1994; Luke & Spector, 1996; De Jong & Pollack, 2001; Cai et al., 2018), hyperedge replacement graph grammars Luerssen & Powers (2003); Luerssen (2005), attribute grammars (Mouret & Doncieux, 2008), CFGs (Jacob & Rehder, 1993; Couchet et al., 2007; Ahmadizar et al., 2015; Ahmad et al., 2019; Assunção et al., 2017; 2019; Lima et al., 2019; de la Fuente Castillo et al., 2020), or And-Or-grammars (Li et al., 2019). Different to these prior works, we construct entire architectures with spatial resolution changes across multiple branches, and propose techniques to incorporate constraints and foster regularity. Orthogonal to the aforementioned approaches, Roberts et al. (2021) searched over neural (XD-)operations, which is orthogonal to our approach, i.e., our predefined primitive computations could be replaced by their proposed XD-operations.

## 5 EXPERIMENTS

In this section, we investigate potential benefits of hierarchical search spaces and our search strategy BANAT. More specifically, we address the following questions:

**Q1** Can hierarchical search spaces yield on par or superior architectures compared to cell-based search spaces with a limited number of evaluations?

**Q2** Can our search strategy BANAT improve performance over common baselines?

**Q3** Does leveraging the hierarchical information improve performance?

**Q4** Do zero-cost proxies work in vast hierarchical search spaces?

**Q5** Can we discover novel architectural patterns (e.g., activation functions)?

To answer questions **Q1**-**Q4**, we introduce a hierarchical search space based on the popular NAS-Bench-201 search space (Dong & Yang, 2020) in Section 5.1. To answer question **Q5**, we search for activation functions (Ramachandran et al., 2017) and defer the search space definition to Appendix J.1. We provide complementary results and analyses in Appendix I.2 and J.3.

### 5.1 HIERARCHICAL NAS-BENCH-201

We propose a hierarchical variant of the popular cell-based NAS-Bench-201 search space (Dong & Yang, 2020) by adding a hierarchical macro space (i.e., spatial resolution flow and wiring at the macro-level) and parameterizable convolutional blocks (i.e., choice of convolutions, activations, and normalizations). We express the hierarchical NAS-Bench-201 search space with CFG $\mathcal{G}_h$ as follows:

$$
\begin{aligned}
\text{D2} \;&::=\; \texttt{Linear3(D1, D1, D0)} \mid \texttt{Linear3(D0, D1, D1)} \mid \texttt{Linear4(D1, D1, D0, D0)} \\
\text{D1} \;&::=\; \texttt{Linear3(C, C, D)} \mid \texttt{Linear4(C, C, C, D)} \mid \texttt{Residual3(C, C, D, D)} \\
\text{D0} \;&::=\; \texttt{Linear3(C, C, CL)} \mid \texttt{Linear4(C, C, C, CL)} \mid \texttt{Residual3(C, C, CL, CL)} \\
\text{D} \;&::=\; \texttt{Linear2(CL, down)} \mid \texttt{Linear3(CL, CL, down)} \mid \texttt{Residual2(C, down, down)} \\
\text{C} \;&::=\; \texttt{Linear2(CL, CL)} \mid \texttt{Linear3(CL, CL)} \mid \texttt{Residual2(CL, CL, CL)} \\
\text{CL} \;&::=\; \texttt{Cell(OP, OP, OP, OP, OP, OP)} \\
\text{OP} \;&::=\; \texttt{zero} \mid \texttt{id} \mid \texttt{BLOCK} \mid \texttt{avg\_pool} \\
\text{BLOCK} \;&::=\; \texttt{Linear3(ACT, CONV, NORM)} \\
\text{ACT} \;&::=\; \texttt{relu} \mid \texttt{hardswish} \mid \texttt{mish} \\
\text{CONV} \;&::=\; \texttt{conv1x1} \mid \texttt{conv3x3} \mid \texttt{dconv3x3} \\
\text{NORM} \;&::=\; \texttt{batch} \mid \texttt{instance} \mid \texttt{layer} \;\;.
\end{aligned}
$$

$$(10)$$

See Appendix A for the terminal vocabulary of topological operators and primitive computations. The productions with the nonterminals {D2, D1, D0, D} define the spatial resolution flow and together with {C} define the macro architecture containing possibly multiple branches. The productions for {CL, OP} construct the NAS-Bench-201 cell and {BLOCK, ACT, CONV, NORM} parameterize the convolutional block. To ensure that we use the same distribution over the primitive computations as in NAS-Bench-201, we reweigh the sampling probabilities of the productions generated by the nonterminal OP, i.e., all production choices have sampling probability of $20\%$, but BLOCK has $40\%$. Note that we omit the stem (i.e., 3x3 convolution followed by batch normalization) and classifier (i.e., batch normalization followed by ReLU, global average pooling, and fully-connected layer) for simplicity. We implemented the merge operation as element-wise summation. Different to the cell-based NAS-Bench-201 search space, we exclude degenerated architectures by introducing a constraint that ensures that each subterm maps the input to the output (i.e., in the associated computational graph there is at least one path from source to sink).

Our search space consists of ca. $\mathbf{10^{446}}$ algebraic architecture terms (please refer to Appendix C on how to compute the search space size), which is significantly larger than other popular search spaces from the literature. For comparison, the cell-based NAS-Bench-201 search space is just a minuscule subspace of size $10^{4.18}$, where we apply only the blue-colored production rules and replace the CL nonterminals with a placeholder terminal $x_1$ that will be substituted by the searched, shared cell.

## 5.2 EVALUATION DETAILS

For all search experiments, we compared the search strategies BANAT, Random Search (RS), Regularized Evolution (RE) (Real et al., 2019; Liu et al., 2018), and BANAT (WL) (Ru et al., 2021). For implementation details of the search strategies, please refer to Appendix H. We ran search for a total of 100 evaluations with a random initial design of 10 on three seeds {777, 888, 999} on the hierarchical NAS-Bench-201 search space or 1000 evaluations with a random initial design of 50 on one seed {777} on the activation function search space using 8 asynchronous workers each with a single NVIDIA RTX 2080 Ti GPU. In each evaluation, we fully trained the architectures and recorded their last validation error. For training details on the hierarchical NAS-Bench-201 search space and activation function search space, please refer to Appendix I.1 or Appendix J.2, respectively.

To assess the modeling performance of our surrogate, we compared regression performance of GPs with different kernels, i.e., our hierarchical WL kernel (hWL), (standard) WL kernel (Ru et al., 2021), and NASBOT's kernel (Kandasamy et al., 2018). We also tried the GCN encoding (Shi et al., 2020) but it could not capture the mapping from the complex graph space to performance, resulting in constant performance predictions. Further, note that the adjacency encoding (Ying et al., 2019) and path encoding (White et al., 2021) cannot be used in our hierarchical search spaces since the former requires the same amount of nodes across graphs and the latter scales exponentially in the number of nodes. We ran 20 trials over the seeds {0, 1, ..., 19} and re-used the data from the search runs. In every trial, we sampled a training and test set of 700 or 500 architecture and validation error pairs, respectively. We fitted the surrogates with a varying number of training samples by randomly choosing samples from the training set without replacement, and recorded Kendall's $\tau$ rank correlation between the predicted and true validation error. To assess zero-cost proxies, we re-used the data from the search runs and recorded Kendall's $\tau$ rank correlation.

## 5.3 RESULTS

In the following we answer all of the questions **Q1-Q5**. Figure 2 compares the results of the cell-based and hierarchical search space design using our search strategy BANAT. Results with BANAT are on par on CIFAR-10/100, superior on ImageNet-16-120, and clearly superior on CIFARTile and AddNIST (answering **Q1**). We emphasize that the NAS community has engineered the cell-based search space to achieve strong performance on those popular image classification datasets for over a decade, making it unsurprising that our improvements are much larger for the novel datasets. Yet, our best found architecture on ImageNet-16-120 from the hierarchical search space also achieves an excellent test error of $52.78\%$ with only $0.626$ MB parameters (Appendix I.2); this is superior to the architecture found by the state-of-the-art method Shapley-NAS (i.e., $53.15\%$) (Xiao et al., 2022) and on par with the *optimal* architecture of the cell-based NAS-Bench-201 search space (i.e., $52.69\%$ with $0.866$ MB). Figure 3 shows that our search strategy BANAT is also superior

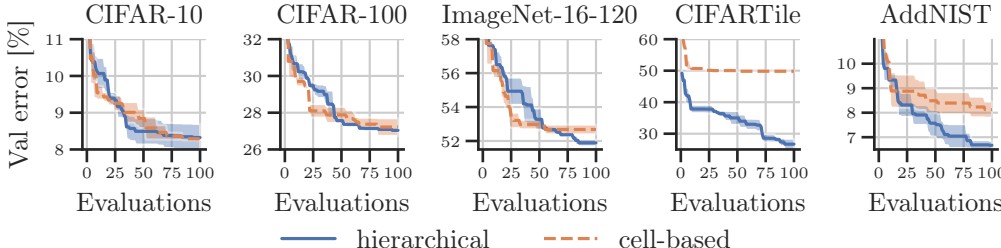

Figure 2: Cell-based vs. hierarchical search space. We plot mean and $\pm 1$ standard error of the validation error on the hierarchical (solid blue) and cell-based (dashed orange) NAS-Bench-201 search space. We compare results using the other search strategies in Appendix I.2.

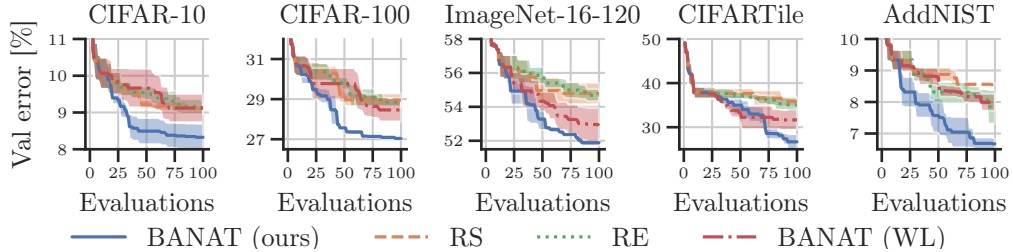

Figure 3: Comparison of search strategies on the hierarchical search space. We plot mean and $\pm 1$ standard error of the validation error on the hierarchical NAS-Bench-201 search space for our search strategy BANAT (solid blue), RS (dashed orange), RE (dotted green), and BANAT (WL) (dash-dotted red). We report test errors, best architectures, and conduct further analyses in Appendix I.2.

to common baselines (answering **Q2**) and leveraging hierarchical information clearly improves performance (answering **Q3**). Further, the evaluation of surrogate performance in Figure 4 shows that incorporating hierarchical information with our hierarchical WL kernel (hWL) improves modeling, especially on smaller amounts of training data (further answering **Q3**). Table 1 shows that the baseline zero-cost proxies `flops` and `l2-norm` yield competitive (or often superior) results to more sophisticated zero-cost proxies; making hierarchical search spaces an interesting future research direction for them (answering **Q4**). Finally, Table 2 shows that we can find novel well-performing activation functions from basic mathematical operations with BANAT (answering **Q5**).

## 6 DISCUSSION AND LIMITATIONS

While our grammar-based construction mechanism is a powerful mechanism to construct huge hierarchical search space, we can not construct any architecture with our grammar-based construction approach (Section 2.2 and 2.3) since we are limited to context-free languages; e.g., architectures of the type $\{a^n b^n c^n | n \in \mathbb{N}_{>0}\}$ cannot be generated by CFGs (this can be proven using Odgen's lemma (Ogden, 1968)). Further, due to the discrete nature of CFGs we can not easily integrate continuous design choices, e.g., dropout probability. Furthermore, our grammar-based mechanism does not (generally) support simple scalability of discovered neural architectures (e.g., repetition of building blocks) without special consideration in the search space design. Nevertheless, our search spaces still significantly increase the expressiveness, including the ability to

| Search strategy | Test error [%] |
|---|---|
| ReLU | 8.93 |
| Swish | 8.61 |
| RS | 8.91 |
| RE | 8.47 |
| BANAT (WL) | 8.32 |
| BANAT (ours) | **8.31** |

Table 2: Results of the activation function search on CIFAR-10 with ResNet-20.

represent common search spaces from the literature (see Appendix E for how we can represent the search spaces of DARTS, Auto-Deeplab, the hierarchical cell search space of Liu et al. (2018), the Mobile-net search space, and the hierarchical random graph generator search space), as well as allowing search for entire neural architectures based around the popular NAS-Bench-201 search space

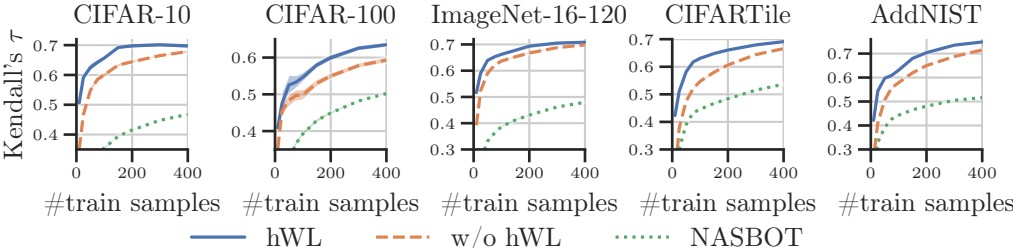

Figure 4: Mean Kendall's $\tau$ rank correlation with $\pm 1$ standard error achieved by a GP with our hierarchical WL kernel (hWL), (standard) WL kernel (WL), and NASBOT (Kandasamy et al., 2018).

Table 1: Kendall's $\tau$ rank correlation of zero-cost-proxies on our hierarchical NAS-Bench-201 space. Note that `zen` and `nwot` may not be suitable for our hierarchical search space, as they are specifically designed for architectures with batch normalization and ReLU non-linearity, respectively. We still report exemplary search results for NASWOT (Mellor et al., 2021) in Appendix I.2 since it is the best non-baseline zero-cost proxy across the datasets.

| Zero-cost proxy | CIFAR-10 | CIFAR-100 | ImageNet16-120 | CIFARTile | AddNIST |
|---|---|---|---|---|---|
| plain (Abdelfattah et al., 2021) | -0.01 | 0.01 | -0.08 | -0.0 | 0.04 |
| params (Ning et al., 2021) | 0.39 | 0.31 | 0.13 | 0.28 | 0.5 |
| flops (Ning et al., 2021) | 0.46 | **0.51** | 0.47 | **0.47** | 0.49 |
| l2-norm (Abdelfattah et al., 2021) | 0.4 | 0.29 | 0.23 | 0.34 | **0.51** |
| zen-score (Lin et al., 2021) | **0.47** | 0.41 | 0.34 | 0.24 | 0.4 |
| fisher (Turner et al., 2020) | -0.06 | -0.03 | -0.05 | 0.03 | 0.2 |
| grad-norm (Abdelfattah et al., 2021) | 0.09 | 0.04 | 0.01 | 0.15 | 0.24 |
| grasp (Wang et al., 2020) | 0.08 | 0.17 | 0.19 | 0.02 | 0.03 |
| snip (Lee et al., 2019) | 0.17 | 0.06 | -0.01 | 0.21 | 0.29 |
| synflow (Tanaka et al., 2020) | 0.06 | 0.24 | 0.28 | -0.18 | -0.08 |
| epe-nas (Lopes et al., 2021) | 0.34 | 0.29 | 0.26 | 0.23 | 0.09 |
| jacov (Mellor et al., 2021) | 0.4 | 0.34 | 0.34 | 0.25 | 0.12 |
| nwot (Mellor et al., 2021) | 0.32 | 0.44 | **0.58** | 0.34 | 0.22 |

(Section 5). Thus, our search space design can facilitate the discovery of novel well-performing neural architectures in those huge search spaces of algebraic architecture terms.

However, there is an inherent trade-off between the expressiveness and the difficulty of search. The much greater expressiveness facilitates search in a richer set of architectures that may include better architectures than in more restrictive search spaces, which however need not exist. Besides that, the (potential) existence of such a well-performing architecture does not result in a search strategy discovering it, even with large amounts of computing power available. Note that the trade-off manifests itself also in the acquisition function optimization of our search strategy BANAT.

In addition, a well-performing neural architecture may not work with current training protocols and hyperparameters due to interaction effects, i.e., training protocols and hyperparameters may be over-optimized for specific types of neural architectures. To overcome this limitation, one could consider a joint optimization of neural architectures, training protocols, and hyperparameters. However, this further fuels the trade-off between expressiveness and the difficulty of search.

# 7 CONCLUSION

We introduced very expressive search spaces of algebraic architecture terms constructed with CFGs. To efficiently search over the huge search spaces, we proposed BANAT, an efficient BO strategy with a tailored kernel leveraging the available hierarchical information. Our experiments indicate that both our search space design and our search strategy can yield strong performance over existing baselines. Our results motivate further steps towards the discovery of neural architectures based on even more atomic primitive computations. Furthermore, future works could (simultaneously) learn the search space (i.e., learn the grammar) or improve search efficiency by means of multi-fidelity optimization or gradient-based search strategies.

## REPRODUCIBILITY STATEMENT

To ensure reproducibility, we address all points of the best practices checklist for NAS research (Lindauer & Hutter, 2020) in Appendix K.

## ETHICS STATEMENT

NAS has immense potential to facilitate systematic, automated discovery of high-performing (novel) architecture designs. However, the restrictive cell-based search spaces most commonly used in NAS render it impossible to discover truly novel neural architectures. With our general formalism based on algebraic terms, we hope to provide fertile foundation towards discovering high-performing and efficient architectures; potentially from scratch. However, search in such huge search spaces is expensive, particularly in the context of the ongoing detrimental climate crisis. While on the one hand, the discovered neural architectures, like other AI technologies, could potentially be exploited to have a negative societal impact; on the other hand, our work could also lead to advances across scientific disciplines like healthcare and chemistry.

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

## A FROM TERMINALS TO PRIMITIVE COMPUTATIONS AND TOPOLOGICAL OPERATORS

Table 3 and Figure 5 describe the primitive computations and topological operators used throughout our experiments in Section 5 and Appendix I, respectively. Note that by adding more primitive computations and/or topological operators we could construct even more expressive search spaces.

Table 3: Primitive computations. "Name" corresponds to the string terminals in our CFGs and "Function" is the associated implementation of the primitive computation in pseudocode. The subscripts $g$, $k$, $s$, and $p$ are abbreviations for groups, kernel size, strides, and padding, respectively. During assembly of neural architectures $\Phi(\omega)$, we replace string terminals with the associated primitive computation.

| Name | Function |
|------|----------|
| `avg_pool` | `AvgPool`$_{k=3,s=1,p=1}$`(x)` |
| `batch` | `BN(x)` |
| `conv1x1` | `Conv`$_{k=1,s=1,p=0}(x)$ |
| `conv3x3` | `Conv`$_{k=3,s=1,p=1}(x)$ |
| `dconv3x3` | `Conv`$_{g=C,k=3,s=1,p=1}(x)$ |
| `down` | `conv3x3`$_{s=1}$`(conv3x3`$_{s=2}$`(x))` + `Conv`$_{k=1,s=1}$`(AvgPool`$_{k=2,s=2}$`(x))` |
| `hardswish` | `Hardswish(x)` |
| `id` | `Identitiy(x)` |
| `instance` | `IN(x)` |
| `layer` | `LN(x)` |
| `mish` | `Mish(x)` |
| `relu` | `ReLU(x)` |
| `zero` | `Zeros(x)` |

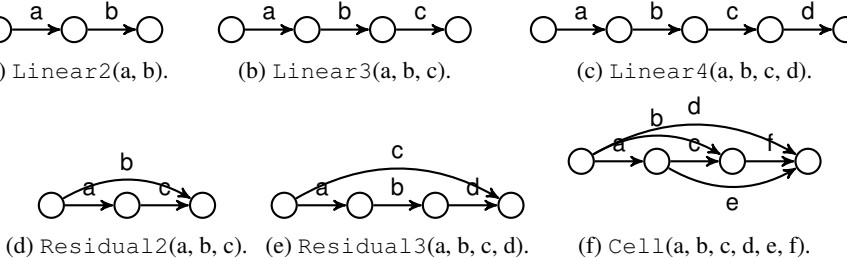

(a) `Linear2`(a, b).    (b) `Linear3`(a, b, c).    (c) `Linear4`(a, b, c, d).

(d) `Residual2`(a, b, c).   (e) `Residual3`(a, b, c, d).   (f) `Cell`(a, b, c, d, e, f).

Figure 5: Topological operators. Each subfigure makes the connection between the topolocial operator and associated computational graph explicit, i.e., the arguments of the graph operators (a, b, ...) are mapped to the respective edges in the computational graph.

## B EXTENDED BACKUS-NAUR FORM

The (extended) Backus-Naur form (Backus, 1959) is a meta-language to describe the syntax of CFGs. We use meta-rules of the form S ::= $\alpha$ where S $\in$ $N$ is a nonterminal and $\alpha$ $\in$ $(N \cup \Sigma)^*$ is a string of nonterminals and/or terminals. We denote nonterminals in UPPER CASE, terminals corresponding to topological operators in `Initial upper case/teletype`, and terminals corresponding to primitive computations in `lower case/teletype`, e.g., S ::= `Residual(S, S, id)`. To compactly express production rules with the same left-hand side nonterminal, we use the vertical bar | to indicate a choice of production rules with the same left-hand side, e.g., S ::= `Linear(S, S, S)` | `Residual(S, S, id)` | `conv`.

## C  SEARCH SPACE SIZE

In this section, we show how to efficiently compute the size of our search spaces constructed by CFGs. There are two cases to consider: (i) a CFG contains cycles (i.e., part of the derivation can be repeated infinitely many times) , yielding an open-ended, infinite search space; and (ii) a CFG contains no cycles, yielding in a finite search space whose size we can compute.

Consider a production A $\rightarrow$ Residual(B, B, B) where Residual is a terminal, and A and B are nonterminals with B $\rightarrow$ conv | id. Consequently, there are $2^3 = 8$ possible instances of the residual block. If we add another production choice for the nonterminal A, e.g., A $\rightarrow$ Linear(B, B, B), we would have $2^3 + 2^3 = 16$ possible instances. Further, adding a production $C \rightarrow$ Linear(A, A, A) would yield a search space size of $(2^3 + 2^3)^3 = 4096$.

More generally, we introduce the function $P_A$ that returns the set of productions for nonterminal A $\in N$, and the function $\mu : P \rightarrow N$ that returns all the nonterminals for a production $p \in P$. We can then recursively compute the size of the search space as follows:

$$f(A) = \sum_{p \in P_A} \left\{ \begin{array}{ll} 1 & , \mu(p) = \emptyset, \\ \prod_{A' \in \mu(p)} f(A') & , otherwise \end{array} \right. . \tag{11}$$

When a CFG contains some constraint, we ensure to only account for valid architectures (i.e., compliant with the constraints) by ignoring productions which would lead to invalid architectures.

## D  MORE DETAILS ON SEARCH SPACE CONSTRAINTS

During the design of the search space, we may want to comply with some constraints, e.g., only consider valid neural architectures or impose structural constraints on architectures. We can guarantee compliance with constraints by modifying sampling (and evolution): we only allow the application of production rules, which guarantee compliance with the constraint(s). In the following, we show exemplary how this can be implemented for the former constraint mentioned above. Note that other constraints can be implemented in a similar manner

To implement the constraint "only consider valid neural architectures", we note that our search space design only creates neural architectures where neither the spatial resolution nor the channels can be mismatched; please refer to Section 2.3 for details. Thus, the only way a neural architecture can become invalid is through zero operations, which could remove edges from the computational graph and possibly disassociate the input from the output. Since we recursively assemble neural architectures, it is sufficient to ensure that the derived algebraic architecture term (i.e., the associated computational graph) is compliant with the constraint, i.e.,there is at least one path from input to output. Thus, during sampling (and similarly during evolution), we modify the current production rule choices when an application of the zero operation would disassociate the input from the output.

## E  COMMON SEARCH SPACES FROM THE LITERATURE

In Section 5.1, we demonstrated how to construct the popular NAS-Bench-201 search space within our algebraic search space design, and below we show how to reconstruct the following popular search spaces: DARTS search space (Liu et al., 2019b), Auto-DeepLab search space (Liu et al., 2019a), hierarchical cell search space (Liu et al., 2018), Mobile-net search space (Tan et al., 2019), and hierarchical random graph generator search space (Ru et al., 2020). For implementation details we refer to the respective works.

### DARTS SEARCH SPACE

The DARTS search space (Liu et al., 2019b) consists of a fixed macro architecture and a cell, i.e., a seven node directed acyclic graph (Darts; see Figure 6 for the topological operator). We omit the fixed macro architecture from our search space design for simplicity. Each cell receives the feature maps from the two preceding cells as input and outputs a single feature map. All intermediate nodes

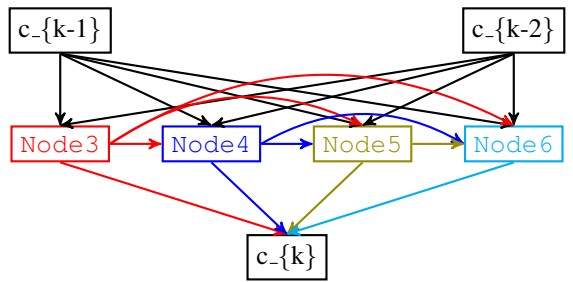

Figure 6: Visualization of the `Darts` topological operator.

(i.e., `Node3`, `Node4`, `Node5`, and `Node6`) is computed based on all of its predecessors. Thus, we can define the DARTS search space as follows:

$$
\begin{aligned}
\text{DARTS} \ &::= \ \texttt{Darts}(\text{NODE3, NODE4, NODE5, NODE6}) \\
\text{NODE3} \ &::= \ \texttt{Node3}(\text{OP, OP}) \\
\text{NODE4} \ &::= \ \texttt{Node4}(\text{OP, OP, OP}) \\
\text{NODE5} \ &::= \ \texttt{Node5}(\text{OP, OP, OP, OP}) \\
\text{NODE6} \ &::= \ \texttt{Node6}(\text{OP, OP, OP, OP, OP}) \\
\text{OP} \ &::= \ \texttt{sep\_conv\_3x3} \mid \texttt{sep\_conv\_5x5} \mid \texttt{dil\_conv\_3x3} \mid \texttt{dil\_conv\_5x5} \\
&\quad\ \mid \texttt{max\_pool} \mid \texttt{avg\_pool} \mid \texttt{id} \mid \texttt{zero} \ ,
\end{aligned}
\tag{12}
$$

where the topological operator `Node3` receives two inputs, applies the operations separately on them, and sums them up. Similarly, `Node4`, `Node5`, and `Node6` apply their operations separately to the given inputs and sum them up. The topological operator `Darts` feeds the corresponding feature maps into each of those topological operators and finally concatenates all intermediate feature maps.

AUTO-DEEPLAB SEARCH SPACE

Auto-DeepLab (Liu et al., 2019a) combines a cell-level with a network-level search space to search for segmentation networks, where the cell is shared across the searched macro architecture, i.e., a twelve step (linear) path across different spatial resolutions. The cell-level design is adopted from Liu et al. (2019b) and, thus, we can re-use the CFG from Equation 12. For the network-level, we introduce a constraint that ensures that the path is of length twelve, i.e., we ensure exactly twelve derivations in our CFG. Further, we overload the nonterminals so that they correspond to the respective spatial resolution level, e.g., D4 indicates that the original input is downsampled by a factor of four; please refer to Section 2.3 for details on overloading nonterminals. For the sake of simplicity, we omit the first two layers and atrous spatial pyramid poolings as they are fixed, and hence define the network-level search space as follows:

$$
\begin{aligned}
\text{D4} \ &::= \ \texttt{Same}(\text{CELL, D4}) \mid \texttt{Down}(\text{CELL, } D8) \\
\text{D8} \ &::= \ \texttt{Up}(\text{CELL, D4}) \mid \texttt{Same}(\text{CELL, D8}) \mid \texttt{Down}(\text{CELL, D16}) \\
\text{D16} \ &::= \ \texttt{Up}(\text{CELL, D8}) \mid \texttt{Same}(\text{CELL, D16}) \mid \texttt{Down}(\text{CELL, D32}) \\
\text{D32} \ &::= \ \texttt{Up}(\text{CELL, D16}) \mid \texttt{Same}(\text{CELL, D32}) \ ,
\end{aligned}
\tag{13}
$$

where the topological operators `Up`, `Same`, and `Down` upsample/halve, do not change/do not change, or downsample/double the spatial resolution/channels, respectively. The placeholder variable `CELL` maps to the shared DARTS cell from the language generated by the CFG from Equation 12.

HIERARCHICAL CELL SEARCH SPACE

The hierarchical cell search space (Liu et al., 2018) consists of a fixed (linear) macro architecture and a hierarchically assembled cell with three levels which is shared across the macro architecture. Thus, we can omit the fixed macro architecture from our search space design for simplicity. Their first, second, and third hierarchical levels correspond to the primitive computations (i.e., `id`, `max_pool`, `avg_pool`, `sep_conv`, `depth_conv`, `conv`, `zero`), six densely connected four node directed acyclic graphs (`DAG4`), and a densely connected five node directed acyclic graph (`DAG5`), respectively. The `zero` operation could lead to directed acyclic graphs which have fewer nodes. Therefore,

we introduce a constraint enforcing that there are always four (level 2) or five (level 3) nodes for every directed acyclic graph. Further, since a densely connected five node directed acyclic graph graph has ten edges, we need to introduce placeholder variables (i.e., `M1`, ..., `M6`) to enforce that only six (possibly) different four node directed acyclic graphs are used, and consequently define a CFG for the third level

$$\text{LEVEL3} ::= \texttt{DAG5}(\underbrace{\text{LEVEL2, ..., LEVEL2}}_{\times 10})$$

$$\text{LEVEL2} ::= \texttt{M1 | M2 | M3 | M4 | M5 | M6 | zero} \quad , \tag{14}$$

mapping the placeholder variables `M1`, ..., `M6` to the six lower-level motifs constructed by the first and second hierarchical level

$$\text{LEVEL2} ::= \texttt{DAG4}(\underbrace{\text{LEVEL1, ..., LEVEL1}}_{\times 6})$$

$$\text{LEVEL1} ::= \texttt{id | max\_pool | avg\_pool | sep\_conv | depth\_conv | conv | zero} \quad . \tag{15}$$

### MOBILE-NET SEARCH SPACE

Factorized hierarchical search spaces, e.g., the Mobile-net search space (Tan et al., 2019), allow for layer diversity. They factorize a (fixed) macro architecture – often based on an already well-performing reference architecture – into separate blocks (e.g., cells). For the sake of simplicity, we assume here a three sequential blocks (`Block`) architecture (`Linear`). In each of those blocks, we search for the convolution operations (CONV), kernel sizes (KSIZE), squeeze-and-excitation ratio (SERATIO) (Hu et al., 2018), skip connections (SKIP), number of output channels (FSIZE), and number of layers per block (#LAYERS), where the latter two are discretized using a reference architecture, e.g., MobileNetV2 (Sandler et al., 2018). Consequently, we can express this search space as follows:

$$
\begin{aligned}
\text{MACRO} &::= \texttt{Linear}(\text{BLOCK, BLOCK, BLOCK}) \\
\text{BLOCK} &::= \texttt{Block}(\text{CONV, KSIZE, SERATIO, SKIP, FSIZE, \#LAYERS}) \\
\text{CONV} &::= \texttt{conv | dconv | mbconv} \\
\text{KSIZE} &::= \texttt{3 | 5} \\
\text{SERATIO} &::= \texttt{0 | 0.25} \\
\text{SKIP} &::= \texttt{pooling | id\_residual | no\_skip} \\
\text{FSIZE} &::= \texttt{0.75 | 1.0 | 1.25} \\
\text{\#LAYERS} &::= \texttt{-1 | 0 | 1} \quad ,
\end{aligned}
\tag{16}
$$

where `conv`, `donv` and `mbconv` correspond to convolution, depthwise convolution, and mobile inverted bottleneck convolution (Sandler et al., 2018), respectively.

### HIERARCHICAL RANDOM GRAPH GENERATOR SEARCH SPACE

The hierarchical random graph generator search space (Ru et al., 2020) consists of three hierarchical levels of random graph generators (i.e., `Watts-Strogatz` (Watts & Strogatz, 1998) and `Erdős-Rényi` (Erdős et al., 1960)). We denote with `Watts-Strogatz_i` the random graph generated by the Watts-Strogatz model with i nodes. Thus, we can represent the search space as follows:

$$
\begin{aligned}
\text{TOP} &::= \texttt{Watts-Strogatz\_3}(\text{K, Pt})(\text{MID, MID, MID}) \mid ... \\
&\quad \mid \texttt{Watts-Strogatz\_10}(\text{K, Pt})(\underbrace{\text{MID, ..., MID}}_{\times 10}) \\
\text{MID} &::= \texttt{Erdős-Rényi\_1}(\text{Pm})(\text{BOT}) \mid ... \\
&\quad \mid \texttt{Erdős-Rényi\_10}(\text{Pm})(\underbrace{\text{BOT, ..., BOT}}_{\times 10}) \\
\text{BOT} &::= \texttt{Watts-Strogatz\_3}(\text{K, Pb})(\text{NODE, NODE, NODE}) \mid ... \\
&\quad \mid \texttt{Watts-Strogatz\_10}(\text{K, Pb})(\underbrace{\text{NODE ..., NODE}}_{\times 10}) \\
\text{K} &::= \texttt{2 | 3 | 4 | 5} \quad ,
\end{aligned}
\tag{17}
$$

---

**Algorithm 1** Bayesian Optimization algorithm (Brochu et al., 2010).

---

**Input:** Initial observed data $\mathcal{D}_t$, a black-box objective function $f$, total number of BO iterations $T$
**Output:** The best recommendation about the global optimizer $\mathbf{x}^*$
**for** $t = 1, \ldots, T$ **do**
    Select the next $\mathbf{x}_{t+1}$ by maximizing acquisition function $\alpha(\mathbf{x}|\mathcal{D}_t)$
    Evaluate the objective function at $f_{t+1} = f(\mathbf{x}_{t+1})$
    $\mathcal{D}_{t+1} \leftarrow \mathcal{D}_t \cup (\mathbf{x}_{t+1}, f_{t+1})$
    Update the surrogate model with $\mathcal{D}_{t+1}$
**end for**

---

where each terminal `Pt`, `Pm`, and `Pb` maps to a continuous number in $[0.1, \ 0.9]$[1] and the placeholder variable `NODE` maps to a primitive computation, e.g., separable convolution. Note that we omit other hyperparameters, such as stage ratio, channel ratio etc., for simplicity.

## F    More details on the search strategy

In this section, we provide more details and examples for our search strategy Bayesian Optimization for Algebraic Neural Architecture Terms (BANAT) presented in Section 3.

### F.1    Bayesian Optimization

Bayesian Optimization (BO) is a powerful family of search techniques for finding the global optimum of a black-box objective problem. It is particularly useful when the objective is expensive to evaluate and thus sample efficiency is highly important (Brochu et al., 2010).

To minimize a black-box objective problem with BO, we first need to build a probabilistic surrogate to model the objective based on the observed data so far. Based on the surrogate model, we design an acquisition function to evaluate the utility of potential candidate points by trading off exploitation (where the posterior mean of the surrogate model is low) and exploration (where the posterior variance of the surrogate model is high). The next candidate points to evaluate is then selected by maximizing the acquisition function (Shahriari et al., 2015). The general procedures of BO is summarized in Algorithm 1.

We adopted the widely used acquisition function, expected improvement (EI) (Mockus et al., 1978), in our BO strategy. EI evaluates the expected amount of improvement of a candidate point $\mathbf{x}$ over the minimal value $f'$ observed so far. Specifically, denote the improvement function as $I(\mathbf{x}) = \max(0, f' - f(\mathbf{x}))$, the EI acquisition function has the form

$$\alpha_{EI}(\mathbf{x}|\mathcal{D}_t) = \mathbb{E}[I(\mathbf{x})|\mathcal{D}_t] = \int_{-\infty}^{f'} (f' - f)\mathcal{N}\left(f; \mu(\mathbf{x}|\mathcal{D}_t), \sigma^2(\mathbf{x}|\mathcal{D}_t)\right) df$$
$$= (f' - f)\Phi\left(f'; \mu(\mathbf{x}|\mathcal{D}_t), \sigma^2(\mathbf{x}|\mathcal{D}_t)\right) + \sigma^2(\mathbf{x}|\mathcal{D}_t)\phi(f'; \mu(\mathbf{x}|\mathcal{D}_t), \sigma^2(\mathbf{x}|\mathcal{D}_t)) \quad,$$

where $\mu(\mathbf{x}|\mathcal{D}_t)$ and $\sigma^2(\mathbf{x}|\mathcal{D}_t)$ are the mean and variance of the predictive posterior distribution at a candidate point $\mathbf{x}$, and $\phi(\cdot)$ and $\Phi(\cdot)$ denote the PDF and CDF of the standard normal distribution, respectively.

To make use of ample distributed computing resource, we adopted Kriging Believer (Ginsbourger et al., 2010) which uses the predictive posterior of the surrogate model to assign hallucinated function values $\{\tilde{f}_p\}_{p \in \{1, \ldots, P\}}$ to the $P$ candidate points with pending evaluations $\{\tilde{\mathbf{x}}_p\}_{p \in \{1, \ldots, P\}}$ and perform next BO recommendation in the batch by pseudo-augmenting the observation data with $\tilde{\mathcal{D}}_p = \{(\tilde{\mathbf{x}}_p, \tilde{f}_p)\}_{p \in \{1, \ldots, P\}}$, namely $\tilde{\mathcal{D}}_t = \mathcal{D}_t \cup \tilde{\mathcal{D}}_p$. The algorithm of Kriging Believer at one BO iteration to select a batch of recommended candidate points is summarized in Algorithm 2.

---

[1]Theoretically, this is not possible with CFGs. However, we can extend the notion of substitution by substituting a string representation of a Python (float) variable for the placeholder variables `Pt`, `Pm`, and `Pb`.

---

**Algorithm 2** Kriging Believer algorithm to select one batch of points.

---

**Input:** Observation data $\mathcal{D}_t$, batch size $b$
**Output:** The batch points $\mathcal{B}_{t+1} = \{\mathbf{x}_{t+1}^{(1)}, \ldots, \mathbf{x}_{t+1}^{(b)}\}$
$\tilde{\mathcal{D}}_t = \mathcal{D}_t \cup \tilde{\mathcal{D}}_p$
**for** $j = 1, \ldots, b$ **do**
 Select the next $\mathbf{x}_{t+1}^{(j)}$ by maximizing acquisition function $\alpha(\mathbf{x}|\tilde{\mathcal{D}}_t)$
 Compute the predictive posterior mean $\mu(\mathbf{x}_{t+1}^{(j)}|\tilde{\mathcal{D}}_t)$
 $\tilde{\mathcal{D}}_t \leftarrow \tilde{\mathcal{D}}_t \cup (\mathbf{x}_{t+1}, \mu(\mathbf{x}_{t+1}^{(j)}|\tilde{\mathcal{D}}_t))$
**end for**

---

---

**Algorithm 3** Weisfeiler-Lehman subtree kernel computation (Shervashidze et al., 2011).

---

**Input:** Graphs $G_1$, $G_2$, maximum iterations $H$
**Output:** Kernel function value between the graphs
Initialize the feature vectors $\phi(G_1) = \phi_0(G_1), \phi(G_2) = \phi_0(G_2)$ with the respective counts of original node labels (i.e., the $h = 0$ WL features)
**for** $h = 1, \ldots H$ **do**
 Assign a multiset $M_h(v) = \{l_{h-1}(u)|u \in \mathcal{N}(v)\}$ to each node $v \in G$, where $l_{h-1}$ is the node label function of the $h-1$-th WL iteration and $\mathcal{N}$ is the node neighbor function
 Sort elements in multiset $M_h(v)$ and concatenate them to string $s_h(v)$
 Compress each string $s_h(v)$ using the hash function $f$ s.t. $f(s_h(v)) = f(s_h(w)) \iff s_h(v) = s_h(u)$
 Add $l_{h-1}$ as prefix for $s_h(v)$
 Concatenate the WL features $\phi_h(G_1), \phi_h(G_2)$ with the respective counts of the *new* labels: $\phi(G_1) = [\phi(G_1), \phi_h(G_1)], \phi(G_2) = [\phi(G_2), \phi_h(G_2)]$
 Set $l_h(v) := f(s_h(v)) \, \forall v \in G$
**end for**
Compute inner product $k = \langle \phi_h(G_1), \phi_h(G_2) \rangle$ between WL features $\phi_h(G_1), \phi_h(G_2)$ in RKHS $\mathcal{H}$

---

### F.2 HIERARCHICAL WEISFEILER-LEHMAN KERNEL

Inspired by Ru et al. (2021), we adopted the Weisfeiler-Lehman (WL) graph kernel (Shervashidze et al., 2011) in the GP surrogate model to handle the graph nature of neural architectures. The basic idea of the WL kernel is to first compare node labels, and then iteratively aggregate labels of neighboring nodes, compress them into a new label and compare them. Algorithm 3 summarizes the WL kernel procedure.

Ru et al. (2021) identified three reasons for using the WL kernel: (1) it is able to compare labeled and directed graphs of different sizes, (2) it is expressive, and (3) it is relatively efficient and scalable. Our search space design can afford a diverse spectrum of neural architectures with very heterogeneous topological structure. Therefore, reason (1) is a very important property of the WL kernel to account for the diversity of neural architectures. Moreover, if we allow many hierarchical levels, we can construct very large neural architectures. Therefore, reasons (2) and (3) are essential for accurate and fast modeling. However, neural architectures in our search spaces may be significantly larger, which makes it difficult for a single WL kernel to capture the more global topological patterns. Moreover, modeling solely based on the final neural architecture ignores the useful macro-level information from earlier hierarchical levels. In our experiments (Section 5 and I), we have found stronger neural architectures by incorporating the hierarchical information in the kernel design, which provides experimental support for above arguments.

However, modeling solely based on the (standard) WL graph kernel neglects the useful hierarchical information from our assembly process. Moreover, the large size of neural architectures make it still challenging to capture the more global topological patterns. We therefore propose to use hierarchical information through a hierarchy of WL graph kernels that take into account the different granularities of the architectures and combine them in a weighted sum. To obtain the different granularities, we use the fold operators $F_l$ that removes algebraic terms beyond the $l$-th hierarchical level. Thereby,

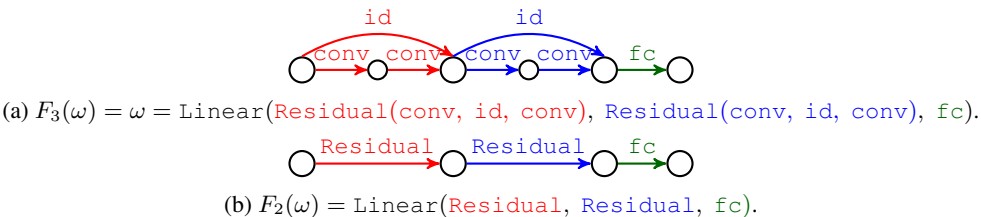

(a) $F_3(\omega) = \omega = $ `Linear(Residual(conv, id, conv), Residual(conv, id, conv), fc)`.

(b) $F_2(\omega) = $ `Linear(Residual, Residual, fc)`.

Figure 7: Labeled graphs $\Phi(F_2)$ and $\Phi(F_3)$ of the folds $F_2$ and $F_3$.

we obtain the folds

$$F_3(\omega) = \omega = \texttt{Linear(Residual(conv, id, conv), Residual(conv, id, conv), fc)}, \tag{18}$$

$$F_2(\omega) = \texttt{Linear(Residual, Residual, fc)} \quad , \quad F_1(\omega) = \texttt{Linear} \quad ,$$

for the algebraic architecture term $\omega$. Note that we ignore the first fold since it does not represent a labeled DAG. Figure 7 visualizes the labeled graphs $\Phi(F_2)$ and $\Phi(F_3)$ of the folds $F_2$ or $F_3$, respectively. These graphs can be fed into (standard) WL graph kernels. Therefore, we can construct a hierarchy of WL graph kernels $k_{WL}$ as follows:

$$k_{hWL}(\omega_i, \omega_j) = \sum_{l=2}^{L} \lambda_l \cdot k_{WL}(\Phi(F_l(\omega_i)), \Phi(F_l(\omega_j))) \quad , \tag{19}$$

where $\omega_i$ and $\omega_j$ are two algebraic architecture terms. Note that $\lambda_l$ govern the importance of the learned graph information across the hierarchical levels and can be optimized through the marginal likelihood.

## F.3 EXAMPLES FOR THE EVOLUTIONARY OPERATIONS

For the evolutionary operations, we adopted ideas from grammar-based genetic programming (McKay et al., 2010; Moss et al., 2020). In the following, we will show how these evolutionary operations manipulate algebraic terms, e.g.,

$$\texttt{Linear(Residual(conv, id, conv), Residual(conv, id, conv), fc)} \quad , \tag{20}$$

from the search space

$$\texttt{S ::= Linear(S, S, S) | Residual(S, S, S) | conv | id | fc} \quad , \tag{21}$$

to generate evolved algebraic terms. Figure 1 shows how we can derive the algebraic term in Equation 20 from the search space in Equation 21. For mutation operations, we first randomly pick a subterm of the algebraic term, e.g., `Residual(conv, id, conv)`. Then, we randomly sample a new subterm with the same nonterminal symbol S as start symbol, e.g., `Linear(conv, id, fc)`, and replace the previous subterm, yielding

$$\texttt{Linear(Linear(conv, id, fc), Residual(conv, id, conv), fc)} \quad . \tag{22}$$

For (self-)crossover operations, we swap two subterms, e.g., `Residual(conv, id, conv)` and `Residual(conv, id, conv)` with the same nonterminal S as start symbol, yielding

$$\texttt{Linear(Residual(conv, id, conv), Residual(conv, id, conv), fc)} \quad . \tag{23}$$

Note that unlike the commonly used crossover operation, which uses two parents, self-crossover has only one parent. In future work, we could also add a *self-copy* operation that copies a subterm to another part of the algebraic term, explicitly regularizing diversity and thus potentially speeding up the search.

## G RELATED WORK BEYOND NEURAL ARCHITECTURE SEARCH

While our work focuses exclusively on NAS, we will discuss below how it relates to the areas of optimizer search (as well as from scratch automated machine learning) and neural-symbolic programming.

Optimizer search is a closely related field to NAS, where we automatically search for an optimizer (i.e., an update function for the weights) instead of an architecture. Initial works used learnable parametric or non-parametric optimizers. While the former approaches (Andrychowicz et al., 2016; Li & Malik, 2017; Chen et al., 2017; 2022a) have poor scalability and generality, the latter works overcome those limitations. Bello et al. (2017) searched for an instantiation of hand-crafted patterns via reinforcement learning, while Wang et al. (2022) proposed a tree-structured search space[2] and searched for optimizers via a modified Monte Carlo sampling approach. AutoML-Zero (Real et al., 2020) took an even more general approach by searching over entire machine learning algorithms, including optimizers, from a generic search space built from basic mathematical operations with an evolutionary algorithm. Chen et al. (2022b) used RE to discover optimizers from a generic search space (inspired by AutoML-Zero) for training vision transformers (Dosovitskiy et al., 2021).

Complementary to the above, there is recent interest in automatically synthesizing programs from domain-specific languages. Gaunt et al. (2017) proposed a hand-crafted program template and simultaneously optimized the parameters of the differentiable program with gradient descent. The `HOUDINI` framework (Valkov et al., 2018) proposed type-directed (top-down) enumeration and evolution approaches over differentiable functional programs. Shah et al. (2020) hierarchically assembled differentiable programs and used neural networks for the approximation of missing expression in partial programs. Cui & Zhu (2021) treated CFGs stochastically with trainable production rule sampling weights, which were optimized with a gradient-based approach (Liu et al., 2019b). However, naïvely applying gradient-based approaches does not work in our search spaces due to the exponential explosion of supernet weights, but still renders an interesting direction for future work.

Compared to these lines of work, we extended CFGs to handle changes in spatial resolution, promote regularity, and (compared to most of them) incorporate constraints, the latter two of which could also be applied in those domains. We also proposed a BO search strategy to search efficiently with a tailored kernel design to handle the hierarchical nature of the search space (i.e., the architectures).

## H  IMPLEMENTATION DETAILS OF THE SEARCH STRATEGIES

**BANAT & BANAT (WL)**   The only difference between BANAT and BANAT (WL) is that the former uses our proposed hierarchy of WL kernels (hWL), whereas the latter only uses a single WL kernel (WL) for the entire architecture (c.f., (Ru et al., 2021)). We ran BANAT asynchronously in parallel throughout our experiments with a batch size of $B = 1$, i.e., at each BO iteration a single architecture is proposed for evaluation. For the acquisition function optimization, we used a pool size of $P = 200$, where the initial population consisted of the current ten best-performing architectures and the remainder were randomly sampled architectures to encourage exploration in the huge search spaces. During evolution, the mutation probability was set to $p_{mut} = 0.5$ and crossover probability was set to $p_{cross} = 0.5$. From the crossovers, half of them were self-crossovers of one parent and the other half were common crossovers between two parents. The tournament selection probability was set to $p_{tour} = 0.2$. We evolved the population at least for ten iterations and a maximum of 50 iterations using a early stopping criterion based on the fitness value improvements over the last five iterations.

**Regularized Evolution (RE)**   RE (Real et al., 2019; Liu et al., 2018) iteratively mutates the best architectures out of a sample of the population. We reduced the population size from 50 to 30 to account for fewer evaluations, and used a sample size of 10. We also ran RE asynchronously for better comparability.

## I  SEARCHING THE HIERARCHICAL NAS-BENCH-201 SEARCH SPACE

In this section, we provide training details (Section I.1) and provide complementary results as well as conduct extensive analyses (Section I.2).

---

[2]Note that the tree-structured search space can equivalently be described with a CFG (with a constraint on the number of maximum depth of the syntax trees).

Table 4: Licenses for the datasets we used in our experiments.

| Dataset | License | URL |
|---|---|---|
| CIFAR-10 (Krizhevsky et al., 2009) | MIT | https://www.cs.toronto.edu/~kriz/cifar.html |
| CIFAR-100 (Krizhevsky et al., 2009) | MIT | https://www.cs.toronto.edu/~kriz/cifar.html |
| ImageNet-16-120 (Chrabaszcz et al., 2017) | MIT | https://patrykchrabaszcz.github.io/Imagenet32/ |
| CIFARTile (Geada et al., 2021) | GNU | https://github.com/RobGeada/cvpr-nas-datasets |
| AddNIST (Geada et al., 2021) | GNU | https://github.com/RobGeada/cvpr-nas-datasets |

## I.1 TRAINING DETAILS

**Training protocol**  We evaluated all search strategies on CIFAR-10/100 (Krizhevsky et al., 2009), ImageNet-16-120 (Chrabaszcz et al., 2017), CIFARTile, and AddNIST (Geada et al., 2021). Note that CIFARTile and AddNIST are novel datasets and therefore have not yet been optimized by the research community. We provide further dataset details below. For training of architectures on CIFAR-10/100 and ImageNet-16-120, we followed Dong & Yang (2020). We trained architectures with SGD with learning rate of $0.1$, Nesterov momentum of $0.9$, weight decay of $0.0005$ with cosine annealing (Loshchilov & Hutter, 2019), and batch size of 256 for 200 epochs. The initial channels were set to 16. For both CIFAR-10 and CIFAR-100, we used random flip with probability $0.5$ followed by a random crop (32x32 with 4 pixel padding) and normalization. For ImageNet-16-120, we used a 16x16 random crop with 2 pixel padding instead. For training of architectures on AddNIST and CIFARTile, we followed the training protocol from the CVPR-NAS 2021 competition (Geada et al., 2021): We trained architectures with SGD with learning rate of $0.01$, momentum of $0.9$, and weight decay of $0.0003$ with cosine annealing, and batch size of 64 for 64 epochs. We set the initial channels to 16 and did not apply any further data augmentation.

**Dataset details**  In Table 4, we provide the licenses for the datasets used in our experiments. For training of architectures on CIFAR-10, CIFAR-100 (Krizhevsky et al., 2009), and ImageNet-16-120 (Chrabaszcz et al., 2017), we followed the dataset splits and training protocol of NAS-Bench-201 (Dong & Yang, 2020). For CIFAR-10, we split the original training set into a new training set with 25k images and validation set with 25k images following Dong & Yang (2020). The test set remained unchanged. For evaluation, we trained architectures on both the training and validation set. For CIFAR-100, the training set remained unchanged, but the test set was partitioned in a validation set and new test set with each 5K images. For ImageNet-16-120, all splits remained unchanged. For AddNIST and CIFARTile, we used the training, validation, and test splits as defined in the CVPR-NAS 2021 competition (Geada et al., 2021).

## I.2 EXTENDED SEARCH RESULTS AND ANALYSES

Supplementary to Figure 2, Figure 8 compares the cell-based vs. hierarchical NAS-Bench-201 search space from Section 6.1 using RS, RE, and BANAT (WL). The cell-based search space design shows on par or stronger performance on all datasets except for CIFARTile for the three search strategies. In contrast, for our proposed search strategy BANAT we find on par (CIFAR-10/100) or superior (ImageNet-16-120, CIFARTile, and AddNIST) performance using the hierarchical search space design. This clearly shows that the increase of the search space does not necessarily yields the discovery of stronger neural architectures. Further, it exemplifies the importance of a strong search strategy to search effectively and efficiently in huge hierarchical search spaces (**Q2**), and provides further evidence that the incorporation of hierarchical information is a key contributor for search efficiency (**Q3**). Based on this, we believe that future work using, e.g., graph neural networks as a surrogate, may benefit from the incorporation of hierarchical information.

We report the test errors of our best found architectures in Table 5. We observe that our search strategy BANAT finds the strongest performing architectures across all dataset (**Q2**, **Q3**). Also note that we achieve better (validation and) test performance on ImageNet-16-120 on the hierarchical than the state-of-the-art search strategy on the cell-based NAS-Bench-201 search space (i.e., $+0.37\%p$ compared to Shapley-NAS (Xiao et al., 2022)) (**Q1**).

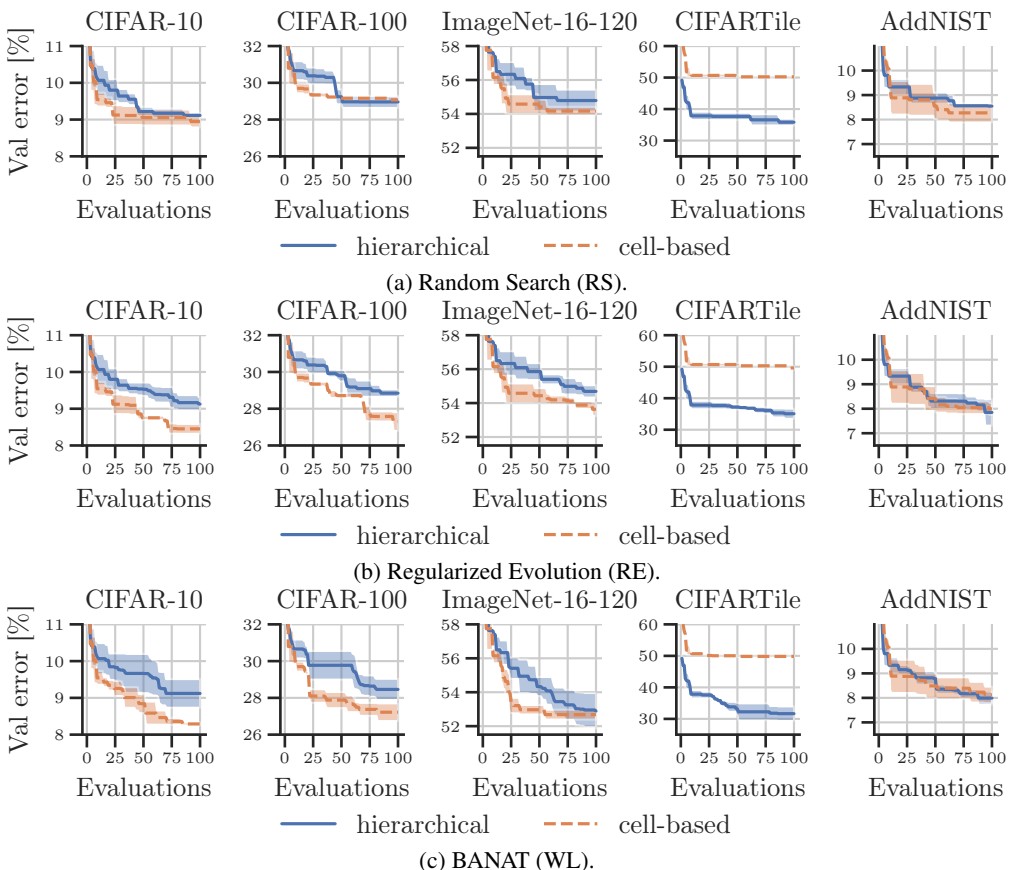

Figure 8: Cell-based vs. hierarchical search spaces. We plot mean and $\pm 1$ standard error of the validation error on the cell-based (dashed orange) and hierarchical (solid blue) NAS-Bench-201 search space using Random Search (RS) (top), Regularized Evolution (RE) (middle), and BANAT (WL) (bottom).

Table 5: Test errors (and ±1 standard error) of popular baseline architectures (e.g., ResNet (He et al., 2016) and EfficientNet (Tan & Le, 2019) variants), and our best found architectures on the cell-based and hierarchical NAS-Bench-201 search space. Note that we picked the ResNet and EfficientNet variant based on the test error, consequently giving an overestimate of their test performance.
† optimal numbers as reported in Dong & Yang (2020).
(best) test error (and ±1 standard error) across three seeds {777, 888, 999} of the best architecture of the three search runs with lowest validation error.

| Method | CIFAR-10 | | CIFAR-100 | | ImageNet-16-120 | | CIFARTile | | AddNIST | |
| --- | --- | --- | --- | --- | --- | --- | --- | --- | --- | --- |
| | cell-based | hierarchical | cell-based | hierarchical | cell-based | hierarchical | cell-based | hierarchical | cell-based | hierarchical |
| Best ResNet (He et al., 2016) | 6.49 ± 0.24 (32) | | 27.1 ± 0.67 (110) | | 53.67 ± 0.18 (56) | | 57.8 ± 0.57 (18) | | 7.78 ± 0.05 (34) | |
| Best EfficientNet (Tan & Le, 2019) | 11.73 ± 0.1 (B0) | | 35.17 ± 0.42 (B6) | | 77.73 ± 0.29 (B0) | | 61.01 ± 0.62 (B0) | | 13.24 ± 0.58 (B1) | |
| NAS-Bench-201 oracle† | 5.63 | | 26.49 | | 52.69 | | - | | - | |
| RS | 6.39 ± 0.18 | 6.77 ± 0.1 | 28.75 ± 0.57 | 29.49 ± 0.57 | 54.83 ± 0.78 | 54.7 ± 0.82 | 52.72 ± 0.45 | 40.93 ± 0.81 | 7.82 ± 0.36 | 8.05 ± 0.29 |
| NASWOT (N=10) (Mellor et al., 2021) | 6.55 ± 0.1 | 8.18 ± 0.46 | 29.35 ± 0.53 | 31.73 ± 0.96 | 56.8 ± 1.35 | 58.66 ± 0.29 | 41.83 ± 2.29 | 49.46 ± 2.95 | 10.11 ± 0.69 | 11.81 ± 1.55 |
| NASWOT (N=100) (Mellor et al., 2021) | 6.59 ± 0.17 | 8.56 ± 0.87 | 28.91 ± 0.25 | 31.65 ± 1.95 | 55.99 ± 1.3 | 58.47 ± 2.74 | 41.63 ± 1.02 | 43.31 ± 2.0 | 10.75 ± 0.23 | 14.47 ± 1.44 |
| NASWOT (N=1000) (Mellor et al., 2021) | 6.68 ± 0.12 | 8.26 ± 0.38 | 29.37 ± 0.17 | 31.66 ± 0.72 | 58.93 ± 2.92 | 58.33 ± 0.91 | 39.61 ± 1.12 | 45.66 ± 1.29 | 10.68 ± 0.27 | 13.57 ± 1.89 |
| NASWOT (N=10000) (Mellor et al., 2021) | 6.98 ± 0.43 | 8.4 ± 0.52 | 29.95 ± 0.42 | 32.09 ± 1.61 | 54.2 ± 0.49 | 57.58 ± 1.53 | 39.9 ± 1.2 | 42.45 ± 0.67 | 10.72 ± 0.53 | 14.82 ± 0.66 |
| RE (Real et al., 2019; Liu et al., 2018) | **5.76 ± 0.17** | 6.88 ± 0.24 | 27.68 ± 0.55 | 30.0 ± 0.32 | 53.92 ± 0.6 | 55.39 ± 0.54 | 52.79 ± 0.59 | 40.99 ± 2.89 | **7.69 ± 0.35** | 7.56 ± 0.69 |
| BANAT (WL) (Ru et al., 2021) | **5.68 ± 0.11** | 6.98 ± 0.5 | **27.66 ± 0.18** | 28.7 ± 0.64 | **53.67 ± 0.39** | 53.47 ± 0.86 | 52.81 ± 0.27 | 35.75 ± 1.58 | 7.86 ± 0.41 | 8.2 ± 0.37 |
| BANAT | **5.68 ± 0.11** | **6.0 ± 0.16** | **27.66 ± 0.18** | **27.57 ± 0.46** | **53.67 ± 0.39** | **53.43 ± 0.61** | 52.81 ± 0.27 | **32.28 ± 2.39** | 7.86 ± 0.41 | **6.09 ± 0.34** |
| BANAT (best) | 5.64 ± 0.14 | 5.65 ± 0.09 | 27.03 ± 0.23 | 27.63 ± 0.2 | 53.54 ± 0.43 | 52.78 ± 0.23 | 53.18 ± 0.91 | 30.33 ± 0.77 | 8.04 ± 0.45 | 6.33 ± 0.59 |

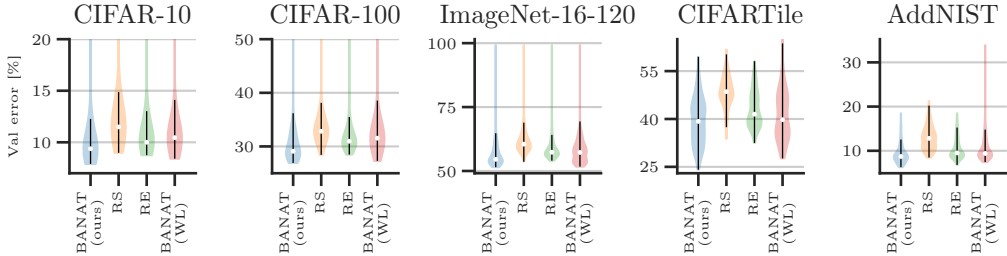

Figure 9: Density estimates for the validation error of all architecture candidates proposed by the search strategies (i.e., BANAT, RS, RE, and BANAT (WL)) and across datasets (i.e., CIFAR-10/100, ImageNet-16-120, CIFARTile, AddNIST) from our experiments in Section 5.

**Search costs** Search time varied across datasets from ca. 0.5 days (CIFAR-10) to ca. 1.8 days (ImageNet-16-120) using eight asynchronous workers, each with an NVIDIA RTX 2080 Ti GPU, for ca. 4 to ca. 14.4 GPU days in total.

**Is our search strategy BANAT exploring well-performing architectures during search?** To investigate the question, we studied density estimates of the validation error of proposed candidates for all search strategies across our experiments from Section 5. This provides a better view for whether search strategies are exploring well-performing architectures or wasting computational resources on low-performing architectures. Figure 9 shows that our proposed search strategy BANAT explored better architecture candidates across all the datasets, i.e., it has smaller median validation errors and the distributions are further shifted towards smaller validation errors than for the other search strategies.

**What distinguishes top-performing neural architectures from the other ones?** To understand what distinguishes top-performing neural architecture from other ones, we analyze the impact of maximum depth on performance and the frequency of production rules in the worst-10%, top-10%, or other neural architectures, respectively. In another analysis, we marginalize out the validation error of every production rule; thereby relating the contribution of a production rule with the performance of the architecture. Note, however, that both analyses ignore the topological information, i.e., a topological operator or primitive computation may have a different effect at different stages of the architecture.

Figure 10 shows no particular trend (e.g., more depth yields better performance) across the datasets, indicating that depth may not be the most important factor for performance in our hierarchical search space. In contrast, Figure 11 and Figure 12 show that particularly macro-level production rules (i.e., for the nonterminals D2, D1, D0, and D) have a large effect on the performance of an architecture. Interestingly, we find that that top-performing architectures (almost exclusively) use the topological operator `Residual3` for derivations from the nonterminal D1 across search spaces. This hints that a residual connection at the macro-level could be a strong topological structure, but remains to be evaluated for a variety of architectures. Cell-level production choices have less effect on performance. However, we hypothesize that this may also be due to the neglect of topological information. We leave further analysis for future work.

**What is the impact of flexible parameterization of convolutional blocks?** To investigate the impact of the flexible parameterization of the convolutional blocks (i.e., activation functions, normalizations, and type of convolution), we removed the flexible parameterization and allowed only the same primitive computations as in the cell-based NAS-Bench-201 search space, while still searching over the macro architecture. More explicitly, we only allow ReLU non-linearity as the activation function, batch normalization as the normalization, and $1 \times 1$ or $3 \times 3$ convolutions. Figure 13 shows that for all datasets except CIFAR-100, flexible parameterization of the convolutional blocks improves performance of the found architectures. Interestingly, we find an architecture on CIFAR-100, which achieves $26.24\%$ test error with $1.307\,\mathrm{MB}$ and $167.172\,\mathrm{M}$ number of parameters or FLOPs, respectively. This architecture is superior to the optimal architecture in the cell-based NAS-Bench-201 search space. Note that this architecture is also pareto-optimal for test error vs. number of parameters and test error vs. number of FLOPs.

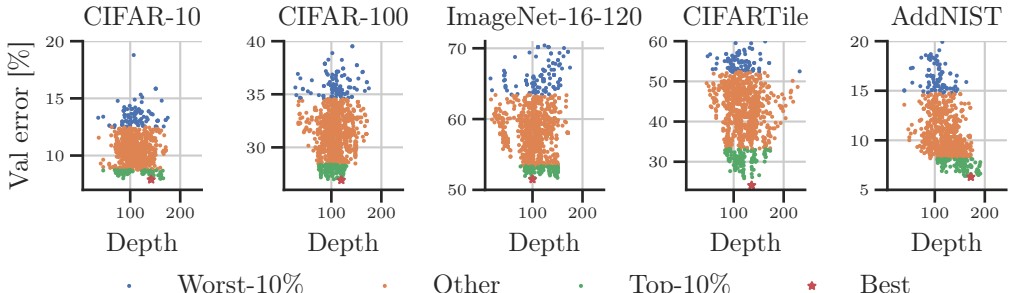

Figure 10: Validation error over maximal depth of all architecture candidates proposed by the search strategies (i.e., BANAT, RS, RE, and BANAT (WL)) and across datasets (i.e., CIFAR-10/100, ImageNet-16-120, CIFARTile, AddNIST) from our experiments in Section 5.

**Test error vs. number of parameters and FLOPs** Figure 14 shows the test error vs. the number of parameters or FLOPs. Our best found architectures fall well within the parameter and FLOPs ranges of the cell-based NAS-Bench-201 search space across all datasets, except for the parameters on CIFAR-10. Note that our best found architecture on ImageNet-16-120 is pareto-optimal for test error vs. number of parameters and test error vs. number of FLOPs.

**Best architectures** Below we report the best found architecture per dataset on the hierarchical NAS-Bench-201 search space (Section 5.1) for each dataset. Figure 15 visualizes the novel and diverse design of the architectures (including stem and classifier head).

CIFAR-10 (mean test error $5.65\,\%$, #params $2.204\,\mathrm{MB}$, FLOPs $127.673\,\mathrm{M}$):

> Linear4(Residual3(Residual2(Cell(id, zero, Linear1(Linear3(hardswish, conv1x1, layer)), Linear1(Linear3(hardswish, conv3x3, layer)), zero, Linear1(Linear3(mish, conv3x3, instance))), Cell(Linear1(Linear3(relu, dconv3x3, layer)), id, avg_pool, Linear1(Linear3(relu, dconv3x3, layer)), id, zero), Cell(zero, Linear1(Linear3(relu, conv1x1, layer)), id, Linear1(Linear3(hardswish, conv1x1, instance)), Linear1(Linear3(hardswish, conv3x3, layer)), Linear1(Linear3(hardswish, dconv3x3, layer)))), Residual2(Cell(id, zero, Linear1(Linear3(relu, conv1x1, layer)), Linear1(Linear3(mish, conv1x1, layer)), Linear1(Linear3(hardswish, conv3x3, layer)), zero), Cell(id, zero, id, Linear1(Linear3(relu, conv3x3, batch)), id, id), Cell(Linear1(Linear3(hardswish, conv3x3, layer)), Linear1(Linear3(hardswish, conv1x1, layer)), Linear1(Linear3(relu, conv1x1, layer)), Linear1(Linear3(relu, conv3x3, layer)), zero, id)), Residual2(Cell(Linear1(Linear3(hardswish, conv1x1, instance)), Linear1(Linear3(hardswish, dconv3x3, batch)), Linear1(Linear3(mish, dconv3x3, instance)), Linear1(Linear3(relu, conv1x1, batch)), id, id), down, down), Residual2(Cell(Linear1(Linear3(hardswish, conv1x1, layer)), Linear1(Linear3(hardswish, dconv3x3, batch)), Linear1(Linear3(relu, conv1x1, batch)), Linear1(Linear3(hardswish, conv3x3, layer)), id, avg_pool), down, down)), Residual3(Residual2(Cell(id, zero, Linear1(Linear3(hardswish, conv1x1, layer)), Linear1(Linear3(hardswish, conv3x3, layer)), id, Linear1(Linear3(mish, conv3x3, instance))), Cell(Linear1(Linear3(relu, dconv3x3, layer)), id, avg_pool, Linear1(Linear3(relu, dconv3x3, layer)), id, zero), Cell(zero, Linear1(Linear3(relu, conv1x1, layer)), id, Linear1(Linear3(hardswish, conv1x1, instance)), Linear1(Linear3(hardswish, conv1x1, layer)), Linear1(Linear3(hardswish, dconv3x3, layer)))), Residual2(Cell(id, zero, Linear1(Linear3(mish, conv1x1, layer)), Linear1(Linear3(mish, conv3x3, layer)), Linear1(Linear3(hardswish, dconv3x3, batch)), zero), Cell(id, zero, id, Linear1(Linear3(relu, conv3x3, batch)), id, id), Cell(Linear1(Linear3(hardswish, conv3x3, layer)), Linear1(Linear3(hardswish, conv1x1, layer)), Linear1(Linear3(mish, conv1x1, batch)), Linear1(Linear3(relu, conv3x3, instance)), zero, id)), Residual2(Cell(Linear1(Linear3(relu, conv1x1, batch)), Linear1(Linear3(hardswish, dconv3x3, batch)), id, Linear1(Linear3(relu, conv1x1, batch)), id, id), down, down), Residual2(Cell(Linear1(Linear3(hardswish, conv1x1, layer)), Linear1(Linear3(hardswish, dconv3x3, batch)), Linear1(Linear3(relu, conv1x1, batch)), Linear1(Linear3(hardswish, conv3x3, layer)), id, avg_pool), down, down)), Linear3(Residual2(Cell(Linear1(Linear3(hardswish, conv3x3, batch)), Linear1(Linear3(relu, conv1x1, instance)), Linear1(Linear3(relu, conv1x1, layer)), Linear1(Linear3(relu, conv1x1, layer)), Linear1(Linear3(relu, conv1x1, layer)), id), Cell(Linear1(Linear3(relu, conv1x1, batch)), id, id, Linear1(Linear3(relu,

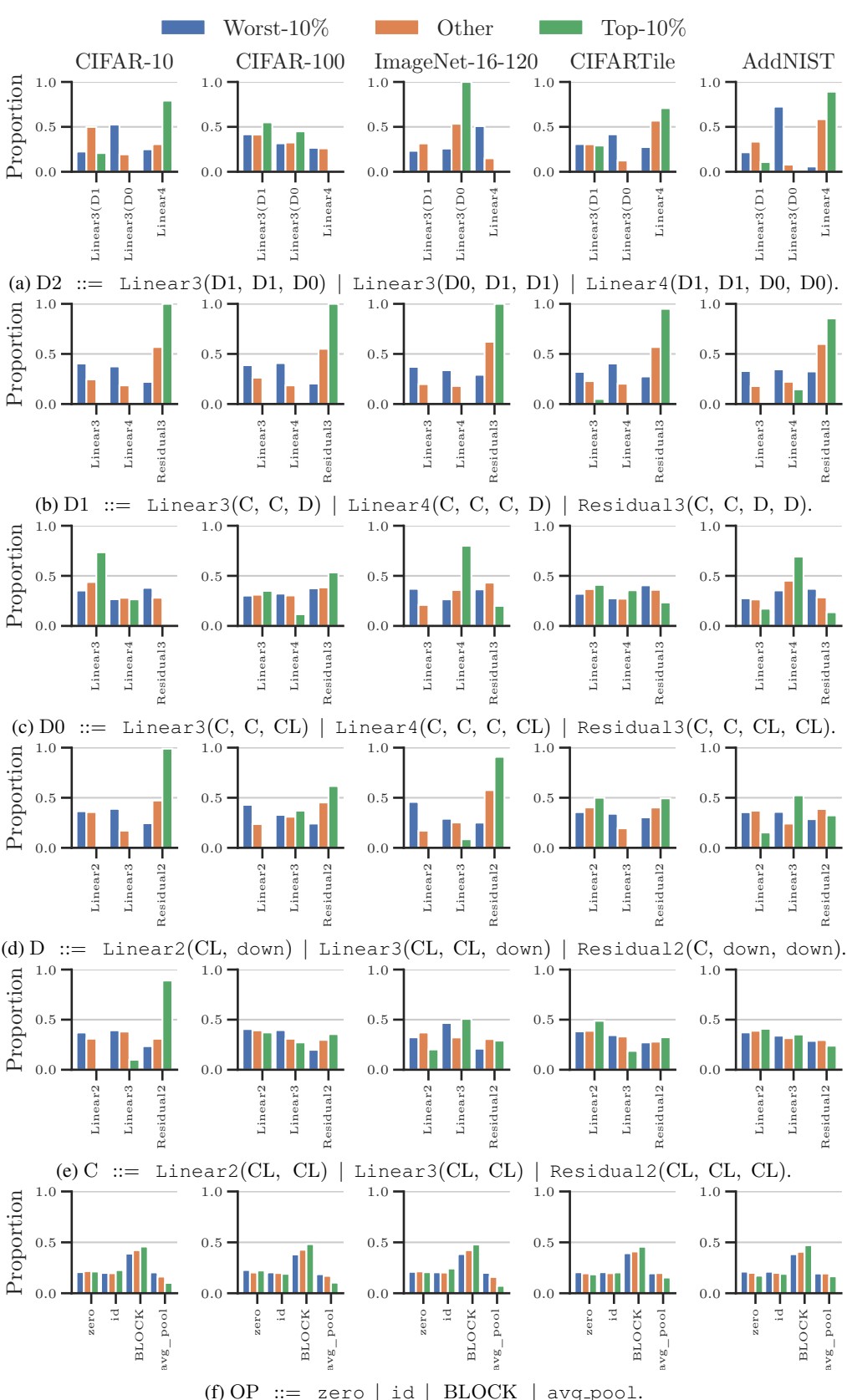

(a) D2 ::= Linear3(D1, D1, D0) | Linear3(D0, D1, D1) | Linear4(D1, D1, D0, D0).

(b) D1 ::= Linear3(C, C, D) | Linear4(C, C, C, D) | Residual3(C, C, D, D).

(c) D0 ::= Linear3(C, C, CL) | Linear4(C, C, C, CL) | Residual3(C, C, CL, CL).

(d) D ::= Linear2(CL, down) | Linear3(CL, CL, down) | Residual2(C, down, down).

(e) C ::= Linear2(CL, CL) | Linear3(CL, CL) | Residual2(CL, CL, CL).

(f) OP ::= zero | id | BLOCK | avg_pool.

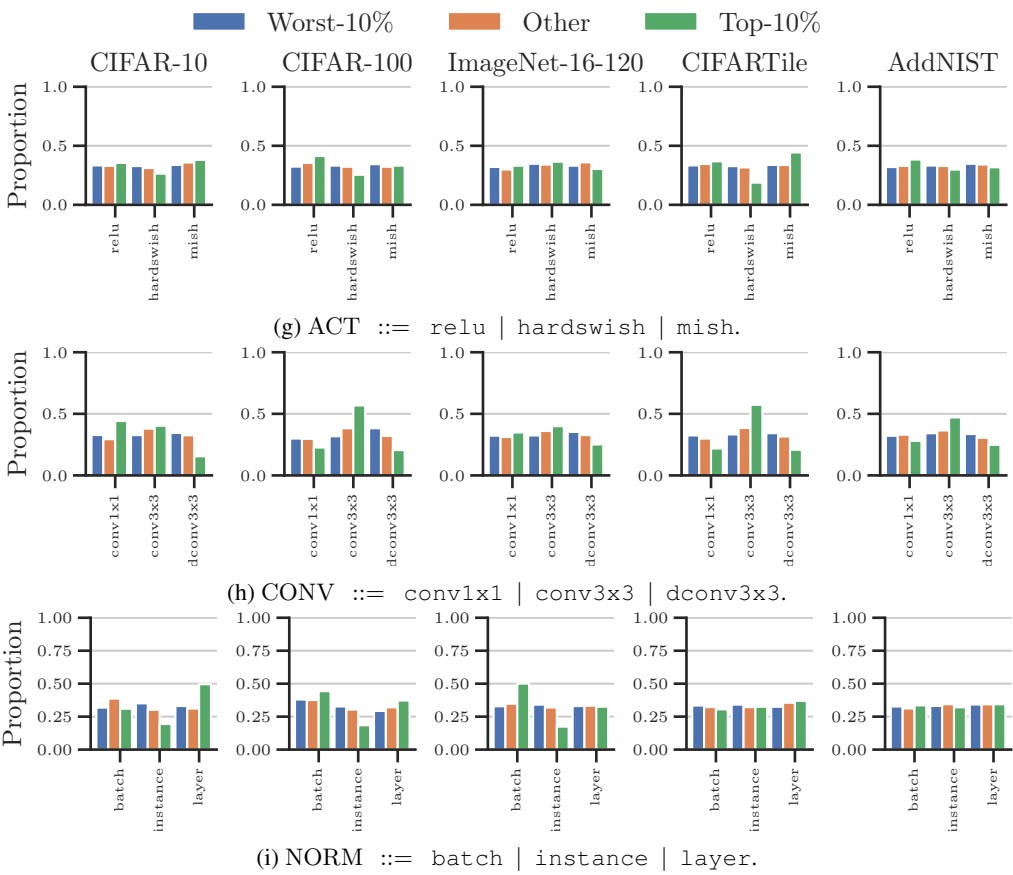

Figure 11: Comparison of the proportion of production rules in the worst-10% (blue), top-10% (green) and other (orange) neural architectures from our experiments in Section 5.

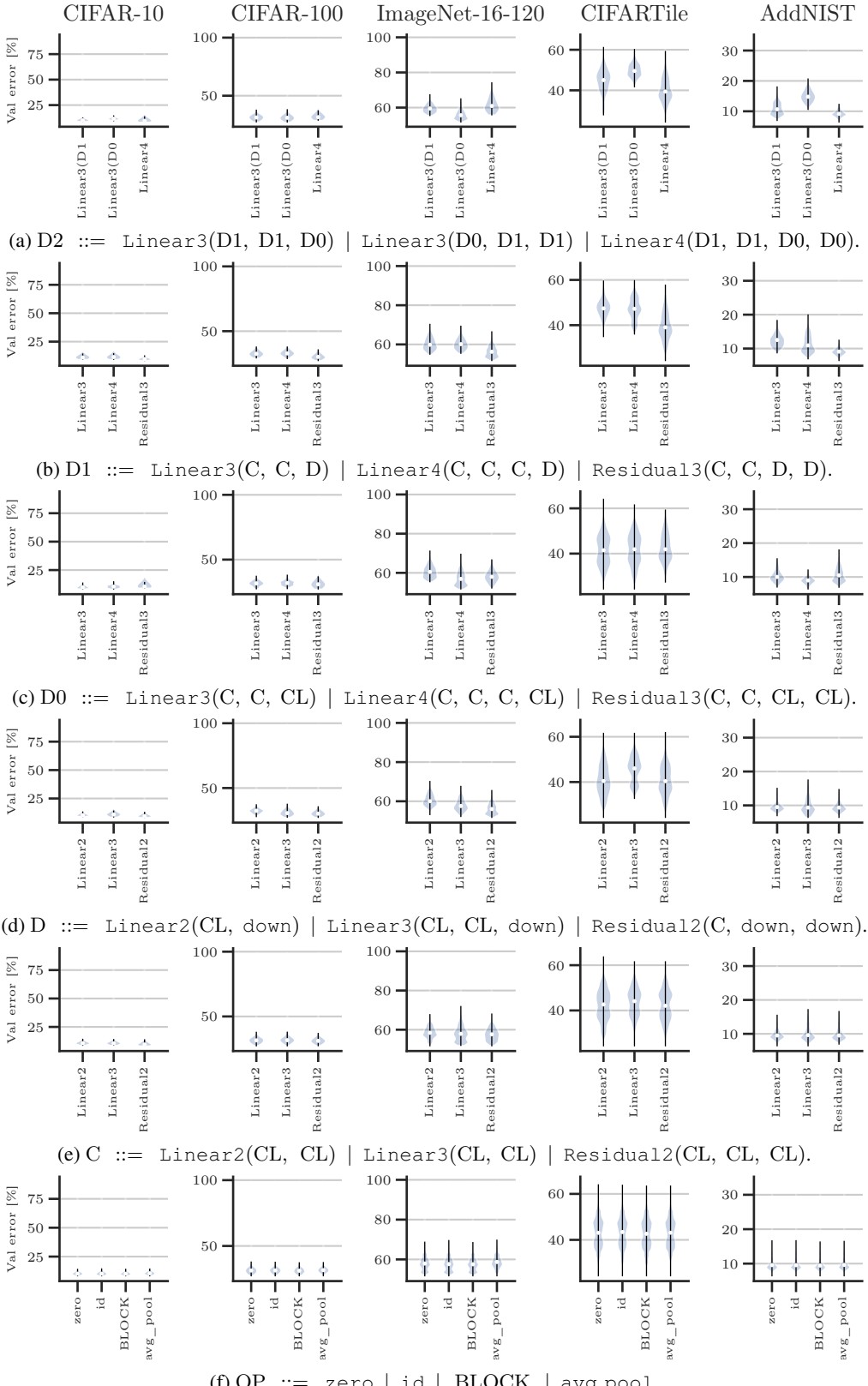

(a) D2 ::= Linear3(D1, D1, D0) | Linear3(D0, D1, D1) | Linear4(D1, D1, D0, D0).

(b) D1 ::= Linear3(C, C, D) | Linear4(C, C, C, D) | Residual3(C, C, D, D).

(c) D0 ::= Linear3(C, C, CL) | Linear4(C, C, C, CL) | Residual3(C, C, CL, CL).

(d) D ::= Linear2(CL, down) | Linear3(CL, CL, down) | Residual2(C, down, down).

(e) C ::= Linear2(CL, CL) | Linear3(CL, CL) | Residual2(CL, CL, CL).

(f) OP ::= zero | id | BLOCK | avg_pool.

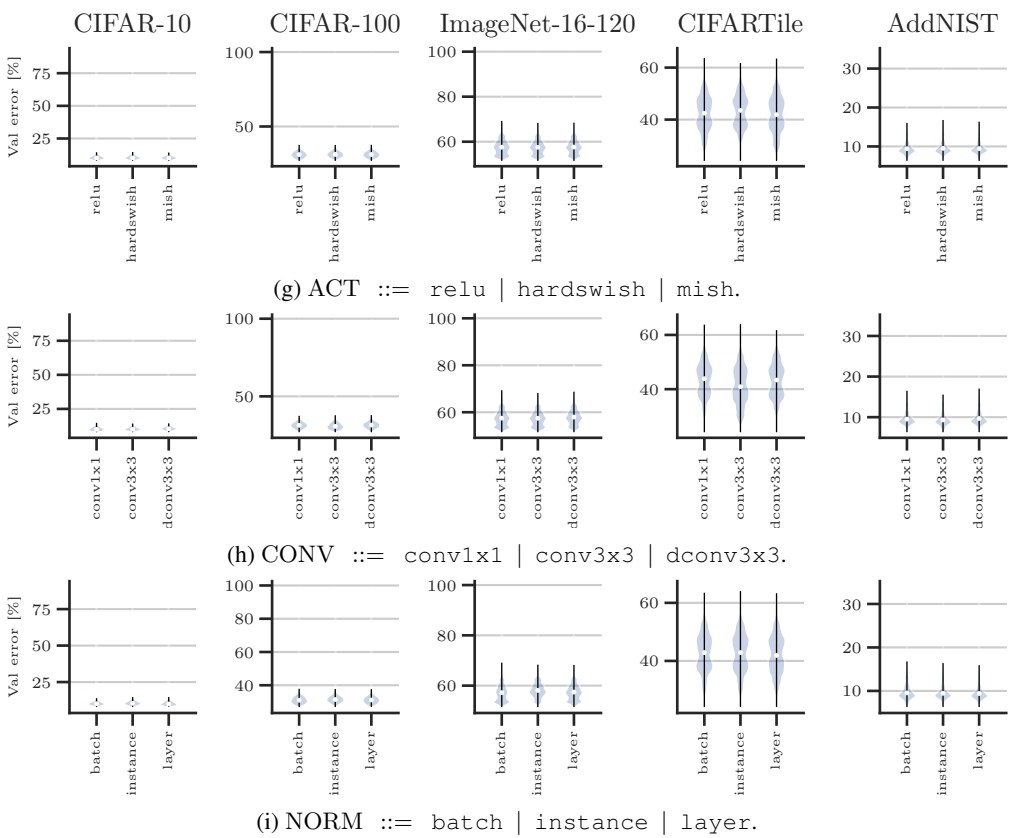

Figure 12: Marginalized performance of every production rule in our hierarchical NAS-Bench-201 search space from Section 5.

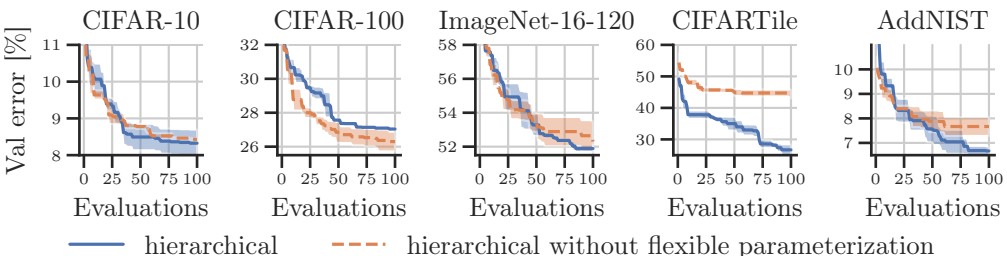

Figure 13: Impact of flexible parameterization of convolutional blocks in the hierarchical NAS-Bench-201 search space.

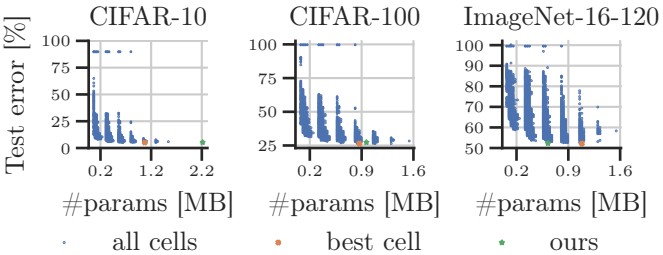

(a) Test error vs. number of parameters (#params (MB)).

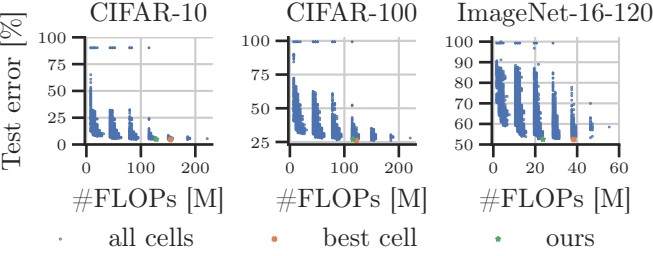

(b) Test error vs. number of FLOPs (#FLOPs (M)).

Figure 14: Test error vs. number of parameters (a) and FLOPs (b) for each architecture candidate in the cell-based search space (blue dots), the best cell (orange cross), and our best found architecture (green star).

conv1x1, layer)), id, id), Cell(id, Linear1(Linear3(relu, conv1x1, instance)), Linear1(Linear3(relu, conv1x1, instance)), Linear1(Linear3(relu, conv1x1, layer)), zero, Linear1(Linear3(hardswish, conv3x3, batch)))), Residual2(Cell(Linear1(Linear3(hardswish, conv3x3, layer)), Linear1(Linear3(relu, conv3x3, instance)), Linear1(Linear3(mish, conv1x1, layer)), Linear1(Linear3(relu, conv1x1, layer)), Linear1(Linear3(relu, conv3x3, layer)), id), Cell(Linear1(Linear3(relu, conv1x1, batch)), id, id, Linear1(Linear3(relu, conv3x3, batch)), id, id), Cell(id, Linear1(Linear3(relu, conv1x1, instance)), Linear1(Linear3(relu, conv1x1, instance)), Linear1(Linear3(relu, dconv3x3, layer)), zero, Linear1(Linear3(hardswish, dconv3x3, layer)))), Cell(Linear1(Linear3(relu, dconv3x3, instance)), zero, zero, id, zero, id)), Linear4(Residual2(Cell(Linear1(Linear3(hardswish, conv3x3, layer)), Linear1(Linear3(relu, conv3x3, layer)), Linear1(Linear3(relu, conv1x1, layer)), Linear1(Linear3(relu, conv1x1, layer)), Linear1(Linear3(relu, conv3x3, layer)), id), Cell(Linear1(Linear3(relu, conv1x1, batch)), id, id, Linear1(Linear3(relu, conv3x3, batch)), id, id), Cell(id, Linear1(Linear3(relu, conv1x1, instance)), Linear1(Linear3(relu, conv1x1, instance)), Linear1(Linear3(relu, conv1x1, layer)), zero, Linear1(Linear3(hardswish, conv3x3, batch)))), Residual2(Cell(Linear1(Linear3(hardswish, conv3x3, layer)), Linear1(Linear3(relu, conv3x3, layer)), Linear1(Linear3(relu, conv1x1, layer)), Linear1(Linear3(relu, conv1x1, layer)), Linear1(Linear3(relu, conv3x3, layer)), id), Cell(Linear1(Linear3(relu, conv1x1, batch)), id, id, Linear1(Linear3(relu, conv3x3, batch)), id, id), Cell(id, Linear1(Linear3(relu, conv1x1, instance)), Linear1(Linear3(relu, conv1x1, instance)), Linear1(Linear3(relu, conv1x1, layer)), zero, Linear1(Linear3(hardswish, conv3x3, layer)))), Residual2(Cell(Linear1(Linear3(hardswish, conv3x3, layer)), Linear1(Linear3(relu, conv3x3, layer)), Linear1(Linear3(relu, conv1x1, layer)), Linear1(Linear3(relu, conv1x1, layer)), Linear1(Linear3(relu, conv3x3, layer)), id), Cell(Linear1(Linear3(relu, conv1x1, batch)), id, id, Linear1(Linear3(relu, conv3x3, batch)), id, id), Cell(id, Linear1(Linear3(relu, conv1x1, instance)), Linear1(Linear3(relu, conv1x1, instance)), Linear1(Linear3(relu, conv1x1, layer)), zero, Linear1(Linear3(hardswish, conv3x3, layer)))), Cell(id, Linear1(Linear3(hardswish, conv1x1, layer)), Linear1(Linear3(mish, conv1x1, batch)), id, zero, id))) .

CIFAR-100 (mean test error 27.63 %, #params 0.962 MB, FLOPs 115.243 M):

Linear3(Residual3(Linear3(Cell(Linear3(mish, conv3x3, layer), avg_pool, Linear3(hardswish, conv1x1, instance), zero, Linear3(mish, conv3x3, batch), zero), Cell(Linear3(hardswish, dconv3x3, batch), zero, Linear3(hardswish, dconv3x3, batch), Linear3(relu, dconv3x3, batch), id, id), Cell(Linear3(mish, conv3x3, batch), zero, id, zero, Linear3(hardswish, dconv3x3, batch), id)), Linear2(Cell(id, zero, Linear3(mish, conv3x3, batch), zero, zero, Linear3(mish, conv1x1, batch)), Cell(zero, zero, zero, id, zero, avg_pool)), Cell(Linear3(relu, conv3x3, batch), zero, Linear3(hardswish, conv3x3, instance), id, id, avg_pool), Cell(id, id, zero, zero, id, id)), Residual3(Linear3(Cell(Linear3(mish, conv3x3, layer), id, Linear3(hardswish, dconv3x3, layer), Linear3(hardswish, dconv3x3, batch), Linear3(mish, conv3x3, instance), Linear3(mish, conv3x3, batch)), Cell(Linear3(hardswish, conv1x1, layer), id, Linear3(hardswish, dconv3x3, batch), Linear3(relu, conv3x3, layer), id, id), Cell(Linear3(relu, conv3x3, instance), zero, id, zero, Linear3(mish, conv3x3, batch), avg_pool)), Linear3(Cell(zero, id, Linear3(hardswish, conv1x1, layer), Linear3(mish, conv3x3, instance), Linear3(mish, conv3x3, instance), zero), Cell(Linear3(hardswish, conv1x1, layer), id, Linear3(hardswish, dconv3x3, batch), Linear3(relu, conv3x3, batch), id, id), Cell(Linear3(relu, conv3x3, instance), zero, id, zero, Linear3(mish, conv3x3, layer), avg_pool)), Residual2(Cell(zero, id, zero, Linear3(mish, conv3x3, layer), avg_pool, Linear3(mish, conv3x3, layer)), down, down), Residual2(Cell(zero, id, zero, Linear3(mish, conv3x3, batch), avg_pool, Linear3(mish, conv3x3, layer)), down, down)), Residual3(Linear3(Cell(Linear3(mish, conv3x3, layer), id, Linear3(hardswish, dconv3x3, layer), Linear3(hardswish, dconv3x3, batch), Linear3(mish, conv3x3, instance), Linear3(mish, conv3x3, batch)), Cell(Linear3(hardswish, conv1x1, layer), id, Linear3(hardswish, dconv3x3, batch), Linear3(relu, conv3x3, layer), id, id), Cell(Linear3(relu, conv3x3, instance), zero, id, zero, Linear3(mish, conv3x3, batch), avg_pool)), Linear3(Cell(Linear3(mish, conv3x3, batch), id, Linear3(hardswish, conv1x1, batch), Linear3(mish, conv3x3, instance), Linear3(mish, conv3x3, instance), zero), Cell(Linear3(hardswish, conv1x1, layer), id, Linear3(hardswish, dconv3x3, batch), Linear3(hardswish, dconv3x3, batch), id, id), Cell(Linear3(relu, conv3x3, instance), zero, id, zero, Linear3(mish, conv3x3, layer), avg_pool)), Residual2(Cell(zero, id, zero, Linear3(mish, conv3x3, layer), avg_pool, Linear3(mish, conv3x3, layer)), down, down), Residual2(Cell(zero, id, zero, Linear3(mish, conv3x3, batch), avg_pool, Linear3(mish, conv3x3, layer)), down, down)))

ImageNet-16-120 (mean test error $52.78\,\%$, #params $0.626\,\text{MB}$, FLOPs $23.771\,\text{M}$):

Linear3(Linear4(Residual2(Cell(id, avg_pool, id, id, Linear3(relu, dconv3x3, layer), zero), Cell(Linear3(hardswish, conv1x1, batch), zero, zero, Linear3(mish, dconv3x3, layer), zero, zero), Cell(Linear3(relu, dconv3x3, layer), Linear3(mish, dconv3x3, layer), zero, Linear3(hardswish, conv3x3, layer), Linear3(relu, dconv3x3, instance), Linear3(hardswish, conv3x3, instance))), Linear2(Cell(zero, Linear3(relu, conv3x3, layer), Linear3(mish, conv1x1, batch), Linear3(mish, conv1x1, batch), avg_pool, Linear3(relu, conv3x3, layer)), Cell(id, id, Linear3(mish, conv3x3, layer), Linear3(relu, conv3x3, instance), id, id)), Residual2(Cell(zero, avg_pool, Linear3(mish, conv1x1, batch), Linear3(mish, conv1x1, layer), zero, zero), Cell(id, Linear3(relu, dconv3x3, layer), zero, zero, Linear3(relu, dconv3x3, instance), zero), Cell(id, Linear3(relu, conv3x3, layer), id, zero, zero, id)), Cell(zero, Linear3(hardswish, conv3x3, layer), avg_pool, zero, Linear3(hardswish, conv1x1, layer), id)), Residual3(Residual2(Cell(Linear3(relu, conv1x1, instance), Linear3(mish, conv1x1, layer), Linear3(mish, conv1x1, instance), zero, Linear3(hardswish, dconv3x3, layer), id), Cell(id, avg_pool, avg_pool, Linear3(relu, conv1x1, instance), id, zero), Cell(avg_pool, Linear3(mish, conv3x3, instance), Linear3(mish, conv1x1, instance), Linear3(relu, dconv3x3, batch), id, Linear3(hardswish, conv3x3, instance))), Linear2(Cell(zero, Linear3(relu, conv3x3, layer), Linear3(mish, conv1x1, batch), Linear3(mish, conv1x1, batch), avg_pool, Linear3(relu, conv3x3, instance)), Cell(id, zero, Linear3(mish, conv3x3, layer), Linear3(relu, conv3x3, instance), id, id)), Residual2(Cell(Linear3(mish, conv3x3, layer), Linear3(mish, conv1x1, batch), id, Linear3(mish, conv1x1, layer), zero, id), down, down), Residual2(Cell(Linear3(relu, conv3x3, layer), zero, Linear3(relu, dconv3x3, layer), Linear3(mish, conv1x1, layer), zero, id), down, down)), Residual3(Residual2(Cell(Linear3(mish, conv1x1, instance), Linear3(mish, conv1x1, layer), Linear3(mish, conv1x1, instance), avg_pool, Linear3(hardswish, dconv3x3, layer), id), Cell(id, avg_pool, avg_pool, Linear3(relu, conv1x1, instance), id, zero), Cell(avg_pool, Linear3(mish, conv3x3, instance), Linear3(mish, conv1x1, instance), Linear3(relu, dconv3x3, batch), id, Linear3(hardswish, conv3x3, instance))), Linear2(Cell(zero, Linear3(relu, conv3x3, layer), Linear3(mish, conv1x1, batch), Linear3(mish, conv1x1, batch), avg_pool, Linear3(relu, conv3x3, layer)), Cell(id,

zero, Linear3(mish, conv3x3, layer), Linear3(relu, conv3x3, instance), id, id)), Residual2(Cell(Linear3(relu, conv3x3, layer), avg_pool, id, Linear3(mish, conv3x3, instance), zero, id), down, down), Residual2(Cell(Linear3(relu, conv3x3, layer), zero, Linear3(relu, dconv3x3, instance), Linear3(mish, conv1x1, layer), zero, id), down, down)))

CIFARTile (mean test error 30.33 %, #params 2.356 MB, FLOPs 372.114 M):

Linear4(Residual3(Residual2(Cell(Linear3(hardswish, conv3x3, instance), id, zero, Linear3(relu, dconv3x3, instance), Linear3(mish, conv1x1, instance), avg_pool), Cell(avg_pool, avg_pool, id, zero, Linear3(hardswish, conv3x3, batch), avg_pool), Cell(Linear3(relu, dconv3x3, instance), zero, id, Linear3(relu, dconv3x3, layer), id, id)), Residual2(Cell(zero, zero, Linear3(mish, conv1x1, instance), Linear3(mish, conv3x3, batch), zero, id), Cell(Linear3(mish, conv3x3, instance), zero, Linear3(relu, dconv3x3, batch), id, Linear3(mish, conv3x3, batch), id), Cell(Linear3(hardswish, dconv3x3, batch), Linear3(relu, conv3x3, batch), Linear3(relu, conv1x1, batch), zero, Linear3(relu, conv3x3, batch), id)), Linear2(Cell(Linear3(relu, dconv3x3, layer), Linear3(mish, conv1x1, layer), id, zero, Linear3(mish, conv3x3, batch), Linear3(relu, dconv3x3, layer)), down), Linear2(Cell(id, Linear3(hardswish, conv1x1, layer), id, Linear3(relu, conv1x1, instance), avg_pool, Linear3(relu, conv1x1, layer)), down)), Residual3(Residual2(Cell(id, avg_pool, avg_pool, Linear3(hardswish, dconv3x3, instance), Linear3(mish, conv1x1, layer), Linear3(hardswish, dconv3x3, instance)), Cell(id, id, Linear3(relu, dconv3x3, layer), id, id, zero), Cell(Linear3(relu, conv3x3, layer), id, avg_pool, Linear3(mish, dconv3x3, instance), Linear3(relu, conv1x1, layer), zero)), Residual2(Cell(Linear3(mish, conv3x3, batch), Linear3(mish, conv3x3, instance), zero, avg_pool, avg_pool, Linear3(mish, conv1x1, batch)), Cell(Linear3(mish, conv1x1, batch), Linear3(relu, dconv3x3, layer), zero, id, avg_pool, avg_pool), Cell(avg_pool, Linear3(hardswish, conv1x1, instance), id, avg_pool, avg_pool, Linear3(hardswish, conv1x1, instance))), Residual2(Cell(Linear3(relu, dconv3x3, batch), avg_pool, id, avg_pool, id, zero), down, down), Residual2(Cell(zero, zero, Linear3(relu, dconv3x3, batch), avg_pool, Linear3(hardswish, conv1x1, instance), avg_pool), down, down)), Linear4(Linear3(Cell(Linear3(hardswish, conv3x3, batch), Linear3(hardswish, conv3x3, batch), Linear3(relu, conv1x1, instance), id, Linear3(relu, conv1x1, layer), Linear3(relu, conv3x3, layer)), Cell(id, Linear3(relu, conv3x3, instance), Linear3(hardswish, conv1x1, instance), Linear3(relu, conv3x3, layer), avg_pool, Linear3(mish, conv1x1, layer)), Cell(zero, zero, id, Linear3(relu, conv3x3, batch), id, Linear3(relu, conv1x1, layer))), Linear3(Cell(Linear3(hardswish, conv3x3, batch), Linear3(hardswish, conv3x3, batch), Linear3(relu, conv1x1, instance), Linear3(relu, dconv3x3, layer), Linear3(mish, conv1x1, layer), Linear3(relu, conv3x3, batch)), Cell(id, Linear3(relu, conv3x3, instance), Linear3(hardswish, conv1x1, instance), Linear3(relu, dconv3x3, instance), avg_pool, Linear3(mish, conv1x1, layer)), Cell(zero, zero, id, Linear3(relu, conv3x3, batch), id, avg_pool)), Linear3(Cell(id, id, avg_pool, Linear3(mish, conv1x1, layer), Linear3(mish, conv3x3, batch), zero), Cell(id, Linear3(relu, conv1x1, batch), avg_pool, Linear3(relu, conv1x1, layer), avg_pool, zero), Cell(zero, Linear3(relu, conv1x1, batch), Linear3(mish, dconv3x3, batch), Linear3(mish, conv1x1, batch), id, id)), Cell(id, Linear3(hardswish, conv1x1, layer), zero, id, zero, id)), Linear3(Linear2(Cell(id, zero, Linear3(mish, dconv3x3, instance), Linear3(mish, conv3x3, batch), Linear3(mish, dconv3x3, instance), Linear3(relu, conv1x1, instance)), Cell(Linear3(relu, dconv3x3, instance), avg_pool, Linear3(mish, conv1x1, instance), Linear3(hardswish, dconv3x3, instance), id, Linear3(hardswish, conv1x1, layer))), Linear2(Cell(zero, zero, Linear3(mish, dconv3x3, instance), Linear3(relu, conv3x3, instance), Linear3(hardswish, conv3x3, batch), avg_pool), Cell(id, id, Linear3(hardswish, conv1x1, instance), avg_pool, zero, Linear3(hardswish, conv3x3, batch))), Cell(avg_pool, Linear3(mish, dconv3x3, layer), zero, avg_pool, avg_pool, zero)))    .

AddNIST (mean test error 6.33 %, #params 2.853 MB, FLOPs 593.856 M):

Linear4(Residual3(Linear3(Cell(id, Linear3(hardswish, dconv3x3, batch), Linear3(relu, conv1x1, layer), Linear3(mish, conv3x3, batch), avg_pool, zero), Cell(zero, zero, avg_pool, id, avg_pool, Linear3(hardswish, conv1x1, instance)), Cell(Linear3(relu, conv3x3, layer), id, zero, Linear3(mish, conv3x3, instance), id, avg_pool)), Linear2(Cell(id, Linear3(relu, conv3x3, layer), Linear3(relu, conv3x3, layer), Linear3(hardswish, conv3x3, batch), id, Linear3(relu, conv3x3, layer)), Cell(Linear3(mish, conv3x3, instance), id, Linear3(mish, conv3x3, batch), id, avg_pool, id)), Linear3(Cell(zero, id, Linear3(relu, dconv3x3, instance), Linear3(relu, dconv3x3, layer), Linear3(relu, dconv3x3, instance), Linear3(mish, conv3x3, batch)), Cell(Linear3(mish, conv1x1, instance), zero, Linear3(relu, conv3x3, instance), id, zero, Linear3(relu, conv3x3, batch)), down), Lin-

ear3(Cell(zero, avg_pool, Linear3(hardswish, dconv3x3, layer), Linear3(relu, conv3x3, layer), Linear3(hardswish, conv1x1, instance), Linear3(hardswish, conv3x3, batch)), Cell(Linear3(hardswish, conv3x3, batch), Linear3(hardswish, conv1x1, layer), Linear3(mish, conv1x1, batch), id, Linear3(hardswish, conv3x3, batch), zero), down)), Residual3(Linear2(Cell(Linear3(mish, conv1x1, layer), avg_pool, Linear3(hardswish, dconv3x3, batch), Linear3(mish, dconv3x3, batch), id, Linear3(mish, conv3x3, layer)), Cell(zero, Linear3(relu, dconv3x3, layer), Linear3(hardswish, conv3x3, instance), avg_pool, avg_pool, zero)), Linear3(Cell(Linear3(relu, conv3x3, batch), id, Linear3(relu, conv3x3, layer), Linear3(mish, conv1x1, instance), id, Linear3(relu, dconv3x3, batch)), Cell(Linear3(mish, conv3x3, batch), Linear3(mish, conv1x1, instance), Linear3(mish, conv3x3, instance), zero, Linear3(mish, dconv3x3, layer), Linear3(relu, conv3x3, batch)), Cell(avg_pool, Linear3(mish, conv1x1, instance), Linear3(relu, conv3x3, batch), avg_pool, id, Linear3(mish, dconv3x3, batch))), Linear3(Cell(zero, avg_pool, Linear3(hardswish, dconv3x3, layer), Linear3(relu, conv3x3, batch), Linear3(hardswish, conv1x1, batch), Linear3(hardswish, conv3x3, batch)), Cell(avg_pool, Linear3(hardswish, dconv3x3, layer), Linear3(mish, conv1x1, batch), id, Linear3(hardswish, conv3x3, batch), zero), down), Residual2(Cell(zero, Linear3(mish, conv1x1, instance), Linear3(hardswish, conv1x1, instance), avg_pool, Linear3(relu, conv1x1, layer), Linear3(hardswish, dconv3x3, batch)), down, down)), Linear4(Linear2(Cell(Linear3(relu, conv3x3, instance), id, Linear3(relu, conv3x3, batch), avg_pool, zero, id), Cell(avg_pool, Linear3(hardswish, conv3x3, layer), avg_pool, Linear3(mish, conv3x3, batch), Linear3(relu, conv3x3, batch), id)), Linear2(Cell(Linear3(mish, conv1x1, layer), avg_pool, Linear3(hardswish, dconv3x3, batch), Linear3(mish, dconv3x3, batch), id, Linear3(mish, conv3x3, layer)), Cell(zero, Linear3(relu, dconv3x3, layer), Linear3(hardswish, conv3x3, instance), avg_pool, avg_pool, zero)), Linear2(Cell(id, Linear3(relu, conv3x3, instance), Linear3(relu, conv3x3, layer), Linear3(hardswish, dconv3x3, batch), id, Linear3(relu, conv3x3, layer)), Cell(Linear3(mish, conv1x1, batch), id, avg_pool, id, avg_pool, id)), Cell(id, Linear3(relu, conv3x3, layer), Linear3(mish, conv1x1, instance), Linear3(hardswish, conv3x3, batch), Linear3(mish, dconv3x3, instance), Linear3(hardswish, conv1x1, instance))), Linear4(Linear2(Cell(Linear3(relu, conv3x3, instance), id, Linear3(relu, conv3x3, batch), avg_pool, zero, id), Cell(zero, Linear3(relu, conv3x3, batch), avg_pool, Linear3(mish, conv3x3, batch), Linear3(relu, dconv3x3, instance), id)), Linear3(Cell(Linear3(relu, conv3x3, batch), id, Linear3(relu, conv3x3, layer), Linear3(mish, conv1x1, layer), id, Linear3(relu, dconv3x3, instance)), Cell(Linear3(mish, conv3x3, batch), Linear3(mish, conv1x1, instance), Linear3(hardswish, dconv3x3, instance), zero, Linear3(mish, dconv3x3, layer), Linear3(relu, conv3x3, batch)), Cell(avg_pool, Linear3(mish, conv1x1, instance), Linear3(relu, conv3x3, batch), avg_pool, id, Linear3(mish, dconv3x3, batch))), Linear2(Cell(id, Linear3(relu, conv3x3, layer), Linear3(hardswish, conv3x3, layer), Linear3(hardswish, dconv3x3, batch), id, Linear3(relu, conv3x3, layer)), Cell(Linear3(mish, conv3x3, batch), id, avg_pool, id, avg_pool, id)), Cell(id, Linear3(relu, conv3x3, layer), Linear3(mish, conv1x1, instance), Linear3(hardswish, conv3x3, batch), Linear3(mish, dconv3x3, instance), Linear3(mish, conv3x3, instance)))) .

The above reported architectures already yield strong performance. In particular, on ImageNet-16-120, CIFARTile, and AddNIST we find superior architectures with comparable number of parameters and FLOPs to architectures from common cell-based search spaces. When running the search for 50 additional evaluations, we further improved performance to a test error of $51.98\%$ on ImageNet-16-120, which is superior to the *optimal* architecture in the cell-based NAS-Bench-201 search space by $0.71\%$p. We also found an architecture on CIFAR-100, when only allowing for the same primitive computations as in the cell-based NAS-Bench-201 search space, that achieves a test error of $26.24\%$ and, thus, is also superior to the *optimal* architecture in the cell-based NAS-Bench-201 search space by $0.25\%$p. The found architectures from our huge hierarchical search spaces show that the search over a larger variety of architectures can indeed improve performance with a small search budget, even though the search problem is significantly more difficult.

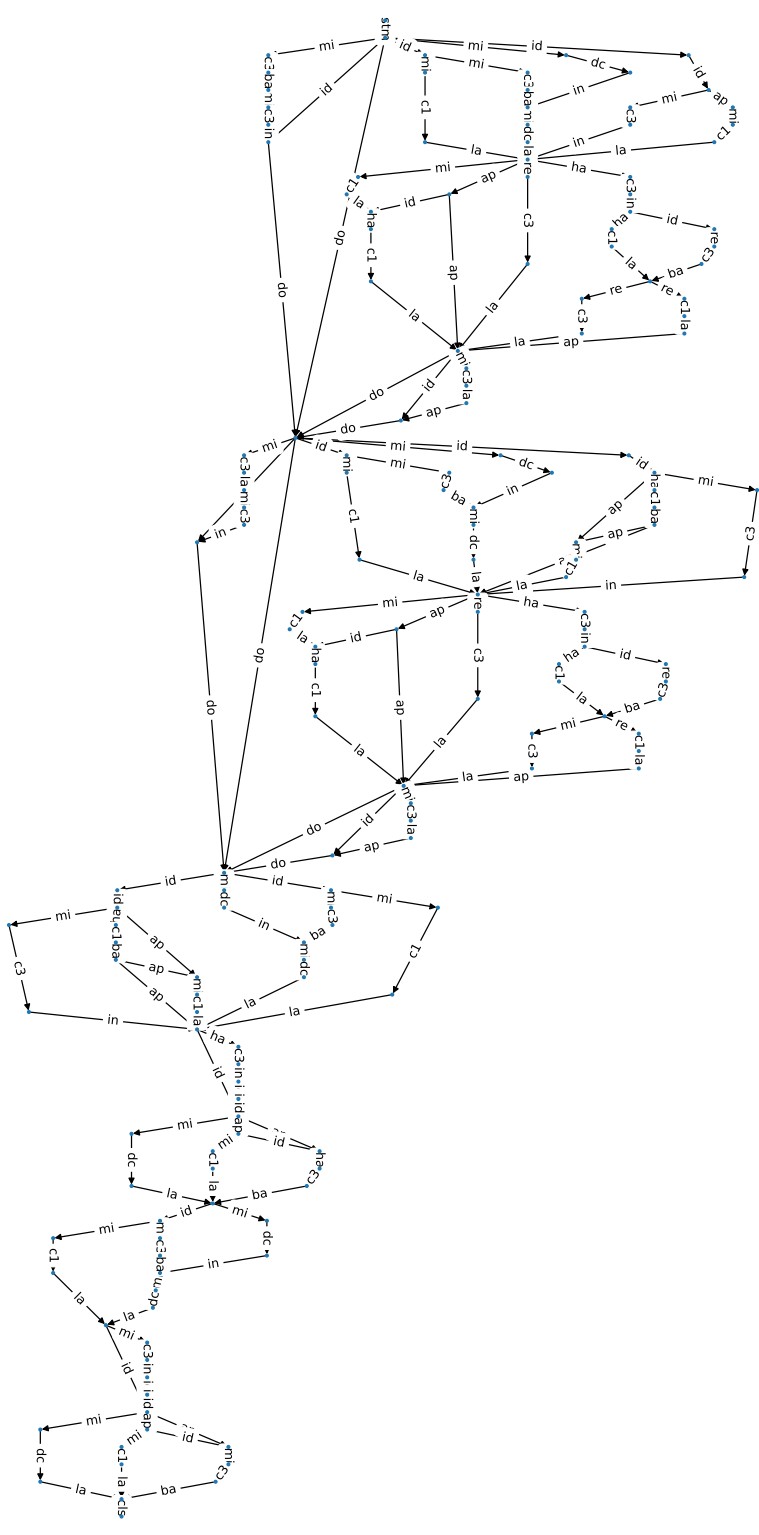

(a) CIFAR-10.

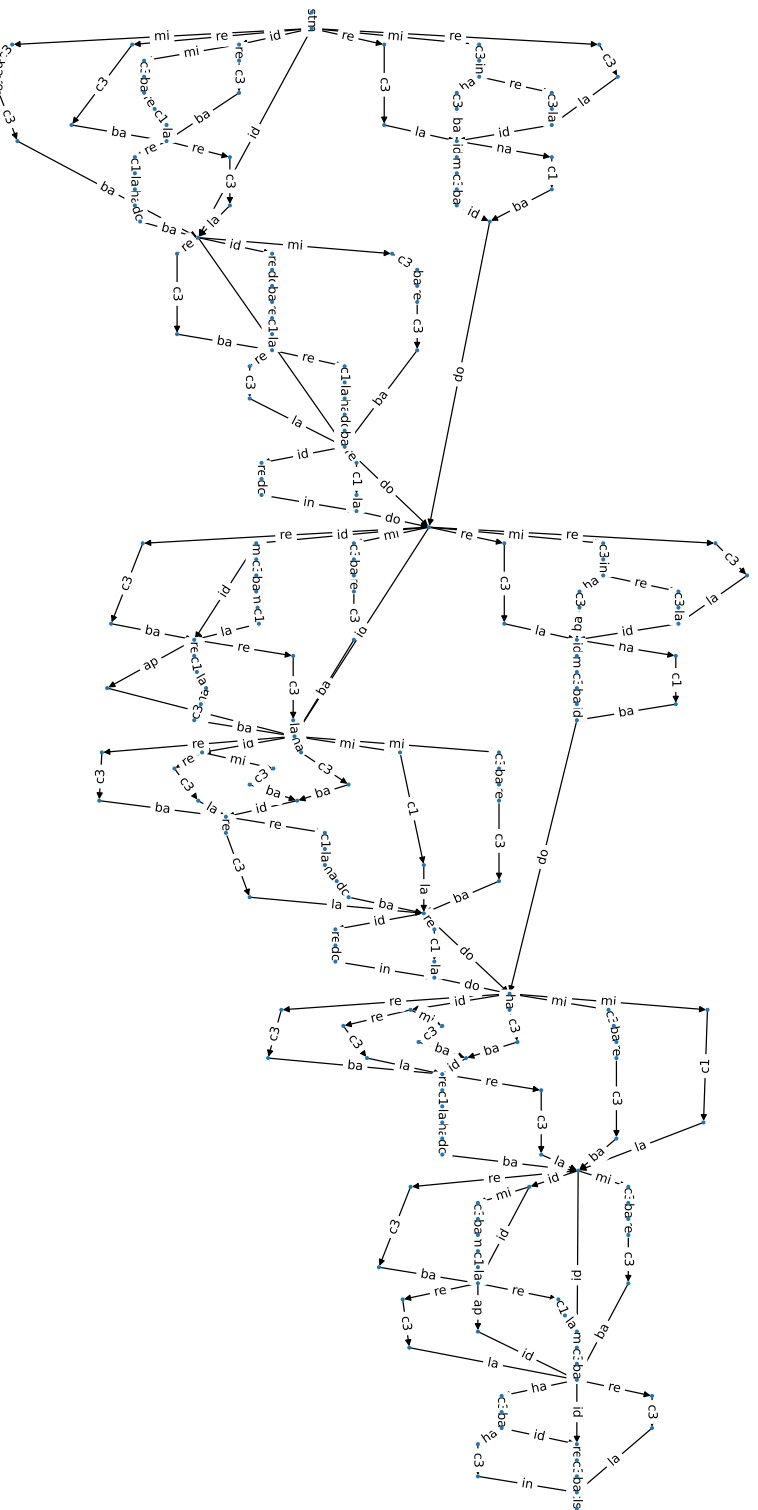

(b) CIFAR-100.

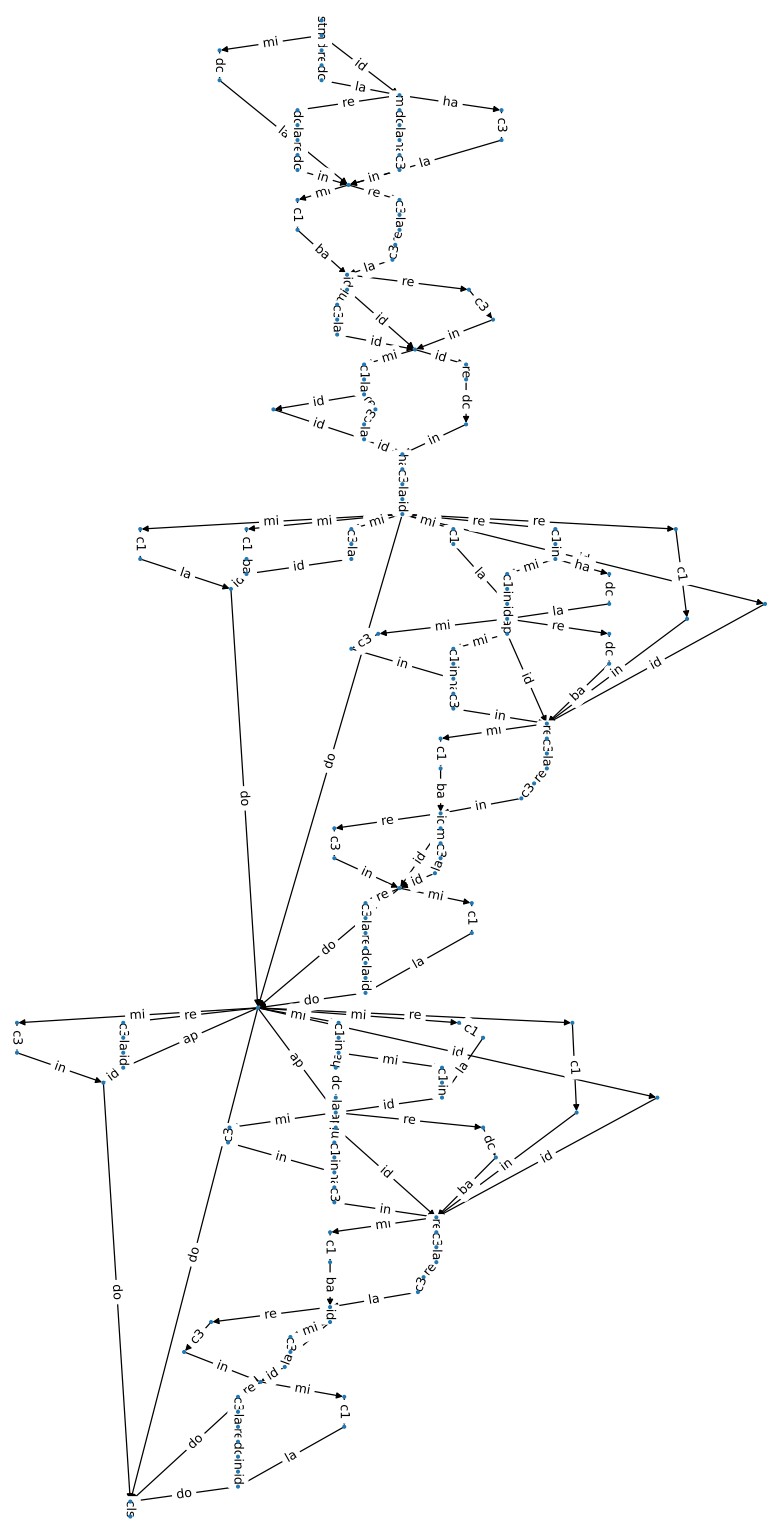

(c) ImageNet16-120.

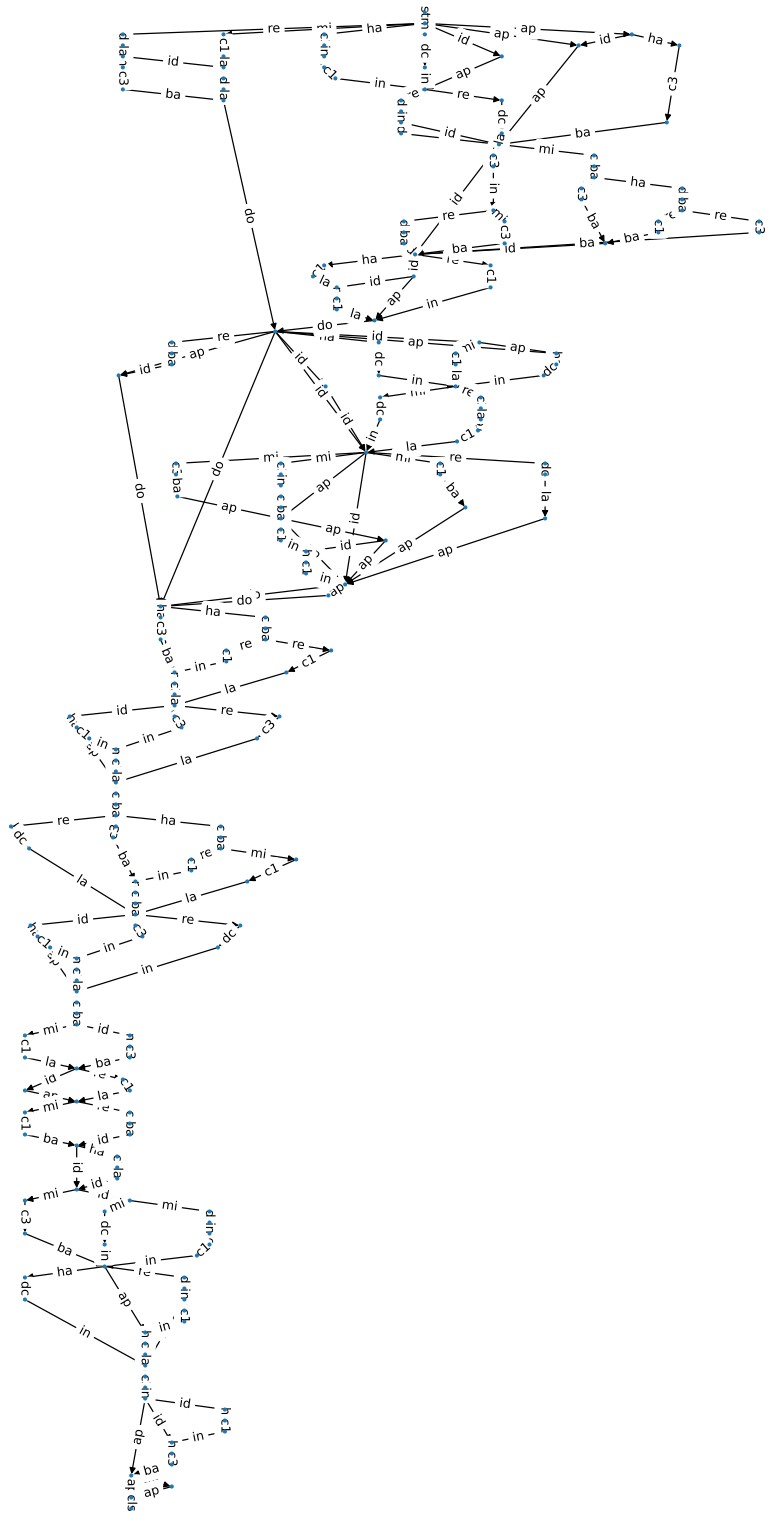

(d) CIFARTile.

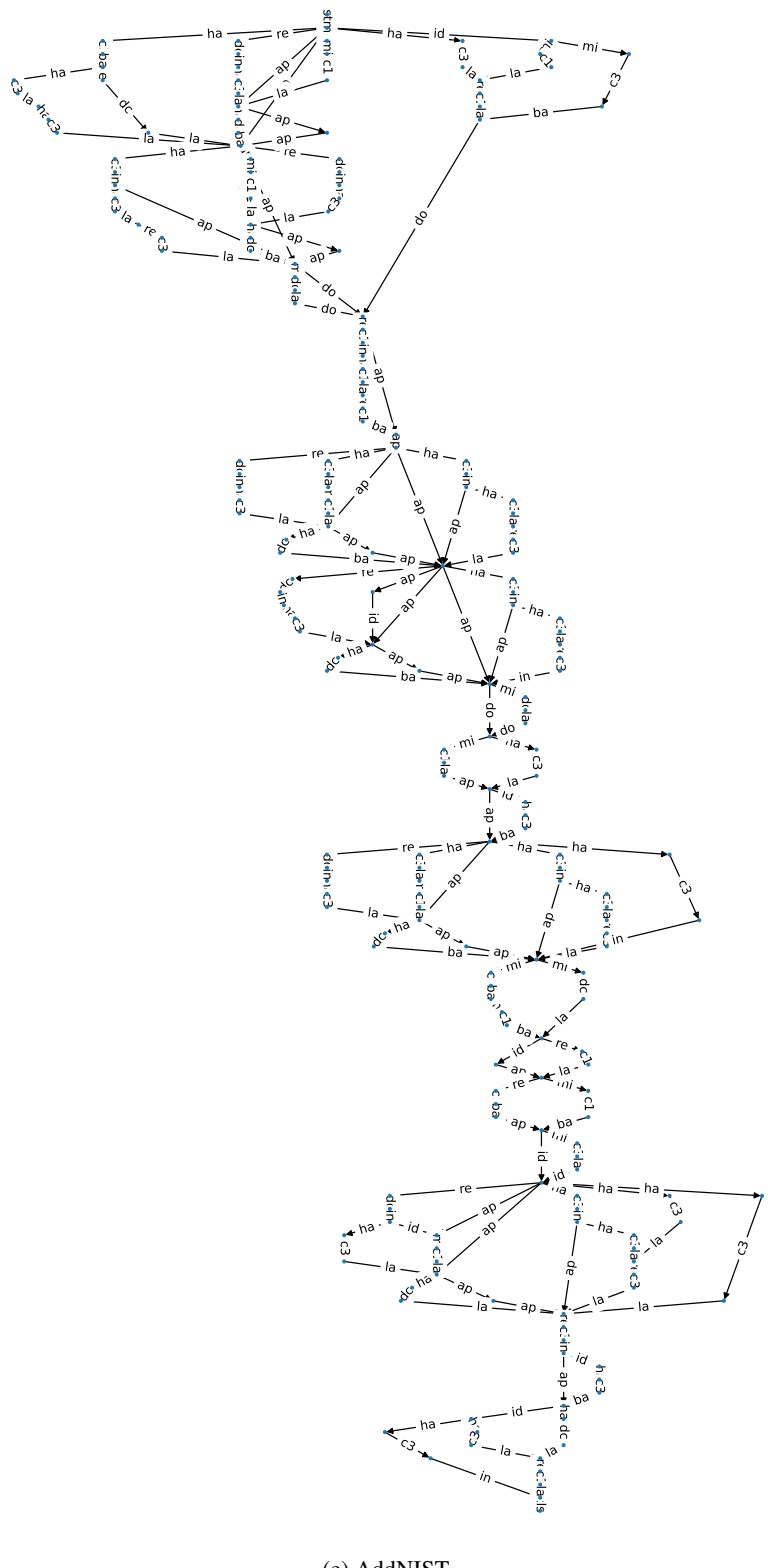

(e) AddNIST.

Figure 15: Visualization of the best found architectures in our hierarchical NAS-Bench-201 search space. Abbreviations are defined as follows: ap=avg_pool, ba=batch, c1=conv1x1, c3=conv3x3, cls=classifier, dc=dconv3x3, ha=hardswish, in=instance, la=layer, mi=mish, re=relu, and stm=stem. Best viewed with zoom.

Table 6: Definition of the binary and unary operations in the activation function search space $\mathcal{G}_{act}$. The symbol $\beta$ indicates a per-channel trainable parameter and $\sigma(\cdot)$ is the sigmoid function.

| add | $x_1 + x_2$ | bgaussian_sq | $\exp(-\beta(x_1-x_2)^2)$ | cubic | $x^3$ | cos | $\cos x$ | umax | $\max(x,0)$ |
|---|---|---|---|---|---|---|---|---|---|
| multi | $x_1 \cdot x_2$ | bgaussian_abs | $\exp(-\beta\lvert x_1-x_2\rvert)$ | square_root | $\sqrt{x}$ | sinh | $\sinh x$ | umin | $\min(x,0)$ |
| sub | $x_1 - x_2$ | wavg | $\beta x_1 + (1-\beta)x_2$ | mconst | $\beta$ | cosh | $\cosh(x)$ | sigmoid | $\sigma(x)$ |
| div | $\frac{x_1}{x_2+\epsilon}$ | id | $x$ | aconst | $x+\beta$ | tanh | $\tanh(x)$ | logexp | $\log(1+\exp(x))$ |
| bmax | $\max(x_1,x_2)$ | neg | $-x$ | log | $\log(\lvert x\rvert+\epsilon)$ | asinh | $\sinh^{-1}(x)$ | gaussian | $\exp(-x^2)$ |
| bmin | $\min(x_1,x_2)$ | abs | $\lvert x\rvert$ | exp | $\exp(x)$ | atanh | $\tanh^{-1}(x)$ | erf | $\mathrm{erf}(x)$ |
| bsigmoid | $\sigma(x_1)\cdot x_2$ | square | $x^2$ | sin | $\sin(x)$ | sinc | $\mathrm{sinc}(x)$ | const | $\beta$ |

## J SEARCHING FOR ACTIVATION FUNCTIONS

### J.1 SEARCH SPACE

To search for activation functions, we adopted the search space from Ramachandran et al. (2017). More specifically, we define the following CFG $\mathcal{G}_{act}$:

$$
\begin{aligned}
\text{L2} &::= \texttt{BinTopo(L1, L1, BINOP)} \mid \texttt{UnTopo(L1)} \\
\text{L1} &::= \texttt{BinTopo(UNOP, UNOP, BINOP)} \mid \texttt{UnTopo(UNOP)} \\
\text{BINOP} &::= \texttt{add} \mid \texttt{multi} \mid \texttt{sub} \mid \texttt{div} \mid \texttt{bmax} \mid \texttt{bmin} \mid \texttt{bsigmoid} \mid \texttt{bgaussian\_sq} \\
&\quad\mid \texttt{bgaussian\_abs} \mid \texttt{wavg} \\
\text{UNOP} &::= \texttt{id} \mid \texttt{neg} \mid \texttt{abs} \mid \texttt{square} \mid \texttt{cubic} \mid \texttt{square\_root} \mid \texttt{mconst} \mid \texttt{aconst} \\
&\quad\mid \texttt{log} \mid \texttt{exp} \mid \texttt{sin} \mid \texttt{cos} \mid \texttt{sinh} \mid \texttt{cosh} \mid \texttt{tanh} \mid \texttt{asinh} \mid \texttt{atanh} \mid \texttt{sinc} \\
&\quad\mid \texttt{umax} \mid \texttt{umin} \mid \texttt{sigmoid} \mid \texttt{logexp} \mid \texttt{gaussian} \mid \texttt{erf} \mid \texttt{const} \quad,
\end{aligned}
\tag{24}
$$

where the binary operations (BINOP) and unary operations (UNOP) are specified in Table 6 and `BinTopo` and `UnTopo` define the topology of binary or unary operations, respectively. The search space $L(\mathcal{G}_{act})$ comprises ca. $10^{8.6}$ potential activation functions. Note that the search space size can be varied by adding more levels or by using a recursive formulation, i.e., L ::= $\texttt{BinOp(L,L)} \mid \texttt{UnOp(L)} \mid \texttt{id}$.

### J.2 TRAINING DETAILS

Similar to Ramachandran et al. (2017), we trained a ResNet-20 (He et al., 2016) on CIFAR-10[3] (Krizhevsky et al., 2009). Specifically, we trained the network with SGD with learning rate of 0.1, momentum of 0.9, and weight decay of 0.0001, with a batch size of 128 for 64K steps. The learning rate is divided by 10 after 32K and 48K steps. We used random flip with probability 0.5 followed by a random crop (32x32 with 4 pixel padding) and normalization. We split the training data into 45K training or 5K validation samples, respectively, and report final test error on the unseen test set (10K samples).

### J.3 EXTENDED SEARCH RESULTS

**Search costs** The search cost was ca. 5 GPU days using eight asynchronous workers, each with an NVIDIA RTX 2080 Ti GPU, for ca. 40 GPU days in total.

**Best found activation function** Our best found activation function for ResNet-20 is as follows:

$$
\beta(\sigma(-x)\cdot\min(x,0)) + (1-\beta)(\min(\max(x,0),\mathrm{erf}\,x)) \quad.
\tag{25}
$$

Figure 16 visualizes and compares the found activation function with common activation functions from the literature.

## K BEST PRACTICES CHECKLIST FOR NAS RESEARCH

NAS research has faced challenges with reproducibility and fair empirical comparisons for a long time (Li & Talwalkar, 2020; Yang et al., 2020a). In order to promote fair and reproducible NAS

---

[3]Table 4 provides the license of CIFAR-10.

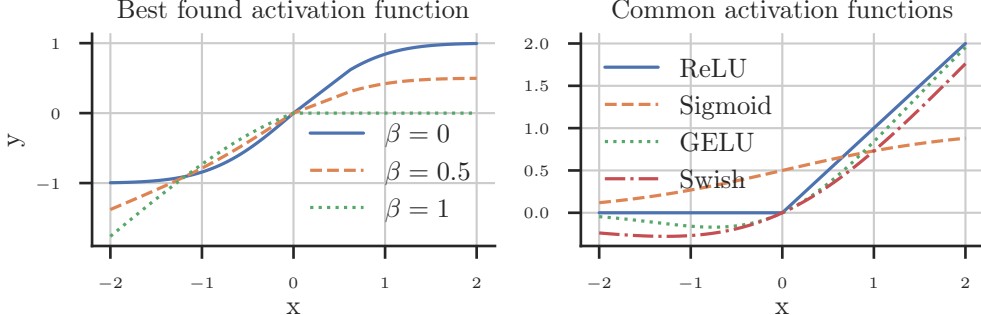

Figure 16: Visualization of our best found activation function (see Equation 25) for different values of the trainable parameter $\beta$ (left) and common activation function from the literature (right).

research, Lindauer & Hutter (2020) created a best practices checklist for NAS research. Below, we address all points on the checklist.

1. **Best Practices for Releasing Code**
   (a) *Did you release code for the training pipeline used to evaluate the final architectures?, Did you release code for the search space?, Did you release code for your NAS method?* [Yes] All code can be found at `https://anonymous.4open.science/r/iclr23_tdnafs`.
   (b) *Did you release the hyperparameters used for the final evaluation pipeline, as well as random seeds?* [Yes] We provide experimental details (including hyperparameters as well as random seeds) in Section 5, Appendix I.1 and J.2.
   (c) *Did you release hyperparameters for your NAS method, as well as random seeds?* [Yes] We discuss implementation details (including hyperparameters as well as random seeds) of our NAS method in Section 5 and Appendix H.

2. **Best practices for comparing NAS methods**
   (a) *For all NAS methods you compare, did you use exactly the same NAS benchmark, including the same dataset (with the same training-test split), search space and code for training the architectures and hyperparameters for that code?* [Yes] We always used the same training pipelines during search and final evaluation.
   (b) *Did you control for confounding factors (different hardware, versions of deep learning libraries, different runtimes for the different methods)?* [Yes] We kept hardware, software versions, and runtimes fixed across our experiments to reduce effects of confounding factors.
   (c) *Did you run ablation studies?* [Yes] We compared different kernels (hWL, WL, NAS-BOT, and GCN) and investigated the surrogate performance in Section 5. We also analyzed the distribution of architectures during search, frequency of production rules, impact of the flexible parameterization of the convolutional blocks, test error vs. number of parameters and FLOPs in Appendix I.2.
   (d) *Did you use the same evaluation protocol for the methods being compared?* [Yes]
   (e) *Did you compare performance over time?* [Yes] We plotted results over the number of iterations, as typically done for black-box optimizers, and report the total search cost in Appendix I.2. For the activation function search, we do not compare performance over time.
   (f) *Did you compare to random search?* [Yes]
   (g) *Did you perform multiple runs of your experiments and report seeds?* [Yes] We ran all search runs on the hierarchical or cell-based NAS-Bench-201 over three seeds, the search run for an activation function for one seed, and surrogate regression experiments over 20 seeds.
   (h) *Did you use tabular or surrogate benchmarks for in-depth evaluations?* [N/A] At the time of this work, there existed no surrogate benchmark for hierarchical NAS.

3. **Best practices for reporting important details**

    (a) *Did you report how you tuned hyperparameters, and what time and resources this required?* [N/A] We have not tuned any hyperparameter.

    (b) *Did you report the time for the entire end-to-end NAS method (rather than, e.g., only for the search phase?* [Yes] We report search times in Appendix I.2 and J.3. For the hierarchical NAS-Bench-201 experiments, the training protocols are exactly the same for the search and final evaluation (except for CIFAR-10), there are no additional costs for final evaluation. For the activation function search experiment, we used the same training protocol for training and evaluation.

    (c) *Did you report all the details of your experimental setup?* [Yes] We report all experimental details in Section 5, Appendix I.1, and J.2.

