# OpenReview forum: "Towards Discovering Neural Architectures from Scratch"
_ICLR.cc/2023/Conference — Submitted to ICLR 2023_

### Official Review · Reviewer_hw2q · 2022-10-23

**Confidence:** 3
**Correctness:** 4
**Technical Novelty And Significance:** 3
**Empirical Novelty And Significance:** 3
**Recommendation:** 6

**Clarity, Quality, Novelty And Reproducibility:**

The paper is clear and well-written. The search space is novel for the NAS community.

I don't seem to find the code in this submission, but the author mentioned in the abstract that they open source in PyTorch and Tensorflow.

**Strength And Weaknesses:**

Strength:

- The presented search space is novel and different from existing architectural spaces. By itself, it counts as a solid contribution to the community.
- The authors demonstrated the effectiveness of this search space over hand-engineered cell-based ones, leveraging the BO-hWL algorithm.

Weakness:

- [Minor] The search space itself still relies on existing patterns designed by human priors, e.g. the Cell(OP, OP, OP) operators, residual, and convolutions. Moreover, considering the sheer size of the search space, the search cost could be prohibitive beyond small-scale experiments. However, I don’t think it diminishes the contributions of this work. These issues could be left for future works to explore.
- [Minor] Related work: The authors did a throughout survey of relevant NAS works in Section 5. Though it might also be worthwhile to also mention the topics of program synthesis or Neural-Symbolic Programming. The NSP community also has several existing works on adopting contextual-free grammar for discovering programs [1, 2]. Some also mentioned NAS since architectures can be viewed as a subset of differentiable programs [1, 2]. This work also inspires some recent AutoML methods, such as [3] where an algebraic search space is also proposed, but for a different task.

[1] Shah et al, Learning Differentiable Programs with Admissible Neural Heuristics. NeurIPS 2020
[2] Cui and Zhu, Differentiable Synthesis of Program Architectures. NeurIPS 2021
[3] Wang et al. Efficient Non-Parametric Optimizer Search for Diverse Tasks. NeurIPS 2022

**Summary Of The Paper:**

This paper presents a novel symbolic search space for NAS that allows generating architectures without predefined connection patterns.

The search space is derived from contextual-free gramma where the networks are presented as algebraic terms formed from neural operators.

To navigate through this search space, the author leverages Bayesian Optimization with a hierarchical WL kernel.

Experiments are conducted on NAS-Bench-201, where the author shows that architectures derived from this search space surpass those generated by prior cell-based search spaces.

**Summary Of The Review:**

The search space that this paper presents is valuable to the NAS community, and I vote for acceptance.

---

> ### Author Response · Authors · 2022-11-12
> **Response to Reviewer hw2q**
>
> We would like to thank you for taking the time to make encouraging comments and constructive criticisms. We address them below. To facilitate an active discussion, we reply to you now and include updates to the paper shortly.
>
> > Reliance on human priors
>
> Yes, any algorithm requires some atomic building blocks, e.g., from basic mathematical operators. To demonstrate that our approach can discover structures from very basic primitives, we conducted an additional experiment, where we searched for activation functions based on basic mathematical operations. Following the search space from Ramachandran et. al (2017), the novel activation function we found achieves 8.31% test error compared to 8.93% (ReLU) and 8.61% (Swish) with a ResNet-20.
>
> > Contextualizing the work across fields
>
> We have added a section in the appendix (and below) to contextualize our work across the fields of neural-symbolic/differentiable programming and more general AutoML approaches. Below we have the excerpt we will add:
>
> ```
> While our work focuses exclusively on NAS, we will discuss below how it relates to the areas of optimizer search (as well as from scratch automated machine learning) and neural-symbolic programming.
>
> Optimizer search is a closely related field to NAS, where we automatically search for an optimizer (i.e., an update function for the weights) instead of an architecture. Initial works used learnable parametric or non-parametric optimizers. While the former approaches (Andrychowicz et al., 2016; Li & Malik, 2017; Chen et al., 2017; 2022a) have poor scalability and generality, the latter works overcome those limitations. Bello et al. (2017) searched for an instantiation of hand-crafted patterns via reinforcement learning, while Wang et al. (2022) proposed a tree-structured search space and searched for optimizers via a modified Monte Carlo sampling approach. AutoML-Zero (Real et al., 2020) took an even more general approach by searching over entire machine learning algorithms, including optimizers, from a generic search space built from basic mathematical operations with an evolutionary algorithm. Chen et al. (2022b) used RE to discover optimizers from a generic search space (inspired by AutoML-Zero) for training vision transformers (Dosovitskiy et al., 2021).
>
> Complementary to the above, recently there is a lot of interest in automatically synthesizing programs from domain-specific languages. Gaunt et al. (2017) proposed a hand-crafted program template and simultaneously optimized the parameters of the differentiable program with gradient descent. The HOUDINI framework (Valkov et al., 2018) proposed type-directed (top-down) enumeration and evolution approaches over differentiable functional programs. Shah et al. (2020) hierarchically assembled differentiable programs and used neural networks for the approximation of missing expression in partial programs. Cui & Zhu (2021) treated CFGs stochastically with trainable production rule sampling weights, which were optimized with a gradient-based approach (Liu et al., 2019b). However, na ̈ıvely applying gradient-based approaches does not work in our search spaces due to the exponential explosion of supernet weights, but still renders an interesting challenge for future work.
>
> Compared to these lines of work, we extended CFGs to handle changes in spatial resolution, promote regularity, and (compared to most of them) incorporate constraints, the latter two of which could also be applied in those domains. We also proposed a BO search strategy to search efficiently with a tailored kernel design to handle the hierarchical nature of the search space (i.e., the architectures).
> ```
>
>
> > I don't seem to find the code in this submission
>
> The link to the code is provided at the end of the introduction section. For your convenience, we also provide the link here: [https://anonymous.4open.science/r/iclr23_tdnafs](https://anonymous.4open.science/r/iclr23_tdnafs).

---

> > ### Comment · Reviewer_hw2q · 2022-11-20
> > **Reply to author's rebuttal**
> >
> > I thank the author for the rebuttal revisions. The author also includes additional experiments on activation function search to demonstrate the effectiveness of the proposed method during the rebuttal. I think this work presents a natural solution towards making NAS search space more generic. I would vote for acceptance.

---

### Official Review · Reviewer_P6Pi · 2022-10-25

**Confidence:** 4
**Correctness:** 2
**Technical Novelty And Significance:** 3
**Empirical Novelty And Significance:** 2
**Recommendation:** 3

**Clarity, Quality, Novelty And Reproducibility:**

The paper is clearly written but it misses some important discussion regarding comparing CFGs with conventional way of defining search spaces. The idea of using formal grammars is novel, so is the introduction of hierarchical Weisfeiler-Lehman kernel (although the latter is a minor modification). The authors provide code which helps with reproducibility.

**Strength And Weaknesses:**


**Strengths:**

 - representing neural networks and related search spaces through formal grammars is an interesting research direction with a potential to deliver a convenient tool to work with complex hierarchical search spaces
 - the paper does a good job at explaining the idea and how different search spaces can be represented using this new approach (although this is presented mainly in the Appendix)
 - limitations of the work seem adequately mentioned (although I would still have some questions, see below)

**Weaknesses:**

 - the paper does not clearly present what the benefits of adopting the proposed approach exactly are -- specifically, there are two main questions that are not sufficiently answered: 1) are there any qualitative gains from adopting grammar-based approach over traditional ways of representing search spaces (e.g., just writing PyTorch code specific to what we want to search)? By qualitative gains I mean that something becomes possible, which otherwise would not be; 2) what are the quantitative gains? That is, is it faster/easier/more flexible to use these grammars compared to using custom PyTorch code? Some important related questions:
    - in Appendix E we can see how DARTS search space can be achieved - however, a common problem with DARTS is that even though when a supernet is created each consecutive node takes all previous outputs as its input (as in Figure 6), when deriving a final architecture we only keep two inputs, meaning that some of these connections have to be removed. Is it possible to efficiently represent this kind of constraints using CFGs?
    - Is it possible to easily obtain supernets from a search space description (if it makes sense for the search space)?
    - It seems that every time a searchable graph-based structure is included in a search space (NB201 cell, DARTS cell, etc.), it is represented a custom, purpose-built terminal - does it mean that CFGs are, broadly speaking, not the best choice when searching for arbitrary connectivity between operations? Or could these structures be expressed with additional production rules and simpler terminals?
 - Related to the above, but focused more on the high-level direction of this research: considering that we have to define all terminals, nonterminals and production rules (which I imagine would involve providing implementation for each element), is the overall effort and process of designing a search space fundamentally different when using CFGs compared to just doing hierarchical NAS? Specifically, I do not see how using CFGs brings us closer to the goal of automatically discovering novel architectures (like Transfomers) - in the end it seems we are still primarily limited by the primitives (operations, architectural patterns, etc.) that we decide to include in our search space, just like it is the case for the most (all?) of the existing methods.
 - evaluation of the proposed method is rather limited - I felt a bit disappointed by its scope and the choice of baselines, specifically:
    - only one, very simple search space is considered; although the fact that NB201 search space is extended does help, still the evaluation does not really match what can be usually found in NAS papers
    - selected baselines are all very simple methods - there are many very efficient NAS methods out there, e.g., the entire research field of zero-cost NAS ("Zero-Cost Proxies for Lightweight NAS", "Neural Architecture Search without Training", "Neural Architecture Search on ImageNet in Four GPU Hours: A Theoretically Inspired Perspective", to name just a few papers) seems like a good fit to "search efficiently in the huge search spaces spanned by our algebraic architecture terms", but the authors only include random, evolutionary and BO search, which are one of the most basic approaches
    - similarly, applicability of some efficient one-shot NAS methods, which constitute an important subfield in NAS, is not explored at all (related to one of my questions above)

**Minor shortcoming and suggestions:**

 - consider changing "Linear" to, for example, "Sequential" - the word "linear" is commonly associated with linear operations, not with a chain of sequential operations
 - "However, there is of course no single search space that can construct any neural architecture" I assume the authors meant that it is impossible to define a single search space _using CFGs_, that would include any neural architecture? I would argue that an implicitly defined search space obtained by considering basic graph operations (such as add a node, add an edge, assign operation, etc.) together with a starting graph (e.g., empty graph) would include any neural network realizable in practice. How big such a search space would be, or if it would be easy to use some searching algorithms within it is a different question - although please note that there exist NAS papers that defined search spaces in such way.
 - Equations 4 and 8 are somewhat redundant, Equation 8 does not seem to add much on top of Equation 4, just minor details - in general, in my opinion, Sections 2 and 3 could be combined and made shorter, and the saved space could be used to include some of the things I mentioned are currently missing from the paper.

**Summary Of The Paper:**

The paper proposes to use context free grammars (CFGs) to represent hierarchical search spaces for neural architecture search, towards the goal of discovering new architectures, rather than refining existing ones (an example of designing Transformers over CNNs is given in the introduction).
Architectures are represented using algebraic terms, and search are then defined by the means of related sets of production rules, terminals and nonterminals, constituting a CFG.
Further, a Bayesian optimization (BO) algorithm, utilizing hierarchical Weisfeiler-Lehman kernel (hWL), is proposed to efficiently search within large search spaces, produced with CFGs.
Experiments are conducted on the NAS-Bench-201 (NB201) search space and its hierarchical variant, using CIFAR-10, CIFAR-100, ImageNet-16-120, CIFARTile and AddNIST datasets.

**Summary Of The Review:**

While I like the motivation presented in the paper and the idea of using formal grammars seems somewhat compelling to me, in my opinion the paper currently does not contain convincing argumentation for using CFGs, neither does it provide sufficiently strong/interesting experimental results to consider it for publication.

---

> ### Author Response · Authors · 2022-11-12
> **Response to Reviewer P6Pi 1/2**
>
>
> We would like to thank you for taking the time to appreciate the value of our work as an “interesting research direction” and making constructive criticisms. To facilitate an active discussion, we reply to you now and include updates to the paper shortly.
>
> > What are the (qualitative & quantitative) benefits of context-free grammars over traditional ways of representing search spaces (e.g., just writing PyTorch code specific to what we want to search)?
>
> Our context-free grammars offer a *compact* and *formally* grounded way to *quickly* define hierarchical search spaces of PyTorch code (i.e., the architecture). We are not aware of any (efficient) search space construction technique in the NAS literature that could fully represent the hierarchical NAS-Bench-201 search space we consider in our paper. While, obviously, we can easily express hierarchically assembled architectures with PyTorch code, the evolution (i.e., mutation and crossover) is more complicated to implement (in fact, you will likely end-up with a context-free grammar-like approach), while we can build upon the *rich literature in grammar-guided evolution/genetic programming* with context-free grammars. The other main benefit of context-free grammars is that we can precisely define what architectures are in and which are not in the search space (i.e., *word problem*). Compared to traditional cell-based search space implementations, which are typically an n-dimensional categorical search space, context-free grammars are significantly more flexible and easier to define.
>
> > Why do we need to sometimes define purpose-built terminals for cells (e.g., DARTS)?
>
> This is no limitation of context-free grammars, but hierarchical search spaces (Liu et al, ICLR 2018) in general. More specifically, the locality property of replacements is the reason that one cannot construct cells, such as DARTS. While this seems limiting at first sight, we would like to note that such arbitrary connectivity is typically only used at cell-level choices, whereas the macro-level architectures are typically very hierarchical. Thus, a mixture of a hierarchical search space at the macro-level combined with a more flexible search space design at the cell-level may be an interesting future research direction to explore.
>
> > There is only one very simple search space and simple baselines?
>
> To show that we can also search over basic mathematical operations, we added an additional search space to search for activation functions. Following the search space from Ramachandran et. al (2017), the novel activation function we find achieves 8.31% test error compared to 8.93% (ReLU) and 8.61% (Swish) with a ResNet-20.
>
> Regarding baselines, aside from random search, we compared against regularized evolution (i.e., used, e.g., in Liu et al, ICLR 2018 as well as it is more generally representing evolutionary algorithms used, e.g., in almost all grammar-based NAS approaches from our related work), and the state-of-the-art Bayesian optimization method (i.e., NAS-BOWL and generally representing Bayesian optimization methods).  Further, we have now conducted a comparison to NASBOT, which is clearly outperformed by our proposed surrogate when applied to our search space (we will add this to Figure 4 in the revised manuscript). We have also tried the GCN surrogate proposed in BONAS  but it did not work, yielding only constant prediction demonstrating that they could not learn a high-quality mapping from the complex graph space to performance. Thus, we do not plot results for this as the correlation metrics are NaN. In the responses below, we add a comparison to zero-cost proxies and discuss one-shot NAS baselines.

---

> > ### Author Response · Authors · 2022-11-12
> > **Response to Reviewer P6Pi 2/2**
> >
> > > Zero-cost proxies for hierarchical search spaces
> >
> > To compare to zero-cost proxies, we ran all 1200 architectures from our search runs of the hierarchical NAS-Bench-201 space across the five datasets and recorded Kendall’s tau rank correlation in the table below:
> >
> > | Zero-cost proxy | Type | CIFAR-10 | CIFAR-100 | ImageNet16-120 | CIFARTile | AddNIST |
> > | ------------- |:-------------:|:-------------:|:-------------:|:-------------:|:-------------:|:-------------:|
> > `plain` | Baseline | -0.01 | 0.01 | -0.08 | -0.0 | 0.04 |
> > `params` | Baseline | 0.39 | 0.31 | 0.13 | 0.28 | 0.5 |
> > `flops` | Baseline | 0.46 | **0.51** | 0.47 | **0.47** | 0.49 |
> > `l2-norm` | Baseline | 0.4 | 0.29 | 0.23 | 0.34 | **0.51** |
> > `zen-score` | Piece. Lin. | **0.47** | 0.41 | 0.34 | 0.24 | 0.4 |
> > `fisher` | Pruning-at-init | -0.06 | -0.03 | -0.05 | 0.03 | 0.2 |
> > `grad-norm` | Pruning-at-init | 0.09 | 0.04 | 0.01 | 0.15 | 0.24 |
> > `grasp` | Pruning-at-init | 0.08 | 0.17 | 0.19 | 0.02 | 0.03 |
> > `snip` | Pruning-at-init | 0.17 | 0.06 | -0.01 | 0.21 | 0.29 |
> > `synflow` | Pruning-at-init | 0.06 | 0.24 | 0.28 | -0.18 | -0.08 |
> > `epe-nas` | Jacobian | 0.34 | 0.29 | 0.26 | 0.23 | 0.09 |
> > `jacov` | Jacobian | 0.4 | 0.34 | 0.34 | 0.25 | 0.12 |
> > `nwot` | Jacobian | 0.32 | 0.44 | **0.58** | 0.34 | 0.22 |
> >
> > We also ran NASWOT for 10, 100, 1000, and 10000 number of architectures and report test errors below:
> >
> > | NASWOT | CIFAR-10 | CIFAR-100 | ImageNet16-120 | CIFARTile | AddNIST |
> > | ------------- |:-------------:|:-------------:|:-------------:|:-------------:|:-------------:|
> > | NASWOT (N=10) | 8.18 | 31.73 | 58.66 | 49.46 | 11.81 |
> > | NASWOT (N=100) | 8.56 | 31.65 | 58.47 | 43.31 | 14.47 |
> > | NASWOT (N=1000) | 8.26 | 31.66 | 58.33 | 45.66 | 13.57 |
> > | NASWOT (N=10000) | 8.4 | 32.09 | 57.58 | 42.45 | 14.82 |
> > | BANAT (ours, N=100) | 6.0 | 27.57 | 53.43 | 32.28 | 6.09 |
> >
> > > One-shot methods for hierarchical search spaces
> >
> > The derivation of the supernet from a search space is simple. In a nutshell, we can assign architectural weights to the production rules recursively. However, the supernet will, unfortunately, scale exponentially in the number of hierarchical levels and thus the memory cost quickly becomes unmanageable without any further adoption of one-shot methods.
> >
> > > I would argue that an implicitly defined search space obtained by considering basic graph operations (such as add a node, add an edge, assign operation, etc.) together with a starting graph (e.g., empty graph) would include any neural network realizable in practice.
> >
> > While such implicit construction techniques may construct any architecture, they will also construct architectures that are invalid. For example, an implicit construction technique ignores the resolution changes induced by assigned operations and may construct a lot of invalid architectures without heavy modification or post-sampling/evolution testing for resolution mismatches.
> >
> > > References
> >
> > (Liu et al, ICLR 2018): Hanxiao Liu, Karen Simonyan, Oriol Vinyals, Chrisantha Fernando, and Koray Kavukcuoglu. Hierarchical Representations for Efficient Architecture Search. In International Conference on Learning Representations, 2018.
> >
> > (Ramachandran et al, arXiv 2017): Prajit Ramachandran, Barret Zoph, and Quoc V. Le. Searching for Activation Functions. arXiv, 2017.

---

> > > ### Comment · Reviewer_P6Pi · 2022-12-01
> > > **Addressing rebuttal**
> > >
> > > I would like to thank the authors for their response.
> > > However, even though it includes some interesting comments, I do not think it is enough to change my position on the submitted paper.
> > > In summary, it seems to me that the authors try to leave too much as an exercise for the reader, meaning that a lot of important information is simply missing from the paper.
> > >
> > > Regarding the benefits of using CFGs (qualitative and quantitative):
> > >
> > >  - the authors say that their work offer a "compact and formally grounded way to quickly define hierarchical search spaces" - while I don't necessarily disagree with this statement, I would argue that the submitted paper contain very little information about: 1) how compact and quick the process of defining search spaces really is, 2) what are the specific benefits of having this grounding in the theory of CFGs.
> > >  Even though the authors mention some things (e.g., mutating), these things are never systematically presented and compared.
> > >  - the paper mentions quite a few related systems, some of them also using CFGs, and simply states that: "Different to these prior works, we construct entire architectures with spatial resolution changes across multiple branches, and propose techniques to incorporate constraints and foster regularity." However, it is not clear from reading the paper what is really meant here, and thus what are the main differentiating points from the prior works. Specifically, although section 2.3 seems to have information about these things, it appears rather superficial to me. For example, the part about constraints more or less says that we can include constraints if we make sure our models never violate the constraints - it is not exactly informative, especially considering that the experiments presented do not seem to show anything related to constraints, implementing them in practice, etc. Similar comment(s) could be made about substitutions (are they used anywhere in the paper?).
> > >
> > > Regarding the answer about representing cells, I am not sure I understand fully, but either way something is off.
> > > So, the answer focuses on the limitations of hierarchical search spaces and mentions that they are unable to represent arbitrary connectivity - does that mean that CFGs are only suitable to represent hierarchical search spaces? If yes, then it goes against some of the comments made in the paper, such as split into G_macro and G_cell mentioned in 2.3 (basically, it says "we can represent cell-based search spaces" but it seems like this is only possible if we delegate this to something else than CFGs...); on the other hand, if it's limitation specific to hierarchical search spaces but not necessarily CFGs, then I don't understand why the authors decided to make a comment like that in the first place.
> > >
> > > Regarding zero-cost proxies, it seems that the authors do not align searching budgets correctly - BANAT should be compared to AREA from the NASWOT paper, rather than simply NASWOT(N=...).
> > >
> > > More generally, I think the paper could benefit greatly from focusing solely on representing search spaces with CFGs.
> > > That would entitle including more comprehensive analysis of what are the benefits and limitations of such an approach as well as including systematic comparison with existing approaches (e.g., on top of what is mentioned in related work, how about comparing to PyGlove?).
> > > I would imagine this to be either more on the theory side where the authors could very specifically show what are the unique, qualitative benefits of employing CFGs _for NAS_, or on the practical side which would mean showing how easy it is to use the developed tool etc. Ideally both.
> > > Consequently, I don't think including BONAT in the same paper is really critical - these two parts feel very orthogonal and make the focus of the paper somewhat diluted.
> > > It is not a problem that the authors include both per se, but right now it seems that because of that neither part is adequately studied (BONAT as a searching methodology would need to be tested on more search spaces, etc.).
> > >
> > > I hope that the comments turn out useful for improving the paper in the future, but unfortunately right now I am not convinced the paper contains enough to recommend acceptance.

---

> > > > ### Author Response · Authors · 2022-12-12
> > > > **More experiments and response to comments 1/2**
> > > >
> > > > We thank you for your very helpful response. We want to address your comments below:
> > > >
> > > > > What are the benefits in the grounding in CFGs? How does our work differ from prior works using CFGs?
> > > >
> > > > The benefits in the grounding in CFGs are as follows:
> > > > - search methods, like ours, can build upon the extensive work of the grammar-guided genetic programming research community, e.g., how to evolve architectures.
> > > > - they define a search space of only valid architectures (c.f., word problem). This is in contrast to implicit search space designs, which do not define a search space of valid architectures and can be hard to reason about and work with.
> > > > - CFGs naturally construct hierarchical spaces and due to their recursive nature, we can describe complex hierarchical search spaces in a very compact and simple manner.
> > > >
> > > > While CFGs already provide a powerful search space design framework, prior works also used them and, thus, we would like to highlight differences to those prior works below. Prior works
> > > > - cannot represent the design pattern of repeating a sub-structure throughout the architecture (e.g., a shared cell design or a shared activation function design).
> > > > - mostly limited themselves to linear macro topologies and often did not allow more complex macro architectures with different branches and downsampling strategies.
> > > > - did not allow for incorporating constraints; thereby could construct invalid neural architectures, e.g., ones where the input is disassociated from the output (as done for our hierarchical NAS-Bench-201 search space).
> > > >
> > > > Overcoming these limitations enhances the representational capacity of CFG-based search space construction and allows our methodology to be an unifying framework to design search spaces (see Appendix E for examples of common search spaces from the literature).
> > > >
> > > > Would you deem a discussion as above helpful to the paper?
> > > >
> > > > > How compact and quick is the process of search space definition?
> > > >
> > > > Compactness stems directly from CFGs. In code, the search space definition looks very similar to the CFG representation (e.g., as Equation 8) and can be modified easily. Would it be helpful to showcase the search space construction with code examples from our open-sourced code directly in the paper?
> > > >
> > > > Regarding how quick the process is to define search spaces: this largely depends on the familiarity with NAS search spaces, complexity of the considered search space, as well as familiarity in designing search spaces with CFGs. After some training, we feel confident that practitioners will be able to define search spaces comfortably and quickly. How do you suggest we incorporate this aspect in the paper?
> > > >
> > > > > Zero-cost experiment with AREA
> > > >
> > > > As suggested by you, we also ran AREA; it improved performance over RE but performed worse than our proposed search strategy BANAT.
> > > >
> > > > | Search strategy| CIFAR-10 | CIFAR-100 | ImageNet16-120 | CIFARTile | AddNIST |
> > > > | ------------- |:-------------:|:-------------:|:-------------:|:-------------:|:-------------:|
> > > > | NASWOT (N=10) | 8.18 | 31.73 | 58.66 | 49.46 | 11.81 |
> > > > | NASWOT (N=100) | 8.56 | 31.65 | 58.47 | 43.31 | 14.47 |
> > > > | NASWOT (N=1000) | 8.26 | 31.66 | 58.33 | 45.66 | 13.57 |
> > > > | NASWOT (N=10000) | 8.4 | 32.09 | 57.58 | 42.45 | 14.82 |
> > > > | RE (N=100) | 6.88 | 30.0 | 55.39 | 40.99 | 7.56 |
> > > > | AREA (N=100) | 6.42 | 27.94 | 53.33 | 37.78 | 7.53 |
> > > > | BANAT (ours, N=100) | 6.0 | 27.57 | 53.43 | 32.28 | 6.09 |

---

> > > > > ### Author Response · Authors · 2022-12-12
> > > > > **More experiments and response to comments 2/2**
> > > > >
> > > > > > Comparison on BANAT on more search spaces
> > > > >
> > > > > Additionally to the hierarchical NAS-Bench-201 search space, activation function space, and ablations in the appendix, we ran BANAT on the DARTS search space as requested by another reviewer. See [this response](https://openreview.net/forum?id=UIpwFLrJiDi&noteId=TdVHax1iN8h) for more details.
> > > > >
> > > > > > Should the work solely focus on the search space construction method?
> > > > >
> > > > > We agree that this would improve the focus of the work, but argue below that our search method BANAT is a crucial contribution, describing how current methods from the literature did not work:
> > > > >
> > > > > - evolutionary algorithms: we tried RE (the most common evolutionary algorithm in the NAS literature) but as can be seen throughout our experiments, it did not work particularly well, sometimes not even outperforming random search.
> > > > > - gradient-based methods: current gradient-based methods do not scale to our hierarchical spaces since the supernets would yield an exponential number of model parameters. Thus, they would require adoption to also work in our hierarchical spaces.
> > > > > - reinforcement learning: reinforcement learning typically is not sample efficient and would also likely require adoption.
> > > > > - zero-cost methods: Zero-cost proxies show very weak rank correlations for our search spaces (see Table 1) and AREA also does not yield better architectures across datasets.
> > > > > - Bayesian optimization: BANANAS’ path encoding does not scale to hierarchical spaces. The same is true for the adjacency encoding. GCNs could not learn the mapping from architectures to performance. NASBOT does not work particularly well. While NAS-BOWL worked best, it often does not even discover architectures on-par with the cell-based search spaces, even though the cell-based spaces are deliberately a subset of our hierarchical spaces.
> > > > >
> > > > > Therefore, we proposed a novel Bayesian optimization algorithm that can take the hierarchical nature of the search spaces into account, since otherwise the paper would suffer from not having "interesting/strong" experimental results. Note that this is still being mentioned as a concern by a couple of reviewers, even though we find architectures that are superior to the best architectures in the cell-based NAS-Bench-201 search space while having a significantly more challenging search space, or show that we can find a well-performing activation function.

---

### Official Review · Reviewer_Y2hs · 2022-10-25

**Confidence:** 3
**Correctness:** 3
**Technical Novelty And Significance:** 3
**Empirical Novelty And Significance:** 3
**Recommendation:** 6

**Clarity, Quality, Novelty And Reproducibility:**

Clarity: difficult to follow, apart from the experimental section.
Quality: the work is methodologically interesting, but some experimental justifications are missing.
Novelty: the work is novel but missing comparisons on related work on from-scratch AutoML, as listed above.
Reproducibility: good.

**Strength And Weaknesses:**

### Strengths:
1. The CFG-based approach for specifying large search spaces seems fairly novel and interesting, in-particular allowing for very large search spaces while dealing with issues of dimensionality and other constraints.
2. The authors provide code and furthermore make an effort to ensure impact via both PyTorch and Tensorflow APIs.
3. The presentation of the experimental section is very clear and directly lays out what questions are being answered. Empirically, the results show that the larger search space can be fruitfully searched for stronger architectures than are contained in the NB201 search space, at least on ImageNet-16-120 and the two recently introduced tasks.

### Weaknesses:
1. The “from Scratch” nature of the work may be somewhat overstated, as the user must still specify primitives such as convolutions being used. The authors motivate the work by noting the Transformer was not discovered by NAS, but is there any hope that a search space generated along these lines would encode its crucial attention mechanism while not being so vast as to be unsearchable? Notably, there has been recent work on “from Scratch” AutoML (Real et al., 2020) and NAS (Roberts et al., 2021) that do aim for such generality.
2. It is not entirely clear that the CFG formalism is crucial for defining and constraining search spaces. It could be useful to compare to other AutoML search space definitions, e.g. the domain-specific language for optimizer search of Bello et al. (2017).
3. The related work noted in the two points above is missing.
4. The methods sections of the paper are difficult to follow without either familiarity with CFGs (not safe to assume for ICLR). There is a lot of relegation of detail to the appendix, both for the search space design and for the search methods.
5. The experimental section could benefit from demonstrations with search spaces beyond NB201, and on tasks beyond computer vision, especially since as the authors note the algorithms for vision have already been highly optimized. Some important comparisons/ablations are missing, such as whether FLOPs are also comparable to NB201, and whether it is important to have multiple kinds of activation function in the search space, or if performance is the same if only the one used by NB201 is allowed?

### Questions:
- Is “any neural architecture can be represented algebraically” a formal claim? If yes where is the definition of a neural architecture and proof of the result? Does this fact hold e.g. for recurrent nets?
- Equation 4: presumably f also includes some fixed training procedure for all networks?
- Why is “our grammar-based mechanism does not (generally) support simple scalability of discovered neural architectures (e.g., repetition of building blocks)” true given the use of the CL to capture NB-201?

### References:
- Bello, Zoph, Vasudevan, Le. *Neural optimizer search with reinforcement learning*. ICML 2017.
- Real, Liang, So, Le. *AutoML-Zero: Evolving machine learning algorithms from scratch*. ICML 2020.
- Roberts, Khodak, Dao, Li, Re, Talwalkar. *Rethinking neural operations for diverse tasks*. NeurIPS 2021.

**Summary Of The Paper:**

This paper introduces a CFG-based approach for specifying architecture search spaces. They demonstrate the expressivity of their approach, propose a BO-based schemes for searching it while handling the massive hierarchical search space, and evaluate by searching for architectures on several vision datasets.

**Summary Of The Review:**

I believe the work to be an interesting approach to discovering neural architectures from scratch, and there are some interesting experimental results, so I would be in favor of the paper appearing at the conference. The fairly limited nature of the empirical evaluation, missing discussion of recent work on from-scratch AutoML, and opaque presentation of the method prevent me from giving a stronger endorsement.

---

> ### Author Response · Authors · 2022-11-12
> **Response to Reviewer Y2hs**
>
> We would like to thank you for taking the time to make encouraging comments and constructive criticisms. To facilitate an active discussion, we reply to you now and include updates to the paper shortly (including your suggestions).
>
> > It is not entirely clear that the CFG formalism is crucial for defining and constraining search spaces.
>
> The main benefit of CFGs is that they give us a formally grounded way to naturally, quickly, and compactly define hierarchical search spaces. Incorporating constraints or handling spatial resolution changes can also be easily implemented. Furthermore, fostering regularity is just leveraging the closure property of context-free languages under substitution and the evolution of architectures (i.e., mutation and crossover) can rely upon the rich literature of grammar-guided evolution/genetic programming. Also, we are not aware of any (efficient) search space construction technique in the NAS literature that could fully represent the hierarchical NAS-Bench-201 search space we consider in our paper without requiring either adaption or post hoc testing of the validity (e.g., there are no shape mismatches) of architectures.
>
> > Discussion of related work across other fields beyond NAS
>
> We will add a section in the Appendix to contextualize our work across the fields of neural-symbolic/differentiable programming, optimizer search, and more general AutoML approaches. For the text we will add please see our [reply to the reviewer hw2q](https://openreview.net/forum?id=UIpwFLrJiDi&noteId=b6kxHRU2Buy).
>
> > Improvement of experimental justifications
>
> As suggested by you, we added an additional search space (i.e., searching for activation functions), which shows that similar to the results on the hierarchical NAS-Bench-201 search space we can find well-performing activation functions.
>
> We will also add a note on the comparability of architectures w.r.t. FLOPs and an ablation on the importance of having multiple kinds of activation functions and normalizations to the appendix.
>
> > Is “any neural architecture can be represented algebraically” a formal claim?
>
> No, we do not make a formal claim here. Intuitively, the universe of algebras can contain any element (including all architectures). Thus, this statement is trivially true. However, note that there is no CFG that can generate (!= represent) any architecture. Proof idea by counterexample: we cannot generate an architecture of the type a^n b^n c^n with n>0, which can be proven using the pumping lemma. How would you suggest we change this sentence to make this more clear?
>
> > Equation 4: presumably f also includes some fixed training procedure for all networks?
>
> Yes, including the same random seed. Note that while this makes perfect sense for pure NAS methods, there are good reasons to believe that the assembled architectures, which may be very different from each other, require different hyperparameters to perform well. Thus, it is likely a fruitful direction to explore hierarchical search spaces in a joint NAS+HPO setting.
>
>
> > Why is “our grammar-based mechanism does not (generally) support simple scalability of discovered neural architectures (e.g., repetition of building blocks)” true given the use of the CL to capture NB-201?
>
> Yes, that is completely right. While we still support scalability for *special* cases, as you mentioned, we wanted to stress that generally, this may not be the case with this statement.

---

> > ### Comment · Reviewer_Y2hs · 2022-11-27
> > **Response**
> >
> > Thank you for the response. After reading this and the other reviews, my opinion remains largely the same, and I will maintain my score.
> >
> > On the question of how to restate “any neural architecture can be represented algebraically,” I think if the claim is trivially true then it would be better to make claims specific to the contribution. It seems that the key observation is not this claim, but the fact that CFGs can use it to "give us a formally grounded way to naturally, quickly, and compactly define hierarchical search spaces."

---

> > > ### Author Response · Authors · 2022-12-12
> > > **Making Formulations more Clear**
> > >
> > > We thank you for your helpful suggestion. The statement “any neural architecture can be represented algebraically” was only intended as a side remark, but we see how our previous formulation could be misinterpreted and will make this more clear. We will also feature the main advantages of using CFGs to define search spaces more prominently.

---

### Official Review · Reviewer_Hxvj · 2022-10-27

**Confidence:** 5
**Correctness:** 2
**Technical Novelty And Significance:** 2
**Empirical Novelty And Significance:** 2
**Recommendation:** 3

**Clarity, Quality, Novelty And Reproducibility:**

## Clarity
- The writing could be improved, especially in the introduction, which focuses a lot on discovering novel architectures although this is not shown in the experiments.
- It is unclear how we go from algebraic representation to graph for the hierarchical Weisfeller-Lehman kernel (hWL).
## Quality
- Quality of the experimental section is low due to missing comparison to other BO NAS methods and just one studied search space of limited size.
## Reproducibility
- Aside from the point of how we go from algebraic representation to graph for hWL, the experimental description is thorough and appears reproducible.


**Strength And Weaknesses:**

Pros:
- The CFG formulation of NAS search spaces is natural and flexible.

Cons:
- The experiments are only conducted on NASBench201 style search spaces, which are small and do not achieve SOTA accuracy.
- BANAT is only evaluated against simple baselines like random search and evolutionary search and not more adaptive competitors, in particular, other Bayesian NAS methods like HNAS (Ru et al., 2021), BANANAS, NASBOT, etc.
- No evidence that the additional expressivity of the search space offered by BANAT results in interesting novel architectures.

**Summary Of The Paper:**

This paper proposes to represent neural architectures as algebraic terms and the design space (i.e. neural architecture search space) as context-free grammars (CFGs).  The authors then develop a Bayesian optimization algorithm building on top of the work by Ru et al., 2021 called BANAT, that exploits the CFG representation to define a hierarchical kernel. Results show BANAT benefits from the hierarchical kernel and outperforms common NAS baselines like random search and evolutionary search on NASBench-201 style search spaces.

**Summary Of The Review:**

The authors present a simple, flexible formulation of neural architectures as algebraic terms and search spaces as CFGs. Along with this flexible search specification, the authors introduce BANAT, a BO-based NAS search method, that uses a hierarchical representation of architectures to learn and transfer across different granularities of representation in architecture space.  Although results look promising on a suite of experiments based on the NASBench201 benchmark, they do not
sufficiently validate NASBAT relative to other baselines, nor do they provide enough coverage of different search spaces, in particular those that reach SOTA on CIFAR-10/ImageNet.

---

> ### Author Response · Authors · 2022-11-12
> **Response to Reviewer Hxvj**
>
> We would like to thank you for taking the time to review this paper and make constructive criticisms. To facilitate an active discussion, we reply to you now and include updates to the paper shortly.
>
> > “The experiments are only conducted on NASBench201 style search spaces, which are small and do not achieve SOTA accuracy”.
>
> We agree that we do not achieve SOTA accuracy numbers on the datasets, but this is mostly due to the training protocol and other design choices such as operation choices, although, on ImageNet-16-120 our search yields state-of-the-art results compared to other NAS methods. However, we strongly disagree that the hierarchical search space we considered is small and try to contextualize the search space size, i.e., compared to the DARTS search space, **our considered hierarchical search space is a factor of ~10^428 larger**. Other common search spaces can be found in the table below.
>
> | Search space        | size |
> | ------------- |:-------------:|
> | NAS-Bench-201     | 1.5*10^4 |
> | NAS-Bench-101      | 4.6 10^5      |
> | MNAS-Net      | 5.6*10^13      |
> | PNAS      | 5.6*10^14      |
> | DARTS      | 10^18      |
> | Hierarchical NAS-Bench-201 (ours) | 10^446      |
>
> Also, we now conducted an additional experiment with a very different search space that we describe in more detail below.
>
> > No evidence that the additional expressivity of the search space offered by BANAT results in interesting novel architectures
>
> To strengthen our evidence for the expressivity and the ability of algebraic NAS spaces and BANAT to design novel architectural structures, we now also searched for activation functions based on basic mathematical operations. Following the search space from Ramachandran et. al (2017), the novel activation function we found achieves 8.31% test error compared to 8.93% (ReLU) and 8.61% (Swish).
>
> Also, the architectures we find in the hierarchical NAS-Bench-201 space are novel and interesting due to the heavy branching with complex resolution changes (we will add visualizations to the Appendix in the revised manuscript). These novel types of architectures can significantly outperform architectures found in the cell-based equivalent of our search space.
>
> > BANAT is only evaluated against simple baselines like random search and evolutionary search and not more adaptive competitors, in particular, other Bayesian NAS methods.
>
> Aside from random search and evolutionary search, we compared against the state-of-the-art Bayesian optimization method (i.e., NAS-BOWL applied to the cell-based search space (Figure 2) and the model from NAS-BOWL applied to our hierarchical space (Figure 3)). Further, we have now conducted a comparison to NASBOT, which is clearly outperformed by our proposed surrogate when applied to our search space (we will add this to Figure 4 in the revised manuscript). We have also tried the GCN surrogate proposed in BONAS, but it did not work, yielding only constant prediction demonstrating that they could not learn a high-quality mapping from the complex graph space to performance. Thus, we do not plot results for this as the correlation metrics are NaN.
>
> Besides that, we would like to note that we cannot apply adjacency matrix-based (Ying et al, ICML 2019; White et al, NeurIPS 2020) nor path-based encodings (White et al, AAAI 2021) work in our search spaces. The former requires the graph inputs with an equal number of nodes, while the latter is not scalable to large graphs like those in our search space since it is exponential in the number of nodes (i.e., r^i where r is the number of possible operations and i is the number of nodes).
>
> Finally, as suggested by another reviewer, we conducted correlation experiments with zero-cost proxies; see the tables in [our reply to this reviewer](https://openreview.net/forum?id=UIpwFLrJiDi&noteId=UDyHIU53zZ).
>
> > References
>
> (Ramachandran et al, arXiv 2017): Prajit Ramachandran, Barret Zoph, and Quoc V. Le. Searching for Activation Functions. arXiv, 2017.
>
> (Ying et al, ICML 2019): Chris Ying, Aaron Klein, Esteban Real, Eric Christiansen, Kevin Murphy, and Frank Hutter. Nas-bench-101: Towards reproducible neural architecture search. In Proceedings of the International Conference on Machine Learning (ICML), 2019.
>
> (White et al, NeurIPS 2020): Colin White, Willie Neiswanger, Sam Nolen and Yash Savani. A study on encodings for neural architecture search. Advances in Neural Information Processing Systems, 2020.
>
> (White et al, AAAI 2021): Colin White, Willie Neiswanger and Yash Savani. Bananas: Bayesian optimization with neural architectures for neural architecture search. Proceedings of the AAAI Conference on Artificial Intelligence, 2021

---

> > ### Comment · Reviewer_Hxvj · 2022-11-17
> > **Post Author Response**
> >
> > Thank you for your response and I appreciate the additional information provided on the search space size and the addition of the activation function search space.  However, I maintain that more compelling experiments are needed.
> >
> > Although the hierarchical search space is high dimensional for NASBench-201, it does not yield significantly better results than the cell-based search space for CIFAR-10, CIFAR-100, and ImageNet-16-120.  Furthermore, the authors show that CFG approach can be used to describe the DARTS and MobileNet search space in the appendix so why not show results for BANAT on one of those search spaces?
> >
> > If it's due to the cost of training on the order of hundreds of architectures from scratch, then I would argue BANAT is not efficient enough to be used in practice.  Alternatively, if BANAT is applied to a smaller hybrid hierarchical and cell-based search space, can it reduce the sample complexity (number of architectures) needed to find a good architecture to on the order of tens?  Is there a way to combine BANAT with supernet-based approaches to reduce search cost?

---

> > > ### Author Response · Authors · 2022-11-18
> > > **Response to Post Author Response**
> > >
> > > Thank you for responding to our initial response. We are glad to hear that you appreciate the additions during the rebuttal. Below, we want to address your questions and concerns.
> > >
> > > > Although the hierarchical search space is high dimensional for NASBench-201, it does not yield significantly better results than the cell-based search space for CIFAR-10, CIFAR-100, and ImageNet-16-120
> > >
> > > We would like to report the following additional results to demonstrate that we can find significantly better-performing architectures, not only on CIFARTile & AddNIST, but also on CIFAR-100 & ImageNet-16-120: In our ablation, when only allowing from the same primitives as in the cell-based NAS-Bench-201 search space (i.e., there is flexible parameterization of the convolutional blocks), we also do find an architecture on CIFAR-100, which is superior to the best architecture in the cell-based search space. Similarly, on ImageNet-16-120, with 50 additional evaluations, we find an architecture that improves upon the oracle solution (+0.7%p test error, 51.98% (ours) vs. 52.69% (oracle)). We would like to highlight that no search strategy, including BO, gradient-based or oracle, can find a better architecture on the cell-based search space, while we also could run even for longer and further widening the gap. This is a significant improvement over many previous approaches for cell-based spaces. As higher-dimensional and conditional search spaces are likely more difficult to search compared to smaller (cell-based) spaces, this shows that BANAT is indeed very efficient. Further, these strong results, on a majority of datasets, indicate that NAS research should further explore search spaces that are much more expressive than the common (low-dimensional, simple) cell-based ones.
> > >
> > > > Furthermore, the authors show that CFG approach can be used to describe the DARTS and MobileNet search space in the appendix so why not show results for BANAT on one of those search spaces?
> > >
> > > In Appendix E we showed the versatility of our search space design with CFGs. While we can represent those spaces, we did not consider them as they are minuscule and less expressive compared to hierarchical search spaces. To make them more interesting, we would need to modify them in a similar fashion as for the NAS-Bench-201 search space.
> > >
> > > > Is there a way to combine BANAT with supernet-based approaches to reduce search cost?
> > >
> > > Due to the exponential explosion in the supernet weights as a function of hierarchical levels, supernet-based approaches would require further adaptation to work in spaces with many hierarchical levels, such as the hierarchical NAS-Bench-201 space we used in the experiments. However, if those issues on the supernet side were resolved, combining with BANAT should be straightforward.
> > >
> > > (Ru et al, ICLR 2021): Binxin Ru, Xingchen Wan, Xiaowen Dong, and Michael Osborne. Interpretable Neural Architecture Search via Bayesian Optimisation with Weisfeiler-Lehman Kernels. In International Conference on Learning Representations, 2021.
> > >
> > > (Xiao et al, CVPR 2022): Han Xiao, Ziwei Wang, Zheng Zhu, Jie Zhou, and Jiwen Lu. Shapley-nas: Discovering operation contribution for neural architecture search. In Proceedings of the IEEE/CVF Conference on Computer Vision and Pattern Recognition, 2022.

---

> > > > ### Comment · Reviewer_Hxvj · 2022-11-22
> > > > **More realistic/large-scale experiments needed**
> > > >
> > > > Although the DARTS search space is smaller than the hierarchical search space used for NASBench-201, the existing NAS results for DARTS are much stronger.  This is also the case for the MobileNet family of search spaces.  I strongly encourage the authors to add additional experiments for flavors of these search spaces even if to just show results on the "simple" DARTS search space for which we do not have tabular lookup results (which we do have for NASBench-201 cell and NASBench-101, 1shot1, etc).  Given the absence of such results, I will maintain my score.

---

> > > > > ### Author Response · Authors · 2022-12-12
> > > > > **More realistic/large-scale experiments added**
> > > > >
> > > > > We thank you for your response and present results on the cell-based DARTS search space below. Due to the volatility of validation errors, we report the average of the last five validation errors across the search runs of our search strategy BANAT and, e.g.,  compare it to the validation errors of the genotypes found by NAS-BOWL and DrNAS. While the validation error more directly indicates the search strength, we also compare the final test error of different search algorithms.
> > > > >
> > > > > | Algorithm       | Avg. val error | Avg. test error | Best test error | #Params(M) | GPU days |
> > > > > | ------------- |:-------------:|:-------------:|:-------------:|:-------------:|:-------------:|
> > > > > | GP-NAS | - | - | 3.79 | 3.9 | 1 |
> > > > > | DARTS (v2) | - | 2.76 $\pm$ 0.09 | - | 3.3 | 4 |
> > > > > | ENAS | - | - | 2.89 | 4.6 | 6 |
> > > > > | ASHA | - | 3.03 $\pm$ 0.13 | 2.85 | 2.2 | 9 |
> > > > > | Random-WS | - | 2.85 $\pm$ 0.08 | 2.71 | 4.3 | 10 |
> > > > > | BANANAS | - | 2.64 | 2.57 | - | 12 |
> > > > > | LaNet | - | 2.53 $\pm$ 0.05 | - | 3.2 | 150 |
> > > > > | AmoebaNet-A | - | 3.34 $\pm$ 0.06 | - | 3.2 | 3150 |
> > > > > | AmoebaNet-B | - | 2.55 $\pm$ 0.05 | - | 2.8 | 3150 |
> > > > > | GDAS | - | 2.93 | - | 3.4 | 0.3 |
> > > > > | DrNAS | - | 2.46 $\pm$ 0.03 | - | 4.1 | 0.6 |
> > > > > | DrNAS (reproduced*) | 12.19 $\pm$ 0.06 | 2.65 $\pm$ 0.09 | 2.51 | 4.1 | 0.6 |
> > > > > | NAS-BOWL | - | 2.61 $\pm$ 0.08 | 2.50 | 3.7 | 3 |
> > > > > | NAS-BOWL (reproduced*) | 11.84 $\pm$ 0.15 | 2.93 $\pm$ 0.1 | 2.83 | 3.7 | 3 |
> > > > > | BANAT (ours) | 11.27 $\pm$ 0.16 | 2.68 $\pm$ 0.12 | 2.55 | 3.9 | 8 |
> > > > >
> > > > > *Note that reproduced for NAS-BOWL and DrNAS refers to re-running search and evaluation using their best-found genotypes (not entire search runs).
> > > > >
> > > > > The strong average validation error of our search strategy BANAT clearly shows that we can find well-performing normal and reduction cells in the cell-based DARTS search space. However, as discussed by many prior works, the rank correlation between search and evaluation can yield worse-performing cells in the evaluation setting. Nevertheless, our search strategy BANAT finds cells competitive to DrNAS. We would like to emphasize that compared to, e.g., DrNAS, our search strategy is more generic as it also works in vast hierarchical search spaces.
> > > > >
> > > > > Also, we would like to reaffirm that in all our experiments we did not use look-up results, e.g., our proposed hierarchical NAS-Bench-201 search space has no tabular lookup available and exhaustive evaluation is not feasible*. Thus, our previous experiments already qualify for “large-scale” and “realistic”.
> > > > >
> > > > > **The hierarchical NAS-Bench-201 search space, including isomorphic architectures, is a factor of ca. 10^5 larger than the number of atoms in the observable universe.*

---

### Official Review · Reviewer_Xuih · 2022-10-28

**Confidence:** 4
**Correctness:** 3
**Technical Novelty And Significance:** 2
**Empirical Novelty And Significance:** 2
**Recommendation:** 3

**Clarity, Quality, Novelty And Reproducibility:**

The paper is written clearly. (I have prior knowledge of CFGs, otherwise would have been harder to follow). The novelty is limited.

**Strength And Weaknesses:**

Strength:
1. This paper defines a language for defining search spaces. Their grammar can be used for all the search spaces.
2. Their search algorithm is able to find better architectures faster than others in hierarchical NAS-Bench 201 search space (as they are using a surrogate model).
3. In their empirical evaluations, it was highlighted that while cell-based search space works well for cifar-10, cifar-100 and imagenet-16-120, it does not perform well on CIFARTile or AddNIST.

** Comments / Questions **
It is a bit hard to understand why we need a special grammar to design the search space. The authors claim it is more flexible and can discover new architectures. But one still needs to design the primitives and how they are connected together. So how can we really discover completely new architectures?
1. Hierarchical search space is much larger than cell spaced search space and is used for scenarios where one is interested in finding architectures with low latency, number of flops etc. So it is not surprising that BANAT was able to discover architectures with lesser number of parameters. It would have been better if they had actually demonstrated how it fared in multi-objective optimization problems and compared against other hierarchical baselines such as MNASNet.
3. The authors claim that their search space design is more flexible than other baselines and is especially beneficial for object detection setting. But they never ran BANAT to find architectures for object detection.
4. For figure4, Kendall Tau is the most commonly metric. Please use that.
5. Please use a diagram to elucidate how the hierarchical search space looks like
6. Please specify the details about bayesian optimization such as the acquisition function used in the main paper rather than in the appendix.
7. The time taken for Imagenet-16-120 is 1.8 * 8 GPU days. Please specify that explicitly.

**Summary Of The Paper:**

This paper proposes using context free grammars to define a search space. This grammar is very flexible and one can use it to define all kinds of search spaces. The production rules can be formed to imposed constraints as well. Their search algorithm, BANAT is a bayesian optimization based one, where the surrogate model is a hierarchical Weisfeiler-Lehman kernel (adapted Weisfeiler-Lehman kernel used by NASBOWL to make it work for their search space) and the acquisition function is expected improvement. They used a hierarchical NASBENCH 201 search space.

**Summary Of The Review:**

 The main novelty is the ability to define a context free grammar for a given search space. It is not evident if this is actually useful in practice. The surrogate model used by BANAT is a minor adaptation of NASBOWL paper which used Weisfeiler-Lehman for a regular NASBENCH search space to make it work for hierarchical setting. They also need to compare their algorithm against other hierarchical search space based algorithms.

---

> ### Author Response · Authors · 2022-11-12
> **Response to Reviewer Xuih 1/2**
>
> We would like to thank you for taking the time to review this paper and make constructive criticisms. To facilitate an active discussion, we reply to you now and include updates to the paper shortly.
>
> > So it is not surprising that BANAT was able to discover architectures with a lesser number of parameters.
>
> We find it surprising that we can find an architecture, which has *on par* performance with the best architecture (oracle) on the NAS-Bench-201 cell-based search space while having *ca. 27% fewer parameters*; not only fewer parameters. Note that this architecture is *better* than the one that has been found with the state-of-the-art Shapley NAS search strategy (Xiao et al, CVPR 2022) in terms of performance. Also, we find better-performing architectures in our hierarchical search space compared to the cell-based search space in our experiments, even though the search problem is more difficult due to the very large size of our search space.
>
> > “The surrogate model [...] is a minor adaptation of NASBOWL”
>
> We agree that our technique to explicitly incorporate hierarchical information requires only a simple modification to the GPWL surrogate in NASBOWL (Shervashidze et. al, JMLR 2011; Ru et al, ICLR 2021), but this modification neatly leverages the topological information at different hierarchical levels from the architecture assembly, which are ignored in GPWL surrogate. We would like to highlight that this modification significantly improves both search and surrogate regression performance, while inheriting all the nice features, such as interpretability, from the WL kernel. With this impact on the performance, the simplicity of our technique is a virtue. Thus, given our empirical evidence, this may be an important ingredient to consider when designing search methods for hierarchical search spaces, e.g., GNNs, since prior works have not explored the incorporation of the hierarchical nature of architectures for modeling.
>
> > There is still the requirement of defining primitives and wirings. So how can you really discover completely novel architectures?
>
> Yes, any algorithm requires some atomic building blocks, e.g., from basic mathematical operators. To demonstrate that our approach can discover structures from very basic primitives, we conducted an additional experiment, where we searched for activation functions based on basic mathematical operations. Following the search space from Ramachandran et. al (2017), the novel activation function we found achieves 8.31% test error compared to 8.93% (ReLU) and 8.61% (Swish) with a ResNet-20.
>
> > The authors claim that their search space design is more flexible than other baselines and is especially beneficial for object detection settings. But they never ran BANAT to find architectures for object detection.
>
> We don’t claim that our work may be especially beneficial for object detection but is a general approach to designing search spaces for a variety of tasks and modalities.
>
> > Compare the algorithm against other hierarchical search space algorithms
>
> We would appreciate more detailed proposals on what kind of algorithms you have in mind. We would like to note that, aside from random search, we already compared against regularized evolution (i.e., used, e.g., in Liu et al, ICLR 2018 as well as it is more generally representing evolutionary algorithms used, e.g., in almost all grammar-based NAS approaches from our related work), and the state-of-the-art Bayesian optimization method (i.e., NAS-BOWL and generally representing Bayesian optimization methods). We agree that there are works that use reinforcement learning (e.g., MnasNet) as well as gradient-based methods. However, reinforcement learning is (arguably) not sample efficient and gradient-based methods do not scale to our hierarchical search space due to the exponential number of weights the supernet would have.

---

> > ### Author Response · Authors · 2022-11-12
> > **Response to Reviewer Xuih 2/2**
> >
> > > Please use a diagram to elucidate how the hierarchical search space looks like
> >
> > Could you please elaborate on what such a diagram should show? We provide visualizations and elaboration of the primitive computations and topological operators in Appendix A of the initial submission.
> >
> > > Please specify the details about bayesian optimization such as the acquisition function
> > used in the main paper rather than in the appendix.
> >
> > Do you refer to the Equation from Appendix F.1 of the initial submission here? EI is explicitly mentioned in the overview paragraph in Section 4 of the initial submission.
> >
> > > The time taken for Imagenet-16-120 is 1.8 * 8 GPU days. Please specify that explicitly. [...] For figure4, Kendall Tau is the most common metric. Please use that.
> >
> > We agree and now adopt these.
> >
> > > References
> >
> > (Liu et al, ICLR 2018): Hanxiao Liu, Karen Simonyan, Oriol Vinyals, Chrisantha Fernando, and Koray Kavukcuoglu. Hierarchical Representations for Efficient Architecture Search. In International Conference on Learning Representations, 2018.
> >
> > (Ramachandran et al, arXiv 2017): Prajit Ramachandran, Barret Zoph, and Quoc V. Le. Searching for Activation Functions. arXiv, 2017.
> >
> > (Ru et al, ICLR 2021): Binxin Ru, Xingchen Wan, Xiaowen Dong, and Michael Osborne. Interpretable Neural Architecture Search via Bayesian Optimisation with Weisfeiler-Lehman Kernels. In International Conference on Learning Representations, 2021.
> >
> > (Shervashidze et al, JMLR 2011): Nino Shervashidze, Pascal Schweitzer, Erik Jan Van Leeuwen, Kurt Mehlhorn, and Karsten M Borgwardt. Weisfeiler-Lehman Graph Kernels. Journal of Machine Learning Research, 2011.
> >
> > (Xiao et al, CVPR 2022): Han Xiao, Ziwei Wang, Zheng Zhu, Jie Zhou, and Jiwen Lu. Shapley-nas: Discovering operation contribution for neural architecture search. In Proceedings of the IEEE/CVF Conference on Computer Vision and Pattern Recognition, 2022.

---

> > > ### Comment · Reviewer_Xuih · 2022-12-08
> > > **Response**
> > >
> > > I apologize for the delayed response. I wanted to clarify a few things:
> > >
> > > 1. When you meant you can discover new architectures and the paper was was titled "from scratch", the readers would imagine coming up with architectures when the building blocks are just operations, along the lines of automl-zero.[1]
> > > 2. While I do agree that your method found architecture with fewer parameters than ShapelyNAS, I was stating the fact that the reason hierarchical search spaces have been used in the past was to find architectures with fewer parameters as is demonstrated by MNasNet. Hierarchical search space provides more control to select a particular block at each stage.
> > > 3. I overlooked figure 5 which illustrates the cell. In general, it would be good to represent the search space in a block diagram as illustrated in Figure 4 of Mnasnet paper or figure 5 of once-for-all paper
> > > 4. Thank you for demonstrating that your algorithm can be used for discovering new activation functions.
> > >
> > > While the idea of using CFG to represent a search space is definitely interesting, the novelty is limited. I stand by my score.
> > >
> > > [1] AutoML-Zero: Evolving Machine Learning Algorithms From Scratch, Real et al.
> > > [2] Once-for-all: Train one network and specialize it for efficient deployment. , Cai et al.

---

> > > > ### Author Response · Authors · 2022-12-12
> > > > **Adressing comments**
> > > >
> > > > Thanks for your response. We want to address your remaining comments below:
> > > >
> > > >
> > > > > Novelty of contributions and discovery from elementary operations
> > > >
> > > > CFGs give us a formally grounded way to naturally, quickly, and compactly define hierarchical search spaces at arbitrary abstraction levels, e.g., for the activation function search we start off from elementary mathematical operations, but we can also work with higher-level (both operational as well as topological) building blocks, e.g., as done in our proposed hierarchical NAS-Bench-201 search space. To further extend our construction mechanism, we proposed simple and efficient techniques (i) to incorporate constraints (i.e., desiderata we want), (ii) foster regularity (i.e., optionally re-use a sub-design like cells or activation functions across the architecture) or (iii) to handle spatial resolutions (or channels) without running into shape mismatches. While these techniques are simple to implement, taken together, they allow us to provide a unifying framework for search space designs (e.g., see Appendix E on how to design search spaces from the literature). To the best of our knowledge, no previous search space design method was able to unify the most common search spaces from the literature as well as describe complex hierarchical search spaces in one single framework. To fully leverage this unifying construction technique, we pair it with a generic search method that explicitly models performance across different abstraction levels.
> > > >
> > > > > "Hierarchical search spaces have been used in the past [...] to find architectures with fewer parameters"
> > > >
> > > > We agree that prior works on hierarchical NAS often focused on a multi-objective setting and hierarchical spaces are a good fit for that, e.g., MnasNet starts from a reference architecture and trades off performance and latency (or possibly other factors). However, our results show that hierarchical spaces are also a good choice to efficiently find *well-performing architectures* that are *qualitatively novel* compared to existing architecture designs (see pages 38-42 in the Appendix).
> > > >
> > > > > Diagram for search space
> > > >
> > > > Thank you for your helpful suggestion. We will add graphical representations of the CFGs (e.g., Equation 10) to make the connection to the graphical representation of architectures more explicit.

---

### Author Response · Authors · 2022-11-15
**Changes to the revised manuscript**

We want to thank all reviewers for their constructive criticisms. We have now uploaded the revised manuscript and want to summarize the main additions:

1. Extended our experiments beyond our hierarchical NAS-Bench-201 search space with an additional activation function search (Section 5.3).
2. Extended our experiments to include extensive comparisons with zero-cost proxies. We further compared with other BO NAS methods (if applicable, if not, justified why we cannot apply it in our hierarchical search space) (Section 5.3).
3. Improved and merged the method sections 2 & 3 of the initial submission into one section (Section 2).
4. Added further justification why CFGs are a well-suited framework for hierarchical search space design (Section 2.2).
5. Demonstrated the novel architecture design of our best found architectures (as well as a novel activation function) (Section 5.3, Appendix I.2 & J.3).
6. Demonstrated the importance of flexible parameterization of convolutional blocks; except for CIFAR-100, where, in contrast, we find an architecture superior to the best architecture in the cell-based NAS-Bench-201 search space when we do not allow for flexible parameterization of convolutional blocks (Appendix I.2).
7. Contextualized our work additionally in domains beyond (pure) NAS (Appendix G).

We believe that all of these points greatly improve the manuscript. We want to thank the reviewers for raising these points and look forward to further discussions to improve the work.

---

> ### Author Response · Authors · 2022-12-12
> **Additional experiments**
>
> To incorporate new feedback from the reviewers, we additionally performed the following experiments and will add them to the manuscript:
>
> 8. Added experiments with the DARTS search space. More details can be found in [this response](https://openreview.net/forum?id=UIpwFLrJiDi&noteId=WQsatjQ0mCm).
>
> 9. Added an additional baseline (AREA). See [this response](https://openreview.net/forum?id=UIpwFLrJiDi&noteId=B0TqzRhiYc) for more details.
>
> We hope these additional experiments clear up the remaining requests of the reviewers.

---

### Decision · Program_Chairs · 2023-01-20

**Decision:**

Reject

**Justification For Why Not Higher Score:**

N/A

**Justification For Why Not Lower Score:**

N/A

**Metareview: Summary, Strengths And Weaknesses:**

The paper presents a new approach to representing neural architectures as algebraic terms and the design space as context-free grammars (CFGs). While the formulation is novel, natural and flexible, the experimental results are limited to small search spaces that do not achieve state-of-the-art accuracy. BANAT is only compared to simple baselines, and there is no evidence that the additional expressivity of the search space leads to novel architectures.

To strengthen the case for the usefulness of method, it would be beneficial to conduct experiments on other search spaces that allow for state-of-the-art results. This would provide a more comprehensive evaluation of effectiveness and allow for a more convincing comparison to other NAS methods, such as HNAS, BANANAS, and NASBOT.

Overall, the CFG representation of NAS search spaces is a promising approach, but additional experimentation is needed to fully demonstrate its potential. We encourage the authors to address these limitations and believe that the paper can be made a valuable contribution to the field.